
# Model evaluation of short-lived climate forcers for the Arctic Monitoring and Assessment Programme: a multi-species, multi-model study

Cynthia H. Whaley[1], Rashed Mahmood[2,30], Knut von Salzen[1], Barbara Winter[3], Sabine Eckhardt[4], Stephen Arnold[5], Stephen Beagley[6], Silvia Becagli[12], Rong-You Chien[7], Jesper Christensen[8], Sujay Manish Damani[1], Kostas Eleftheriadis[29], Nikolaos Evangeliou[4], Greg Faluvegi[9,10], Mark Flanner[11], Joshua S. Fu[7], Michael Gauss[12], Fabio Giardi[13], Wanmin Gong[6], Jens Liengaard Hjorth[8], Lin Huang[6], Ulas Im[8], Yugo Kanaya[14], Srinath Krishnan[24], Zbigniew Klimont[15], Thomas Kühn[16,17], Joakim Langner[18], Kathy S. Law[19], Louis Marelle[19], Andreas Massling[8], Dirk Olivié[12], Tatsuo Onishi[19], Naga Oshima[20], Yiran Peng[21], David A. Plummer[3], Olga Popovicheva[22], Luca Pozzoli[23], Jean-Christophe Raut[19], Maria Sand[24], Laura N. Saunders[25], Julia Schmale[26], Sangeeta Sharma[6], Henrik Skov[8], Fumikazu Taketani[14], Manu A. Thomas[18], Rita Traversi[13], Kostas Tsigaridis[9,10], Svetlana Tsyro[12], Steven Turnock[28,5], Vito Vitale[23], Kaley A. Walker[25], Minqi Wang[21], Duncan Watson-Parris[27], and Tahya Weiss-Gibbons[1]

[1]Canadian Centre for Climate Modelling and Analysis, Environment and Climate Change Canada, Victoria, BC, Canada.
[2]Barcelona Supercomputing Center, Spain.
[3]Canadian Centre for Climate Modelling and Analysis, Environment and Climate Change Canada, Dorval, QC, Canada.
[4]NILU - Norwegian Institute for Air Research, Kjeller, Norway.
[5]Institute of Climate and Atmospheric Science, School of Earth and Environment, University of Leeds, Leeds, United Kingdom.
[6]Climate Chemistry Measurements and Research, Environment and Climate Change Canada, Toronto, ON, Canada.
[7]University of Tennessee, Knoxville, Tennessee, United States.
[8]Department of Environmental Science/Interdisciplinary Centre for Climate Change, Aarhus University, Frederiksborgvej 400, Roskilde, Denmark.
[9]NASA Goddard Institute for Space Studies, New York, NY, USA.
[10]Center for Climate Systems Research, Columbia University; New York, USA.
[11]University of Michigan, Ann Arbor, MI, United States.
[12]Norwegian Meteorological Institute, Oslo, Norway.
[13]Dept. of Chemistry, University of Florence, Florence, Italy.
[14]Japan Agency for Marine-Earth Science and Technology, Yokohama, Japan.
[15]International Institute for Applied Systems Analysis, Laxenburg, Austria.
[16]Department of Applied Physics, University of Eastern Finland, Kuopio, Finland.
[17]Atmospheric Research Centre of Eastern Finland, Finnish Meteorological Institute, Kuopio, Finland.
[18]Swedish Meteorological and Hydrological Institute, Norrköping, Sweden.
[19]LATMOS, CNRS-UVSQ-Sorbonne Université, Paris, France.
[20]Meteorological Research Institute, Japan Meteorological Agency, Tsukuba, Japan.
[21]Department of Earth System Science, Ministry of Education Key Laboratory for Earth System Modeling, Institute for Global Change Studies, Tsinghua University, Beijing, China.
[22]Moscow State University, Moscow, Russia
[23]European Commission, Joint Research Centre, Ispra, Italy.
[24]CICERO Center for International Climate and Environmental Research, Oslo, Norway.
[25]University of Toronto, Toronto, ON, Canada.





[26]Extreme Environments Research Laboratory, École Polytechnique Fédérale de Lausanne; Lausanne, Switzerland.

[27]Atmospheric, Oceanic and Planetary Physics, Department of Physics, University of Oxford, Oxford, UK.

[28]Met Office Hadley Centre, Exeter, UK.

[29]Institute of Nuclear and Radiological Science & Technology, Energy & Safety N.C.S.R. "Demokritos", Attiki, Greece.

[30]Department of Geography, University of Montreal, Montreal, QC, Canada.

**Correspondence:** Cynthia Whaley (cynthia.whaley@ec.gc.ca)

**Abstract.** The Arctic atmosphere is warming rapidly and its relatively pristine environment is sensitive to the long-range transport of atmospheric pollutants. While carbon dioxide is the main cause for global warming, short-lived climate forcers (SLCFs) such as methane, ozone, and particles also play a role in Arctic climate on near-term time scales. Atmospheric modelling is critical for understanding the abundance and distribution of SLCFs throughout the Arctic atmosphere, and is used as a tool

towards determining SLCF impacts on climate and health in the present and in future emissions scenarios. In this study, we evaluate 18 state-of-the-art atmospheric and Earth system models, assessing their representation of Arctic and Northern Hemisphere atmospheric SLCF distributions, considering a wide range of different chemical species (methane, tropospheric ozone and its precursors, black carbon, sulfate, organic aerosol, and particulate matter) and multiple observational datasets. Model simulations over four years (2008-2009 and 2014-2015) conducted for the 2021 Arctic Monitoring and Assessment Programme

(AMAP) SLCF assessment report are thoroughly evaluated against satellite, ground, ship and aircraft-based observations. The results show a large range in model performance, with no one particular model or model type performing well for all regions and all SLCF species. The multi-model mean was able to represent the general features of SLCFs in the Arctic, though vertical mixing, long-range transport, deposition, and wildfire emissions remain highly uncertain processes. These need better representation within atmospheric models to improve their simulation of SLCFs in the Arctic environment.

# 1  Introduction

The Arctic atmosphere is warming 3 times more quickly than the global average (Bush and Lemmen, 2019; NOAA, 2020; AMAP, 2021b; IPCC, 2021). Arctic warming is a manifestation of global warming and the main driver for this is the increasing carbon dioxide ($CO_2$) radiative forcing (IPCC, 2021). Arctic warming is amplified by sea ice and snow feedbacks and affected by local radiative forcings in the Arctic, including radiative forcings by short-lived climate forcers (SLCFs), such as methane,

black carbon, and tropospheric ozone (AMAP, 2015a,b, 2021a). The remote pristine Arctic environment is sensitive to the long-range transport of atmospheric pollutants and deposition (Schmale et al., 2021b). At the same time, it is difficult to carry out in situ measurements (Nguyen et al., 2016; Freud et al., 2017) and satellite observations over the Arctic. The majority of the Arctic surface is ocean, covered with sea-ice that is usually adrift for most part of the year. The Arctic environment is also harsh. These aspects have historically kept surface based measurements sparse. The overwhelming majority of the satellite

observations either depend on the visible spectrum, are limited by the presence of clouds, or have very low sensitivity in the lower troposphere where the atmospheric processes mainly determine the fate of the pollutants. Many satellite measurements also do not have good coverage in the Arctic, given their orbital parameters or problems measuring areas with high albedo (Beer, 2006).





Modelling the Arctic atmosphere comes with its own challenges due to extreme meteorological conditions, its great distance
from major global pollution sources, poorly known local emissions, high gradients in physical and chemical fields, and a
singularity in some model grids at the pole. Models have been improving in the last two decades, but many models still have
inaccurate results in the Arctic (Shindell et al., 2008; Eckhardt et al., 2015; Emmons et al., 2015; Sand et al., 2017; Marelle
et al., 2018). That said, there has recently been a number of improvements in numerous models that have allowed for better
representation of certain processes (Morgenstern et al., 2017; Emmons et al., 2020a; Swart et al., 2019; Holopainen et al., 2020;
Im et al., 2021). In this study, model simulations for the 2021 Arctic Monitoring and Assessment Programme (AMAP) SLCF
assessment report (AMAP, 2021a), have been thoroughly evaluated by comparison to several freely available observational
datasets in the Northern Hemisphere, and assessed in more detail in the Arctic. In order to support the integrated assessment
of climate and human health for AMAP, 6 SLCF species (methane $CH_4$; ozone $O_3$; black carbon BC; sulfate $SO_4^{2-}$; organic
aerosol OA; and fine particulate matter $PM_{2.5}$), and 2 $O_3$ precursors (carbon monoxide CO; and nitrogen dioxide $NO_2$) from
18 atmospheric or Earth system models are compared to numerous observational datasets (from 3 satellite instruments, 7
monitoring networks, and 9 measurement campaigns) for four years (2008-2009 and 2014-2015), with the goal of answering
the following questions:

1. How well do the AMAP SLCF models perform in the context of measurements and their associated uncertainty?:

   What do the best-performing models have in common?

   Are there regional patterns in the model biases?

   Are there patterns in the model biases between SLCF species?

2. How does the model performance impact model applications, such as simulated climate and health impacts?

3. What processes should be improved or studied further for better model performance?

Out of scope of this study are any sensitivity tests by the models to assess different components of model errors. Also out of
scope are the models' simulations of aerosol optical properties and cloud properties (e.g., cloud fraction, cloud droplet number
concentration, cloud scavenging, etc), though those parameters do have a large impact on climate and a tight relationship with
some SLCFs. Their initial evaluation can be found in AMAP (2021a) (Chapter 7). Estimates of effective radiative forcings of
SLCFs in the Arctic by the AMAP participating models are also provided elsewhere (Oshima et al., 2020).

The next section summarizes the models used in this study, with more information in the appendix. Section 3 summarizes
the measurements used for model evaluation. Section 4 presents our model evaluation for each SLCF species, followed by a
summary of all SLCFs. Finally, Section 5 is the conclusion where the questions posed above are answered.

## 2  Models

In this section we briefly describe the models used for the AMAP SLCF study and refer the reader to Appendix A for individual
model descriptions and further information. All models were run globally with the same anthropogenic emissions dataset (see





Section 2.1), and most were run for the years 2008-2009 (as was done for the 2015 AMAP assessment report) and 2014-2015 (to evaluate more recent model results) inclusive for this evaluation, as these were years with numerous Arctic measurements. Unless otherwise indicated, all model output was monthly-averaged.

    The models used for this study are summarized in Table 1. As is shown in the table, not all models provided all SLCF species, and not all models provided all four years. There were 8 chemical transport models (CTMs), 2 chemistry climate

models (CCMs), 3 global climate models (GCMs), and 5 Earth system models (ESMs). Many models used specified or nudged meteorology, which allows the day-to-day variability of the model meteorology to be more closely aligned with the historical evolution of the atmosphere than occurs in a free-running model. The ERA-Interim reanalysis was the most commonly used meteorology (in 7 out of 18 models) but some were free-running (simulating their own meteorology) and some used other reanalysis products (Table 1).

**2.1   Emissions**

  All models used the same anthropogenic emissions dataset, which is called ECLIPSE (Evaluating the Climate and Air Quality Impacts of Short-Lived Pollutants) v6B. These were created using the IIASA-GAINS (International Institute for Applied Systems Analysis - Greenhouse gas - Air pollution Interactions and Synergies) model (Amann et al., 2011; Klimont et al., 2017; Höglund-Isaksson et al., 2020), which provides emissions of long-lived greenhouse gases and shorter-lived species in a

consistent framework. These historical emissions were provided for the years 1990 to 2015 at 5-year intervals, as well as the years 2008-9 and 2014. Those models that simulated the 1990-2015 time periods linearly interpolated the emissions for the years in between. The ECLIPSEv6b emissions include many pollutants, such as $CH_4$, CO, $NO_x$, BC, and $SO_2$. They include the significant sulfur emission reductions which have taken place since the 1980s (Grennfelt et al., 2020). Global anthropogenic BC emissions are estimated to be 6.5 Tg in 2010 and 5.9 Tg in 2020, and global anthropogenic $SO_2$ emissions are estimated to

be 90 Tg in 2010 but declined significantly over the subsequent decade to 50 Tg (AMAP, 2021a). The reductions are mainly due to stringent emissions standards in the energy and industrial sectors, and reduced coal use in the residential sector (AMAP, 2021a). Global anthropogenic methane emissions were 340 Tg in 2015 and 350 Tg in 2020, and are expected to continue to increase, unlike BC and $SO_2$. The largest methane sources in 2015 were agriculture (42% of total emissions), oil and gas (extraction and distribution) (18%), waste (18%) and energy production (including coal mining) (16%) (AMAP, 2021a;

Höglund-Isaksson et al., 2020). CO and $NO_x$ emissions have been declining steadily and are expected to continue declining in the future.

    In comparison to the CMIP6 emissions (Hoesly et al., 2018), ECLIPSEv6b emissions have additionally taken into account the recent declines in emissions from Asia of $SO_2$, BC and $NO_x$ due to recent control measures. Whereas those declines in the CMIP6 emissions were unrealistically small (Wang et al., 2021; von Salzen et al., 2021). The inclusion of emissions from the

flaring sector in Russia was a significant improvement which was not present in the previous version of ECLIPSE emissions that was used in the AMAP (2015a) report.

    For non-agricultural fire emissions, many models utilized the CMIP6 fire emissions, which are based on monthly GFED (Global Fire Emissions Database) v4.1 (van Marle et al., 2017). About half of the models included volcanic emissions or





stratospheric aerosol concentrations from the CMIP6 dataset (Thomason et al., 2018) or other sources, and the other half

did not include volcanic emissions, which mainly impact $SO_2$ and thus, modelled $SO_4^{2-}$. The emissions from the October to December 2014 Honoluraun volcano eruption (Gíslason et al., 2015; Twigg et al., 2016; Ilyinskaya et al., 2017) were included by six models in a separate set of simulations. Similar differences in biogenic and agricultural waste emissions appear in these model simulations, and all are summarized in Table 2.

## 2.2 Chemistry

This section contains a summary of models' chemistry schemes, and we refer the reader to Appendix A and references therein for more details.

### 2.2.1 Methane

All participating models that provided $CH_4$ output prescribed $CH_4$ concentrations based on box model results from Olivié et al. (2021) for 2015 and from Meinshausen et al. (2017) for years prior to 2015. The former utilised the ECLIPSE v6B

anthropogenic $CH_4$ emissions (Section 2.1), along with assumptions for the natural emissions (Olivié et al., 2021; Prather et al., 2012) to provide as input to models' surface or boundary layer $CH_4$ concentrations. Models then allow $CH_4$ to take part in photochemical processes, such as the production of tropospheric $O_3$.

### 2.2.2 Tropospheric chemistry

There is a wide range of tropospheric gas-phase chemistry implemented in the models. Air quality-focused models, such

as DEHM, EMEP MSC-W, GEM-MACH, GEOS-Chem, MATCH, and WRF-Chem have detailed HOx-NOx-hydrocarbon $O_3$ chemistry, with speciated volatile organic compounds (VOCs), and secondary aerosol formation. The GISS-E2.1, MRI-ESM2, and UKESM1 ESMs also use this level of tropospheric chemistry. Whereas, climate-focused models like CanAM5-PAM, CIESM-MAM7, ECHAM-SALSA, and NorESM1 contain bare minimum gas-phase chemistry, and use prescribed $O_3$ fields (e.g., CanAM5-PAM uses CMAM climatological $O_3$ fields). The CCMs are somewhere in between, with simplified

tropospheric and stratospheric chemistry so that they could be run for longer time periods. For example CMAM's tropospheric chemistry consists only of $CH_4$-$NO_x$-$O_3$ chemistry, with no VOCs.

### 2.2.3 Stratospheric chemistry

Only a subset of the participating models have a fully simulated stratosphere. CMAM, MRI-ESM2, GISS-E2.1, OsloCTM, and UKESM1 contain a relatively complete description of the HOx, NOx, Clx and Brx chemistry that controls stratospheric ozone

along with the longer-lived source gases such as $CH_4$, $N_2O$ and CFCs. Other models have a simplified stratosphere, such as GEOS-Chem which has a linearized stratopheric chemistry scheme (Linoz, McLinden et al. (2000)), and WRF-Chem which specifies stratospheric concentrations from climatologies - both of which do not simulate stratospheric chemistry. Finally,



several models have no stratosphere or stratospheric chemistry at all (e.g., CIESM-MAM7, GEM-MACH, DEHM, and EMEP MSC-W).

### 2.2.4 Aerosols

Most models contain speciated aerosols; mineral dust (also known as crustal material), sea salt, BC, OA (sometimes separated into primary and secondary), $SO_4^{2-}$, nitrate ($NO_3^-$), and ammonium ($NH_4^+$). Though some, like CanAM5-PAM and UKESM1, do not simulate $NO_3^-$ and $NH_4^+$, but assume all is in the form $(NH_4)_2SO_4^{2-}$. OA, $SO_4^{2-}$, $NO_3^-$, and $NH_4^+$ are involved in chemical reactions interacting with the gas-phase chemistry. Aerosol size distributions are either prescribed or discretized into log-normal modes or size sections. How the aerosol size distribution varies in space and time depends on many different processes, including emission, aerosol microphysics, aerosol-cloud interactions, and removal. How these processes are parameterized depends on the model and we refer the reader to the appendix and the references therein for more detail.

## 3 Measurements

We have utilized many freely available observational datasets of SLCFs to evaluate the models with. General descriptions are given below under the broad headings of surface monitoring, satellite, and campaign datasets, and there is some additional information in Appendix B.

### 3.1 Surface monitoring datasets

#### 3.1.1 CH$_4$ and O$_3$

Global surface $CH_4$ measurements were obtained from the World Data Centre for Greenhouse Gases (WDCGG). These measurements were made via gas chromatography, which has a $<1\%$ uncertainty range. Surface in situ $O_3$ measurements are typically made via various types of UV absorption monitors, employing the Beer-Lambert law to relate UV absorption of $O_3$ at 254 nm directly to the concentration of $O_3$ in the sample air (e.g., Bauguitte (2014)), which have approximately 3% or 1-2 ppbv uncertainty range. We obtained surface $O_3$ measurements from various networks: the National Air Pollutant Surveillance Program (NAPS) and the Canadian Pollutant Monitoring Network (CAPMON) for Canada; the Chemical Speciation Network (CSN) for the U.S.; the Beijing Air Quality and Hong Kong Environmental Protection Agency for China; the Climate Monitoring and Diagnostics Laboratory (CMDL) for some global sites; the European Monitoring and Evaluation Programme (EMEP); and some individual Arctic monitoring stations like Villum Research Station and Zeppelin Mountain. Many of these measurements were downloaded from the EBAS database. The Arctic $O_3$ measurement locations are shown in Figure 1.

#### 3.1.2 CO, NO, and NO$_2$

CO and $NO_x$ measurements were obtained from the same monitoring networks as $O_3$. CO instrumentation is similar to that for $O_3$, however, it uses gas filter correlation to relate infrared absorption of CO at 4.6 $\mu$m to the concentration of CO in the

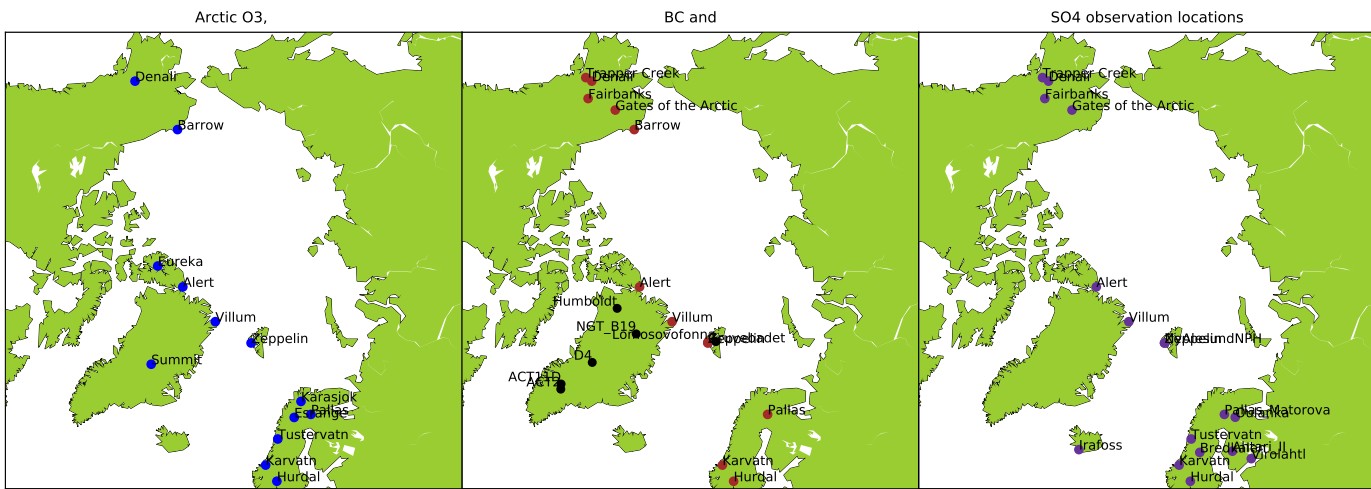

**Figure 1.** Location for Arctic surface in situ measurement locations. (left) $O_3$, (middle) BC in brown and ice cores in black, and (right) $SO_4^{2-}$.

sample air (Biraud, 2011). For $NO_x$, the instrument deploys the characteristic chemiluminescence produced by the reaction between NO and $O_3$, the intensity of which is proportional to the NO concentration. $NO_2$ measurements are approximated using its thermal reduction to NO by a heated (350°C) molybdenum converter (Bauguitte, 2014). Note that this method has an

estimated bias of about 5-20% because of sensitivity to other oxidized nitrogen species, and this has not been corrected for. The bias is on the lower end for high-$NO_x$ conditions, and in the low-$NO_x$ Arctic, can be up to 100% uncertainty.

### 3.1.3 BC and OA

There are various BC measurement methods, exploiting different properties of BC and thus measuring different quantities (Petzold et al., 2013); Elemental carbon (EC) determined by thermal/thermal-optical methods, equivalent BC (eBC) by op-

tical absorption methods, and refractory BC (rBC) by incandescence methods. Table B1 in Appendix B lists the different measurement techniques/instruments that the different monitoring networks and individual Arctic monitoring stations use. As BC emission inventories, including ECLIPSEv6b, are mainly based on emission factors derived from thermal/thermal-optical methods, modelled BC is thus representative of EC.

      The different types of BC measurements (EC, eBC, and rBC) usually agree with each other within a factor of two (AMAP,

2021a; Pileci et al., 2021). Though, it has been shown that, as the aerosol ages, the complex state of mixing of BC particles





causes eBC to increase relative to EC (Zanatta et al., 2018). The absorption and scattering cross sections of coated BC particles vary by more than a factor of two due to different coating structures. He et al. (2015) found an increase of 20-250% in absorption during aging, significantly depending on coating morphology and aging stages. Thus, this complexity impacts model-measurement comparisons at remote Arctic locations where one would expect eBC to have a high, positive uncertainty.

We obtained BC from the Canadian Aerosol Baseline Measurement (CABM) network for Canada; Interagency Monitoring of Protected Visual Environments (IMPROVE) network for the U.S.; EMEP network for Europe; as well as individual Arctic locations. To our knowledge, there were no other freely accessible BC measurements. The major observing networks EMEP, CABM, and IMPROVE measure EC with approximately 10% uncertainty (Sharma et al., 2017). However, given the complexities in different BC measurement types, as mentioned above, the overall uncertainty is about 200%.

Another complexity with model evaluation of BC is that some of the eBC measurements that models are compared to were made from collected particulate matter with different maximum diameters (e.g., $PM_1$, $PM_{2.5}$, and $PM_{10}$). These are included in Table B1 for each of the measurement locations. From the models we use BC from $PM_{2.5}$, as most of the BC is expected to be in the submicron mode.

    Organic Carbon (OC) is also measured via thermal/thermal-optical methods (Chow et al., 1993, 2001, 2004; Huang et al., 2006; Cavalli et al., 2010; Chan et al., 2019; Huang et al., 2021) using the same instrumentation as for EC detection in IMPROVE, CABM, NAPS, and EMEP measurement networks. These OC measurements have approximately 20% or less uncertainty (Chan et al., 2019). Models output organic aerosol (OA), which includes OC and organic matter and is related to OC via a factor of 1.4 (Russell, 2003; Tsigaridis et al., 2014), though this factor has been reported as a range from 1.4 to 2.1 in the literature, depending on the source of OC/OA (Tsigaridis et al., 2014). Nevertheless, we applied a conversion factor of 1.4 to the OC measurements before comparing the modelled OA.

    Arctic BC measurement locations are shown in Figure 1, many of these Arctic aerosol measurements were discussed in Schmale et al. (2021a). We also evaluated modelled BC deposition by comparing it to BC deposition derived from ice core measurements (D4, ACT2: McConnell and Edwards (2008); Humboldt: Bauer et al. (2013); Summit: Keegan et al. (2014); NGT_B19, ACT11d: McConnell et al. (2019)). All of the ice core locations are also shown in Figure 1. Deposition fluxes are not a measured value, but, are derived from the EC concentrations in ice and precipitation estimates.

### 3.1.4   $SO_4^{2-}$

Surface in situ $SO_4^{2-}$ measurements in the major observing networks typically use ion chromatography methods, which have approximately 3% uncertainty range (Solomon et al., 2014). However, $SO_4^{2-}$ measurements have been shown to have up to 20% analytical uncertainty (AMAP, 2021a). $SO_4^{2-}$ datasets were obtained from IMPROVE, EMEP, and CABM networks, often via the EBAS database.

    $SO_4^{2-}$ deposition was also derived from the same ice core measurements mentioned above for BC deposition (D4, ACT2: McConnell and Edwards (2008); Humboldt: Bauer et al. (2013), Summit: Maselli et al. (2017); NGT_B19, ACT11d: McConnell et al. (2019)). The Arctic $SO_4^{2-}$ measurement locations are shown in Figure 1.





### 3.1.5 PM$_{2.5}$

Surface in situ PM$_{2.5}$ measurements are usually made via gravimetric analysis of particulate matter collected on a filter (e.g., Teflon substrate), which has around 1-6% uncertainty range (Malm et al., 2011). These data were obtained from Beijing Air Quality and U.S. Embassy data for China, NAPS for Canada (Dabek-Zlotorzynska et al., 2011), IMRPROVE for the U.S., and EMEP/EBAS for Europe.

### 3.2 Satellite datasets

Satellite observations are useful for evaluating models on larger horizontal spatial scales and for evaluating the 3-dimensional atmosphere - not the surface concentrations. Observations from three satellite instruments were used to evaluate model trace gas distributions in the free troposphere and when appropriate, the lower-stratosphere. These were: the Tropospheric Emission Spectrometer (TES; (Gluck, 2004a,b)), the Atmospheric Chemistry Experiment-Fourier Transform Spectrometer (ACE-FTS; Bernath et al. (2005)), and the Measurements of Pollution in the Troposphere (MOPITT; (Ziskin, 2000)). The vertical profiles

of trace gas volume mixing ratios are derived or retrieved from the satellite-measured emission or absorption spectra, with varying degrees of vertical sensitivity. These remote techniques typically have about a 15% uncertainty on the measurements (e.g., Verstraeten et al. (2013)) though this depends on the specific instrument and the species retrieved (e.g., Sheese et al. (2017)).

Note that while TES and MOPITT have global spatial coverage, their coverage does not extend up into the high Arctic. The

TES instrument on NASA's Aura satellite measures vertical profiles of trace gases such as O$_3$, CH$_4$, NO$_2$, CO, and HNO$_3^-$ from 2004-present. After interpolating all models and TES results to a 1° × 1° horizontal grid, the monthly mean CH$_4$ and O$_3$ from the TES lite products were matched in space and time with models. Models were smoothed with the TES monthly mean averaging kernels prior to comparisons with satellite data. TES measurements started in 2004 and stopped in late 2015, and had poorer coverage in its last few years.

A similar comparison method was used for MOPITT data. The MOPITT instrument on NASA's Terra satellite measures CO from 2000 to present.

The ACE-FTS instrument on CSA's SCISAT satellite has measured trace gases; O$_3$, CO, NO, NO$_2$, CH$_4$, among over 30 others from 2004-present. SCISAT has a high-inclination orbit giving its instruments better coverage in the Arctic. ACE-FTS is a limb-sounding instrument, measuring the solar absorption spectra of dozens of trace gas concentrations from the upper-

troposphere to the thermosphere. This gives us the opportunity to evaluate the 3-D model output in a region of the atmosphere where the radiative forcing of ozone is at its highest. Evaluating models with ACE-FTS measurements also sheds insight into models' transport and upper tropospheric chemistry. As was shown in Kolonjari et al. (2018), 3-hourly model output (rather than monthly mean output) is required for accurate comparisons to ACE-FTS data, thus, only models that provided output at this time frequency were compared to ACE-FTS measurements. The model output was sampled to match the times and

locations of ACE-FTS measurements. We used an updated version of the advanced method in Kolonjari et al. (2018). Instead of taking the model output at the closest time to the ACE-FTS measurement time, the model output was linearly interpolated


onto the ACE-FTS time. This reduces the bias introduced by diurnal cycles, which can cause certain volume mixing ratios (VMRs; e.g. that of NO and $NO_2$) to vary significantly in between model output times. As in Kolonjari et al. (2018), the model output is also interpolated vertically in log pressure space and bilinearly in latitude and longitude to account for spatial variation between model gridpoints.

### 3.3 Measurement campaigns

Finally, there were air- and ship-based measurement campaigns of black carbon that were used for model evaluation. Aircraft campaigns allow for vertical profiles of chemical species to be evaluated, and ship campaigns allow for in situ measurements in the remote Arctic seas.

### 3.3.1 Aircraft campaigns

The flight paths of the aircraft data used for model evaluation of BC are shown in Figure 2. The aircraft campaigns include A-FORCE (Oshima et al., 2012), ARCPAC (Brock et al., 2011), ARCTAS (Jacob et al., 2010), EUCAARI (Hamburger et al., 2011), HIPPO (Schwarz et al., 2010), NETCARE (Schulz et al., 2019), and PAMARCMIP (Stone et al., 2010). Most of these aircraft campaigns occurred during boreal spring and summer months (April to July) except for one (HIPPO) occurring in January and November, and most occured during the 2008-9 time period, with only one (NETCARE) occurring during 2014-15. All of these aircraft campaigns measured rBC from single-particle soot photometers (SP2) (Moteki and Kondo, 2010; Schwarz et al., 2006; Stephens et al., 2003).

The AMAP models that submitted 3-hourly BC output were linearly interpolated on to the aircraft locations in space and time using the Community Intercomparison Suite (CIS; Watson-Parris et al. (2016)) in order to provide representative comparisons and robust evaluation.

### 3.3.2 Ship campaigns

There were three ship-based measurement campaigns in 2014-2015. These were the two Japanese campaigns (MR14-05 and MR15-03 cruises of R/V Mirai) in September of 2014 and 2015 (track from Japan to north of Alaska; Taketani et al. (2016)), and the Russian campaign in October 2015 (track north of Russia, from Arkhangelsk to Severnaya Zemlya and back; Popovicheva et al. (2017)) - both are shown in Section 4.4 (Figure 18). Models that provided 3-hourly BC output were compared to these observations. The Russian measurements of aerosol eBC concentrations were determined continuously using an aethalometer purposely designed by MSU/CAO (Popovicheva et al., 2017). Light attenuation caused by the particles depositing on a quartz fiber filter is measured, the light attenuation coefficient of the collected aerosol was calculated. eBC concentrations were determined continuously by converting the time-resolved light attenuation to the eBC mass corresponding to the same attenuation and characterized by a specific mean mass attenuation coefficient, in calibration with AE33 (Magee Scientific).





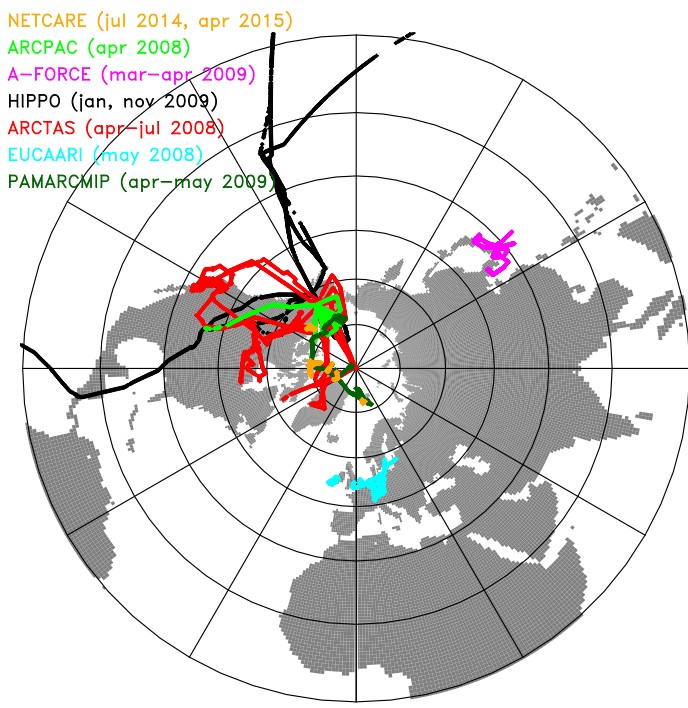

**Figure 2.** Flight tracks of BC aircraft campaigns used in this study.

The Japanese measurements provide rBC (refractory BC). Pileci et al. (2021) showed that rBC and eBC are linearly related, thus, in order to compare the observations to models, we converted rBC to eBC via a factor of 1.8 (eBC = 1.8×rBC; Zanatta et al. (2018); Pileci et al. (2021)).

## 4 Model-measurement Comparisons

In this section, we evaluate modelled SLCFs from the 18 participating models with a focus on performance in the Northern Hemisphere mid-latitudes (defined for our purposes as 30-60°N) and the Arctic (defined here as >60°N for simplicity). We look at spatial patterns in the model biases, as well as the vertical distribution and the seasonal cycles for each species.

### 4.1 Methane

Measured annual mean surface methane is shown in the top left panel of Figure 3, along with model biases in the rest of the panels. Recall, that unlike the rest of the SLCF species in this study, $CH_4$ concentrations were prescribed in these models from the same $CH_4$ dataset (Olivié et al., 2021). That said, the different decisions by modellers in how those $CH_4$ global





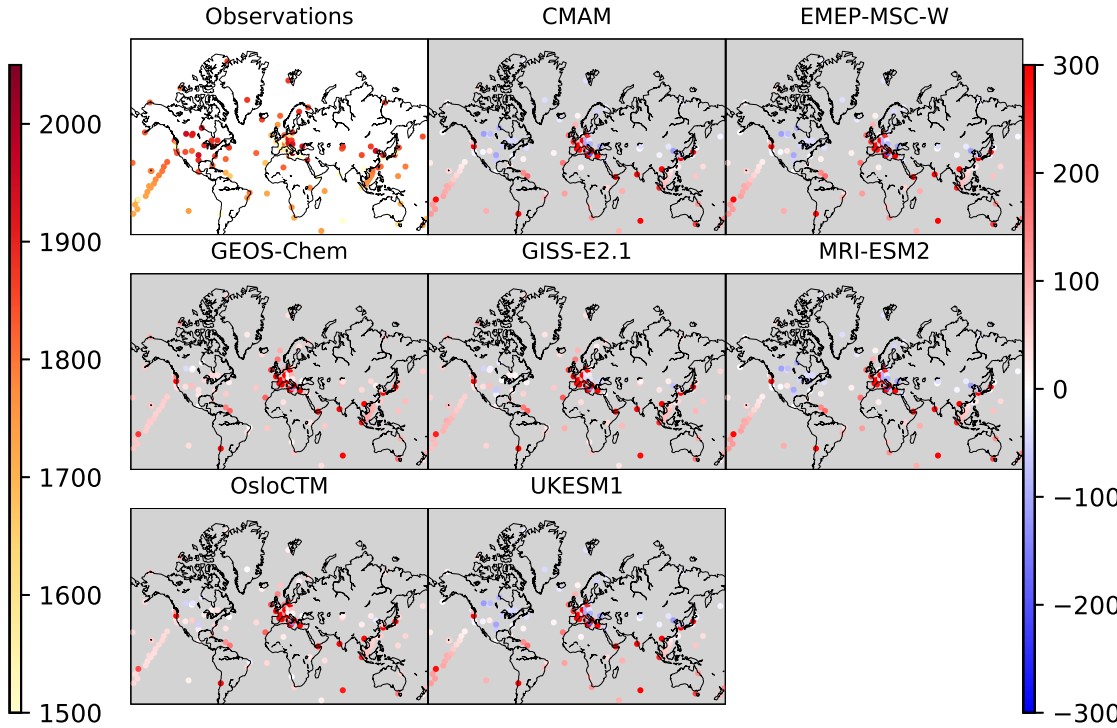

**Figure 3.** (top left) Measured surface-level methane (in ppbv, left colour bar) and (rest of panels) model biases (model minus measurement, also in ppbv, right colour bar) for 2014-2015.

concentrations are distributed make differences in how these models compare to measurements. The mean model biases are small and mainly positive; in the mid-latitudes, the multi-model mean bias is +145 ppbv (or +8.5%), and in the Arctic, the multi-model mean bias is 24 ppbv (or 1.3%), which means that the models simulate the magnitude of surface $CH_4$ well - though still outside the <1% measurement uncertainty range. There is a gradient in $CH_4$ concentrations (higher in the northern hemisphere and lower in the southern hemisphere) that is seen in the measurements (Figure 3,top-left) and reported in the literature (e.g., Dlugokencky et al. (1994)), though not well captured by CMAM, MRI-ESM2, and UKESM1 models, which are all biased low in the northern hemisphere, and biased higher towards the south. That is because of the simplifications made in these models distributions of $CH_4$. For example, CMAM used a single global-average $CH_4$ concentration that is interpolated linearly in time from once-yearly values.

Figure 3 also shows that observed annual mean surface $CH_4$ ranges geographically from about 1500 to 2100 ppbv depending on location, however, the models have a much smaller range due to their prescribing $CH_4$ concentrations as a lower boundary input. For example CMAM $CH_4$ volume mixing ratios only span about ±3 ppbv around 1836 ppbv. The span of MRI-ESM2





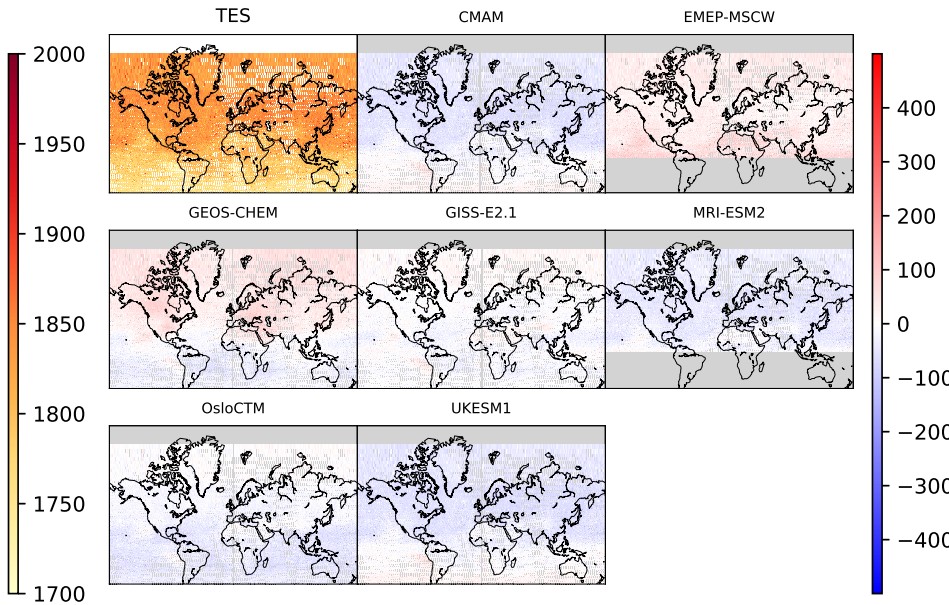

**Figure 4.** (top left) TES measurements of $CH_4$ in the mid-troposphere (at 600 hPa, in ppbv, left colour bar) and (rest of panels) model biases for 2008-9 (model minus measurement, also in ppbv, right colour bar). Results for 2014-15 are similar but had less spatial coverage by the satellite.

surface $CH_4$ is even smaller. GEOS-Chem, GISS-E2.1, and OsloCTM have a more realistic range of 1700-2000 ppbv, though

they still do not get the full variability that is seen in surface $CH_4$ mixing ratios close to $CH_4$ sources. However, in the free-troposphere (above the boundary layer), we have TES satellite measurements of $CH_4$ that show that $CH_4$ is much more smoothly distributed aloft. Thus, the simplification of prescribing $CH_4$ concentrations in the models, is more realistic there (Figure 4, showing the 600 hPa level in the mid-troposphere). Additionally, Figure 4 better illustrates the latitudinal gradient in $CH_4$ over the globe, and its lack in some models, which have more negative biases in the Northern Hemisphere, and more

positive biases in the southern hemisphere. Other models, such as GISS-E2.1, do a good job of capturing the global distribution of $CH_4$.

      In the Arctic, the vertical cross section of $CH_4$ VMRs over time as measured by the ACE-FTS in the mid-to-upper troposphere, and in the stratosphere is shown in Figure 5. There is a large decrease in $CH_4$ above the tropopause. The models are all biased high in the troposphere, and low in the lower-stratosphere, then high again in the upper-stratosphere. This pattern is

true for mid-latitudes as well as in the Arctic, and implies that the altitude of the modelled tropopause is too low. This same conclusion was also found in Law (2021) via comparisons of these models simulations to ozonesonde measurements, and



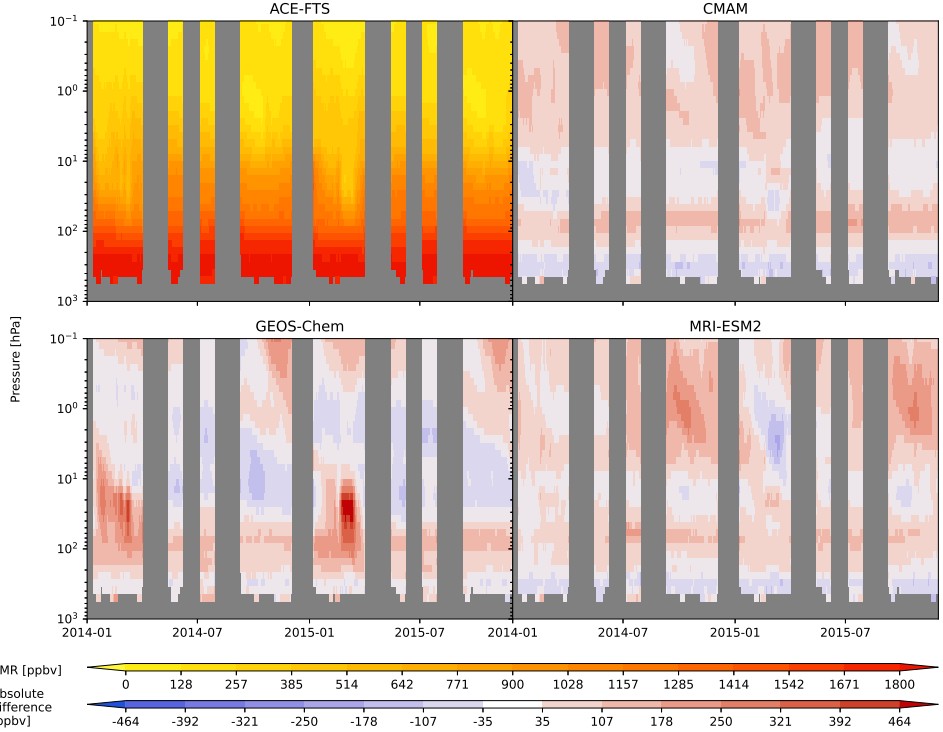

**Figure 5.** (top left) ACE-FTS measurements of Arctic (>60°N) $CH_4$ vs month (in ppbv, top colour bar) and (rest of panels) model biases for 2014-15 (model minus measurement, also in ppbv, lower colour bar). Results for 2008-9 are similar and not shown.

in our ACE-FTS $O_3$ comparison in the next section. The $CH_4$ model-measurement correlation coefficients for ACE-FTS are relatively high (e.g., R=0.48 to 0.86 depending on the model).

Therefore, the general model evaluation for $CH_4$ indicates that because models do not explicitly model $CH_4$ emissions, they 300 do not simulate the surface-level variability of $CH_4$ VMRs. Models differ in their global distribution of $CH_4$, thus only some contain the north-south $CH_4$ gradient. Those that do not have the largest underestimations of Arctic tropospheric $CH_4$. The $CH_4$ evaluation also implies that modelled tropopause height may be too low.

## 4.2 Ozone

Tropospheric $O_3$ is the third most important greenhouse gas (IPCC, 2021), and a regional pollutant that causes damage to 305 human health and ecosystems. $O_3$ is a secondary pollutant, formed in the troposphere via photochemical oxidation of volatile organic compounds in the presence of nitrogen oxides ($NO_x=NO+NO_2$). As such, models must simultaneously simulate the meteorological conditions, precursor species distributions, and the photochemistry correctly in order to accurately simulate $O_3$. That said, since surface $O_3$ is an important contributor to poor air quality, there is significant pressure for models to simulate





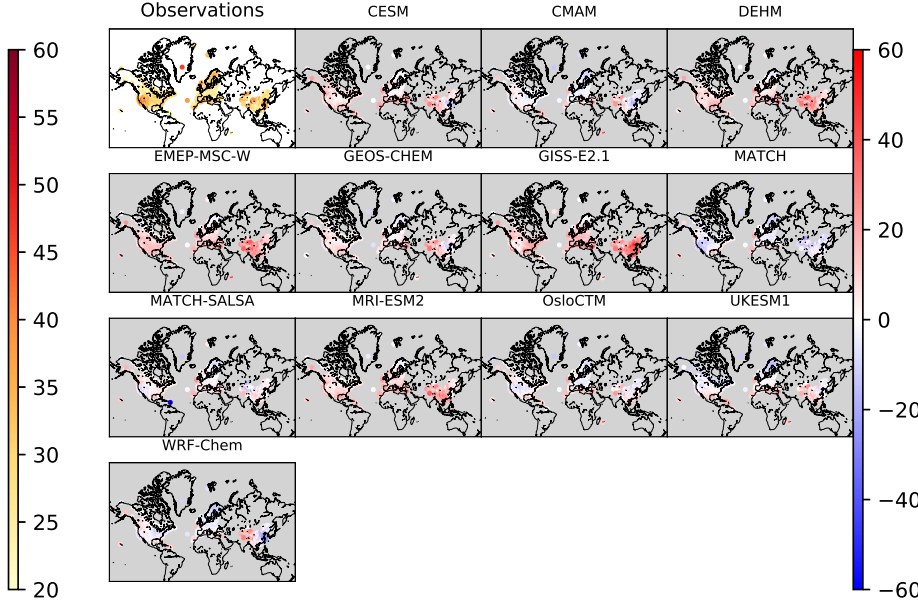

**Figure 6.** (top left) Annual mean in situ surface $O_3$ measurements (in ppbv, left colour bar) and (rest of panels) model biases for 2014-15 (model minus measurement, also in ppbv, right colour bar). Results for 2008-9 are similar but are not available for China.

it accurately, particularly in the heavily populated mid-latitudes (e.g., for air quality forecasting). Only models with prognostic
$O_3$ are included in this section.

Figure 6 shows the in situ annual mean $O_3$ measurements (top left panel), and the model biases (rest of panels). These include averaging $O_3$ from hourly observations (day and night) and 3-hourly or monthly modelled $O_3$ depending on which were available for each model. Surface $O_3$ is over-predicted by most models, which has been documented previously (Makar et al., 2017; Turnock et al., 2020). It has been shown that models can have problems producing low $O_3$ overnight (Brown et al.,
2006; Lin et al., 2008). In the Arctic, simulated surface $O_3$ has more mixed results. Annual mean concentrations are of the order of 40 ppbv, and individual model biases range from -20% to +52% globally on average for 2014-15. The multi-model mean has a bias of +11% for the Arctic, but this is not spatially distributed uniformly. All models overestimated surface $O_3$ in Alaska (mainly due to the overestimation of summertime concentrations, discussed below), and most models have too little $O_3$ at the Greenland location and in northern Europe.
The models were all able to represent the summertime peak in the mid-latitudes seasonal cycle (not shown). In contrast to the more polluted mid-latitudes, where surface $O_3$ peaks in the summertime due to photochemical production being at a maximum, Arctic $O_3$ is more influenced by the Brewer-Dobson circulation, bringing a maximum of tropospheric $O_3$ in the springtime due to photochemical production (Wespes et al., 2012), descent from the stratosphere, and more long-range transport of $O_3$ to the





Arctic. Figure 7, shows this springtime peak in both the western (a, longitude<0°) and eastern (b, longitude>0°) Arctic in the

measurements. However, the models only capture that seasonal cycle in the eastern Arctic [Fig. 7(b)], implying that that the models are representing large scale circulation and possibly stratosphere to troposphere exchange well. But it is interesting to note that the models that have sophisticated representation of stratosphere-troposphere exchange (such as CMAM, MRI-ESM2, UKESM1) do not particularly stand out as better performers in Figure 7, compared to models that do not simulate the stratosphere (such as DEHM, MATCH, MATCH-SALSA). Thus, it's impact on surface $O_3$ may be very small.

In the western Arctic [Alaska mainly, Figure 7(a)], models overestimate summertime Arctic $O_3$, likely due to over-predicting the impact of wildfire emissions on tropospheric $O_3$ concentrations, which is a research topic with high uncertainty (van der Werf et al., 2010; Monks et al., 2015; Arnold et al., 2015). Another possibility is that modellled $O_3$ dry deposition over boreal vegetation is underestimated (Stjernberg et al., 2012; Thorp et al., 2021).

Some Arctic locations are more inclined to get spring-time surface $O_3$ depletion due to bromine explosions and halogen

chemistry (Bottenheim et al., 1986; Barrie et al., 1988; Simpson et al., 2007). None of the model simulations in this study contain the necessary tropospheric halogen chemistry to simulate those events, thus, this partly explains why some models in Figure 7(bottom) overestimate springtime $O_3$ concentrations. That particular feature is explored further on a site-to-site basis in Law (2021).

The next subsection shows that both $O_3$ precursors CO and $NO_2$ are underestimated compared to measurements at all global

locations. This has implications for simulated tropospheric $O_3$ chemistry.

*Free-tropospheric $O_3$: satellite comparisons*

Aircraft-based measurements and ozonesondes can provide insight into the vertical distribution of $O_3$ and these have been well documented [E.g., Tarasick et al. (2019); Law (2021)]. However, model grid boxes may not be representative of those fine spatial scale measurements. In this study, we examine how the model biases change in the vertical when compared to satellite

measurements, which have a larger, "smoothed out" spatial sensitivity due to their viewing geometry and retrieval methods. Specifically, we compare modelled $O_3$ to TES and ACE-FTS satellite-based retrievals. These satellite instruments also have better global coverage than aircraft and sonde-based measurements.

The model fractional biases as compared to TES measurements from near surface up to 100 hPa are shown in Figure 8 for the Arctic (left), and mid-latitudes (right). All models' simulated fractional biases have similar vertical profiles for both the

Arctic and mid-latitudes, with greater negative values at lower levels, and a more positive "bulge" of about 10% around 300 hPa in the Arctic and about 5% around 200 hPa at mid-latitudes. That "bulge" in model biases at 300 hPa was also seen to a greater degree (50-70%) when comparing these models simulations to Arctic ozonesonde measurements in Law (2021). As compared to TES, which has much lower vertical resolution, the results are not as striking. On an average, the models have a negative bias at all vertical levels in the Arctic region and in lower troposphere in the mid latitude region, where as a positive

bias is seen in the upper troposphere below 60°N. This is consistent for the two time periods (2008-9 and 2014-15).

Given that Figure 6 showed positive biases near the mid-latitudes, while Figure 8 shows lower $O_3$ in the free troposphere, these results imply that there is not enough vertical lifting/mixing of tropospheric $O_3$ in most of the models. However, the TES measurements have been shown to be biased high by approximately 13% throughout the troposphere (Verstraeten et al.,





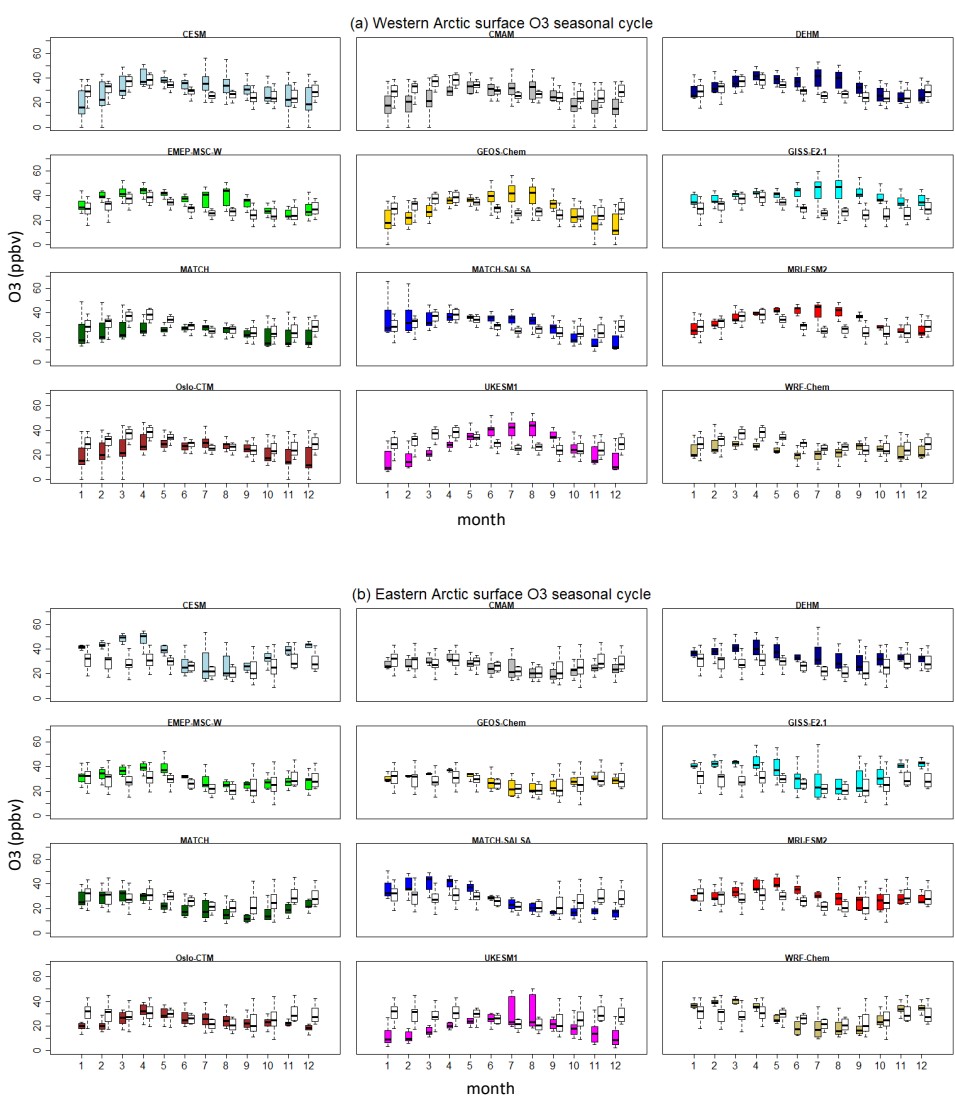

**Figure 7.** Surface $O_3$ monthly range that occurs at the locations in Fig. 6 above $60°$N. The measurements are the black and white box/whiskers, and the models are the coloured box and whiskers. (a) for the western Arctic and (b) for the eastern Arctic for 2014-15. Thick horizontal lines indicate the median $O_3$ VMR in each month, and the box extends to the interquartile range. The whiskers extend to the minimum and maximum monthly mean $O_3$ VMR.



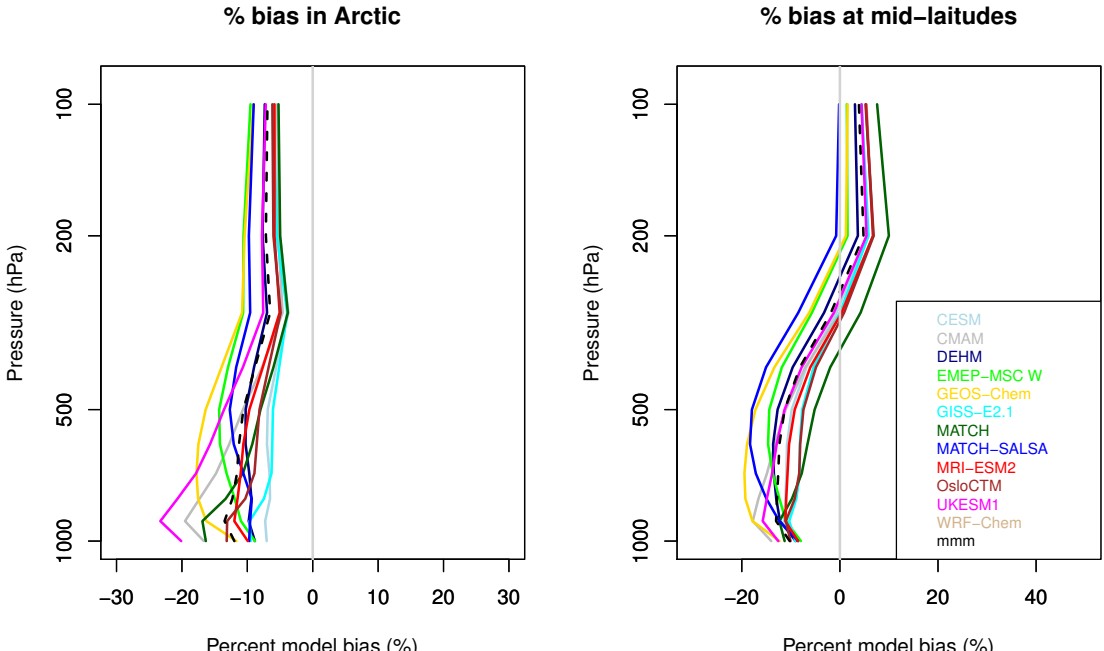

**Figure 8.** Vertical distribution of models' $O_3$ percent biases [(model minus measurement) over measurement] for 2008-9 as compared to the TES measurements; (left) average for Arctic latitudes (>60°N) and (right) average for mid-latitudes (30-60°N). mmm = the multi-model mean.

2013), which is the same amount that the multi-model mean is low. Similarly, ACE-FTS $O_3$ has an uncertainty range between +5-10% when compared to $O_3$ from other satellite limb-view observations (Sheese et al., 2017). Thus, overall, participating models simulate the free tropospheric $O_3$ reasonably well and within the uncertainly limits of the observations.

Therefore, the general model evaluation for $O_3$ indicates that all models overestimate surface $O_3$ at mid-latitudes, and that, combined with a lack of $O_3$ transport to the Arctic, result in modelled Arctic $O_3$ VMRs having relatively little bias (the right answer for the wrong reason). The summertime evaluation implies that models overestimate the $O_3$ produced and transported by wildfires in the western Arctic. The $O_3$ evaluation also implies that modelled tropopause height may be too low.

### 4.3 $O_3$ precursors: Carbon monoxide and nitrogen oxides

Figures 9 and 10 show the comparisons of the models to the surface in situ measurements. The multi-model annual mean underpredicts both CO and $NO_2$ by approximately -55% in the Northern Hemisphere for 2014-15. The 2015 AMAP report showed similar findings for simulated surface CO, as have other studies (AMAP, 2015a; Emmons et al., 2015; Monks et al., 2015; Jiang et al., 2015; Quennehen et al., 2016), pointing to a possible underestimation of CO emissions, and possibly shorter modelled lifetimes of CO due to an overestimation in OH (Miyazaki et al., 2012). The annual mean surface CO underestimation is mainly dominated by the wintertime (e.g., the multi-model mean bias in DJF is -92%), when it has been reported that CO



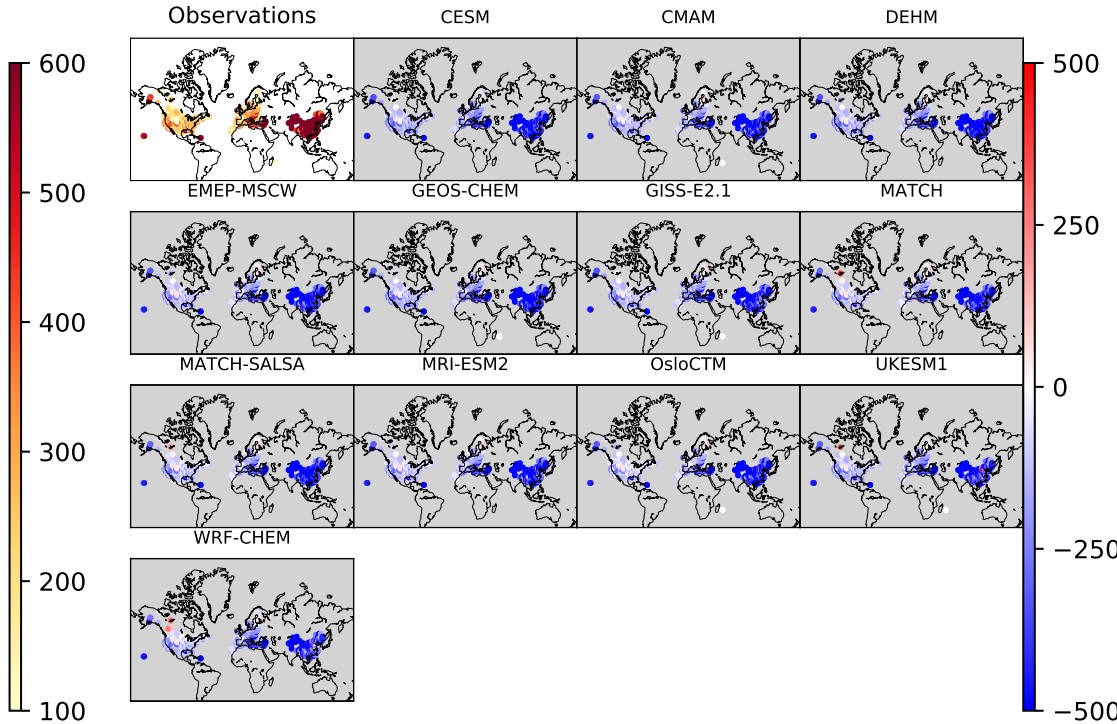

**Figure 9.** Mean CO volume mixing ratios (ppbv, left colour bar) at surface measurement sites, and model bias (model minus measurement in ppbv, right colour bar) for 2014-15. Results from 2008-9 are similar and not shown.

emissions from combustion are too low (e.g., Kasibhatla et al. (2002); Pétron et al. (2002)). All the models exhibit a large negative bias over China which is consistent with the study by Quennehen et al. (2016), and is attributed to the enhanced

destruction of CO by OH radicals, but it was also found in Kasibhatla et al. (2002) and Pétron et al. (2002) that bottom-up CO emission inventories in Asia are greatly underestimated.

In the free troposphere, we compare modelled CO to that measured by MOPITT. Figure 11 shows these comparisons at the 600 hPa level. All models are biased low over land in the mid-latitudes, which was the same finding as in the surface comparisons. However, models were biased high over the oceans at lower latitudes. The biases were more negative at the 600

hPa level than at the 900 hPa level, but with similar spatial patterns. Monks et al. (2015) discussed that models had high biases in the outflow from Asia, and low biases north of there due to lack of transport. Our results are consistent with these findings. The Quennehen et al. (2016) study also suggested that summertime CO transport out of Asia is too zonal. This could explain some of the underestimation in the Arctic in the mid-troposphere.





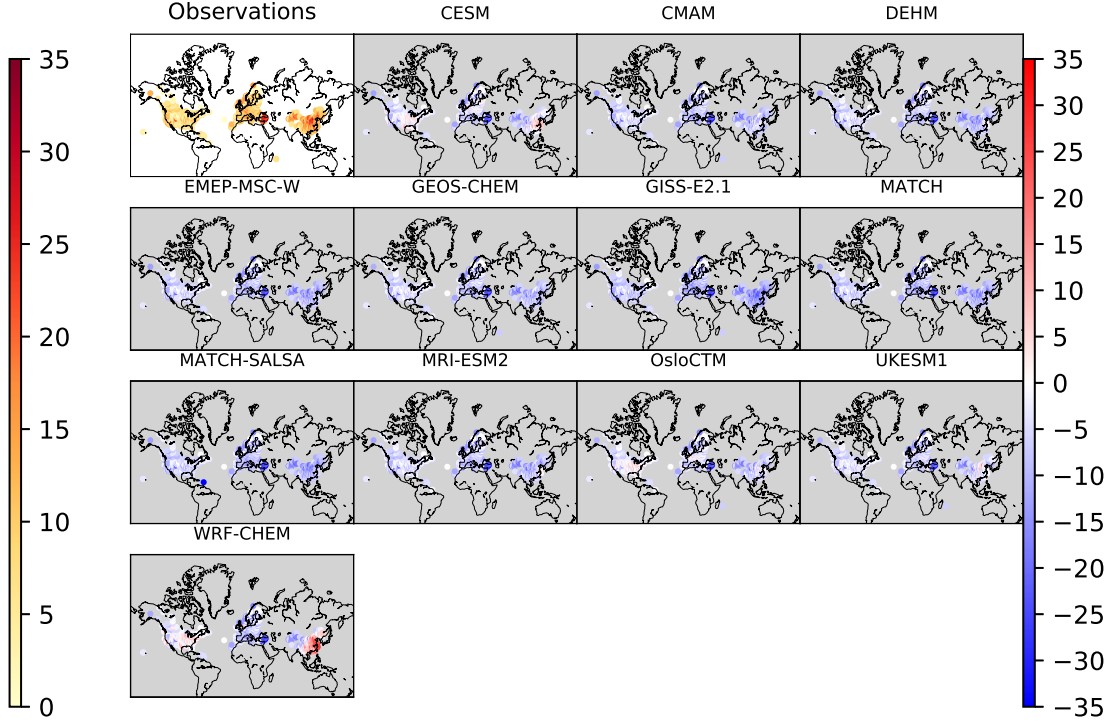

**Figure 10.** Mean $NO_2$ volume mixing ratios (ppbv, left colour bar) at surface measurement sites, and model bias (model minus measurement in ppbv, right colour bar) for 2014-15. Results from 2008-9 are similar and not shown.

In the upper-troposphere and stratosphere, modelled CO and $NO_x$ monthly time series are compared to measurements from

the ACE-FTS satellite instrument (where $NO_x$=NO + $NO_2$ which are measured separately), and those results are shown as Taylor diagrams in Figure 12, along with $O_3$ and $CH_4$ at 150 hPa, which is in the upper-troposphere, lower-stratosphere (UTLS) region. The contours show the model's overall skill as defined in Hegglin et al. (2010). Only the models that simulate the stratosphere were included, and the results show that there is a range in model performance by SLCF species, with no one model performing best for all. Comparison statistics for CO were poorer than those for $O_3$, $CH_4$ and $NO_x$.

**4.4   Black carbon**

In this section, we examine the spatial and seasonal distributions of BC using ground-based measurements which are primarily available in North America, Europe, and several locations in the Arctic, but are also available from two ship-based campaigns, and several aircraft campaigns. Given the limited global data available for both BC and $SO_4^{2-}$ (e.g., we could find none freely available for Asia), we focus the plots on the Arctic region here, and given that the magnitude of BC and $SO_4^{2-}$ does not span





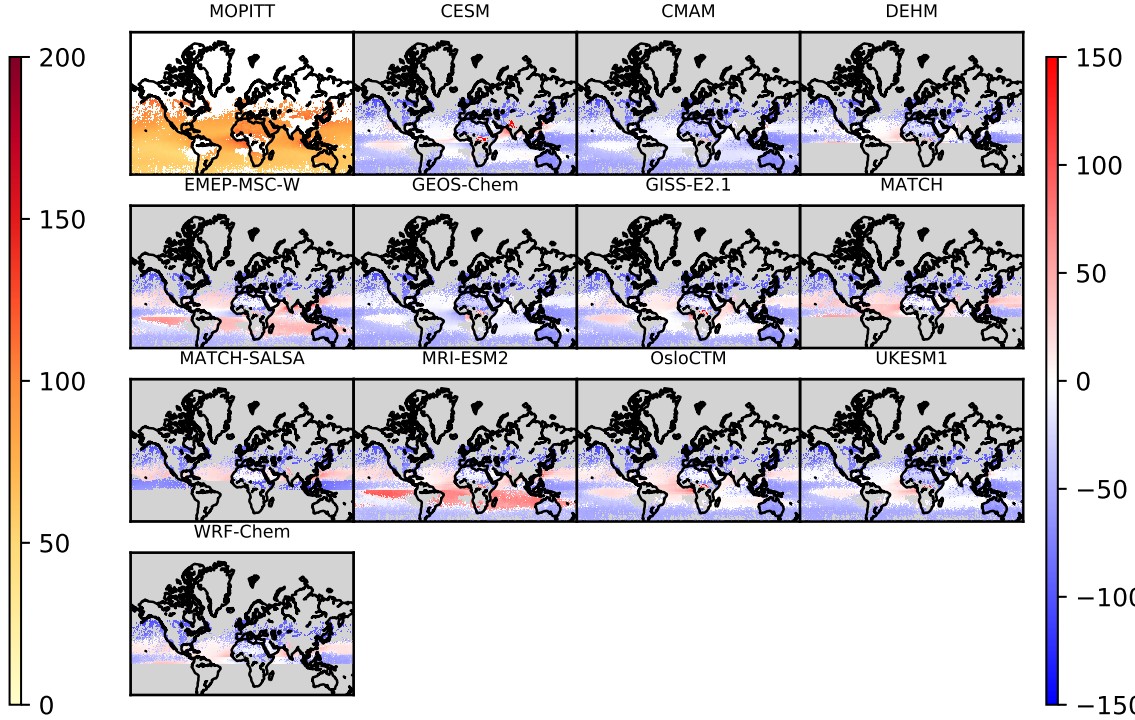

**Figure 11.** (top left) Mean MOPITT CO at 600 hPa (in ppbv, left colour bar) and (rest of panels) model biases (model minus measurement in ppbv, right colour bar) for 2014-15.

a wide range throughout the Arctic, we show model biases as percent rather than absolute differences as was done in previous sections for trace gas species shown globally. We also analyse the BC model-measurement comparisons keeping in mind that because there are various definitions and measurements types for BC, we consider an agreement within a factor of 2 to be within the uncertainty range (Section 3.1.3).

Figure 13 (top left panel) shows annual mean surface-level concentrations of black carbon (BC) at nine Arctic observation stations, and (rest of panels) the model percent biases there. The annual mean BC concentrations are of the order of less than 1 $\mu$g m$^{-3}$ and most models tend to underestimate BC in the high Arctic while overestimating in Alaska and Scandinavia. This result could be partially explained by the discrepancy caused by the use of EC and eBC data which are not the same (Section 3.1.3). As aerosols age during the transport to the high Arctic locations, their eBC (based on absorption converted to mass) gets more and more of a positive bias compared to EC. As models are more representative of EC, they will not be able to agree with eBC measurements in the aged air at high Arctic remote stations, such as Gruvebadet, Zeppelin, Alert, and Utqiagvik.





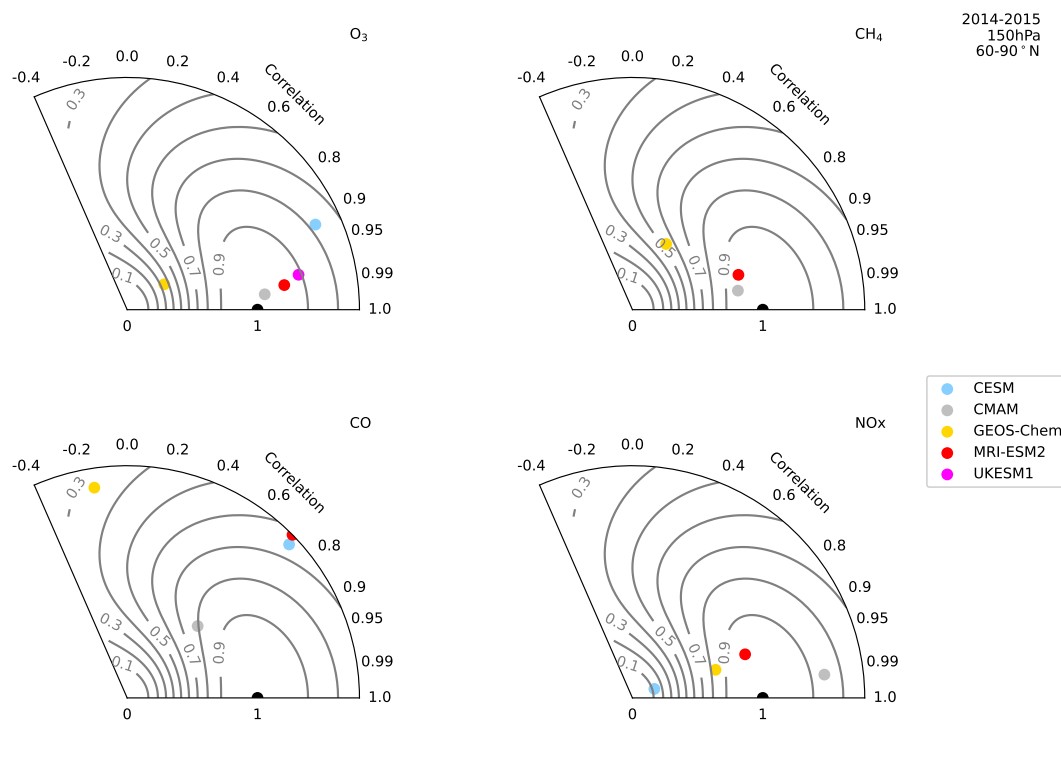

**Figure 12.** Taylor diagrams showing model performance for 2014-15 monthly average trace gases in the Arctic UTLS region at 150 hPa as evaluated against ACE-FTS satellite measurements. The grey contours indicate the skill as defined in Hegglin et al. (2010).

This is in contrast to the Alaskan and European stations, which closer to sources where BC is more fresh, and thus the eBC measurements have lower uncertainty.

That said, a few models (CanAM5-PAM, DEHM, and FLEXPART) overestimate BC concentrations in the high Arctic. Overall individual model biases range between ±100% at individual sites.

The underestimation of high-Arctic atmospheric BC concentrations may be related to excessive BC deposition further south, however, there are very few BC deposition measurements. In the Arctic, we can evaluate total (wet + dry) modelled deposition via derived ice core measurements. There were 6 ice cores on Greenland and one in the European Arctic, in Spitsbergen (Lomosovfonna). Figure 14 shows that all models overestimate BC deposition fluxes at the ice core locations. While the ice cores contain BC data starting in the year 1750, only data after 1990 have been used to match the modelled time period

(1990-2015, 1995-2015, 2008-9 and 2014-15, depending on the model).

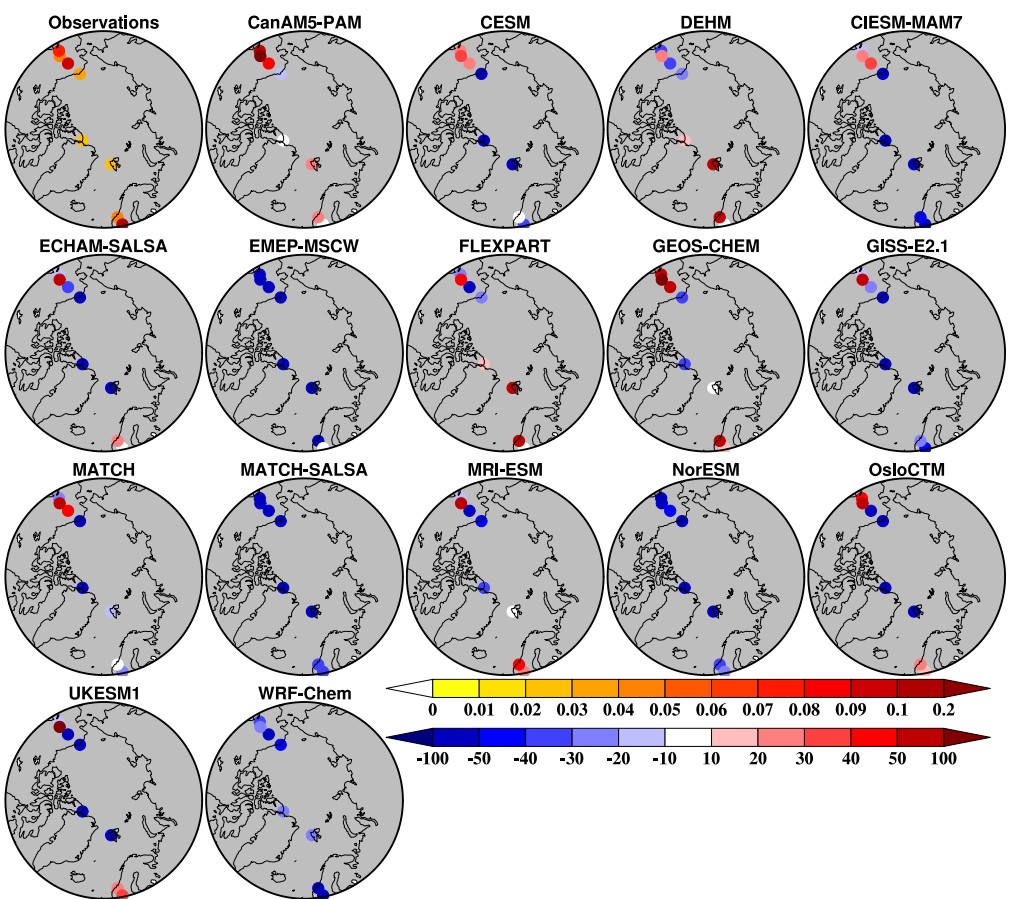

**Figure 13.** Mean BC concentrations (in $\mu$g m$^{-3}$, top colour bar) at surface Arctic measurement sites, and model bias [as (model-measurement)/measurement in %, bottom colour bar] for 2014-15. Results from 2008-9 are similar and not shown.

The measured BC deposition flux values on Greenland vary with elevation (lower fluxes at higher elevation). Summit (3177 masl) has an average of 285 $\mu$g/m$^2$/yr in contrast to ACT2 (2461 masl) with 676 $\mu$g/m$^2$/yr. BC deposition is highest in the European Arctic at Spitsbergen with 856 $\mu$g/m$^2$/yr. For all 7 ice cores used in this comparison the averaged model mean is 3 times as high as the observations. At D4 (2728 masl) the modeled mean corresponds best to the observation, with a mean bias of +83%. At ACT11 (2296 masl) the models have 4 times the deposition flux compared to the measurements. Generally though, the model mean is skewed higher by FLEXPART and DEHM (Figure 14), which also had higher atmospheric BC concentrations. A few models simulated less BC deposition than observed at these sites, and these models also underestimated BC atmospheric concentrations. Thus, it is difficult to conclude deposition biases as being a cause for atmospheric biases, when the two are inter-related parameters.





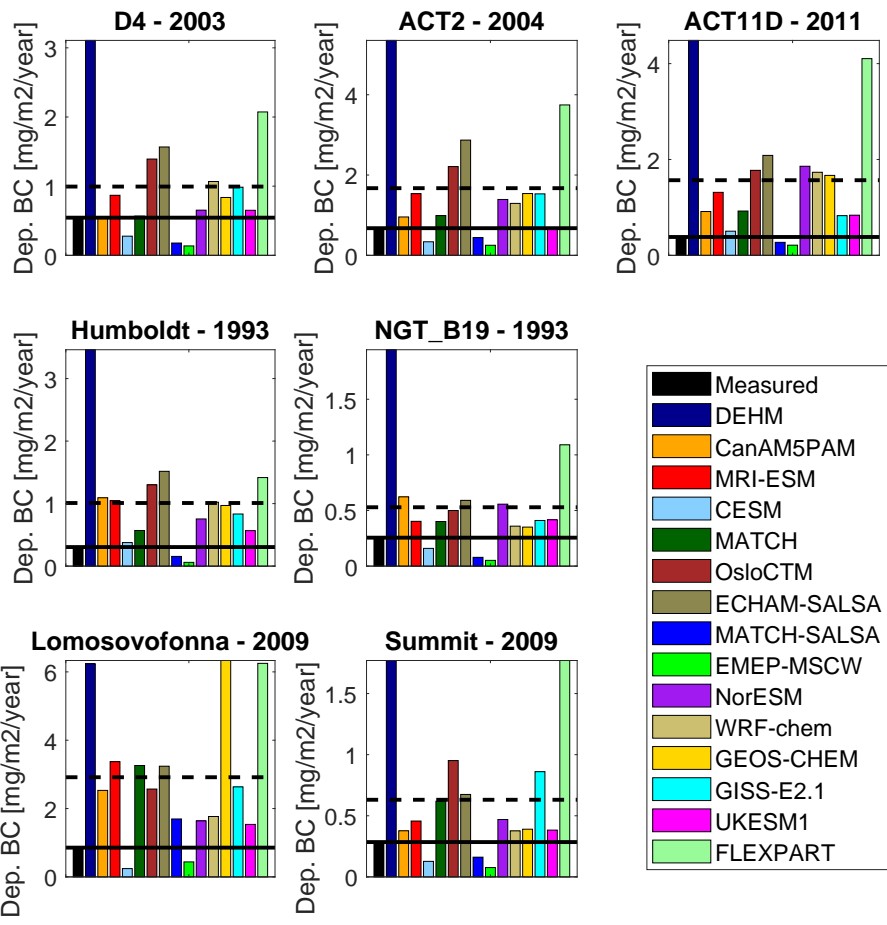

**Figure 14.** Annual average BC deposition flux values for the 7 ice core locations (Fig. 1) for each model based on values from 2008-9 and 2014-15. The observed fluxes are plotted in black and a black line indicates the level of the average observed flux, the black dashed line is the model mean for each location. The period used for plotting is based on all available years after 1990, the title indicates the last available year form the ice core record.



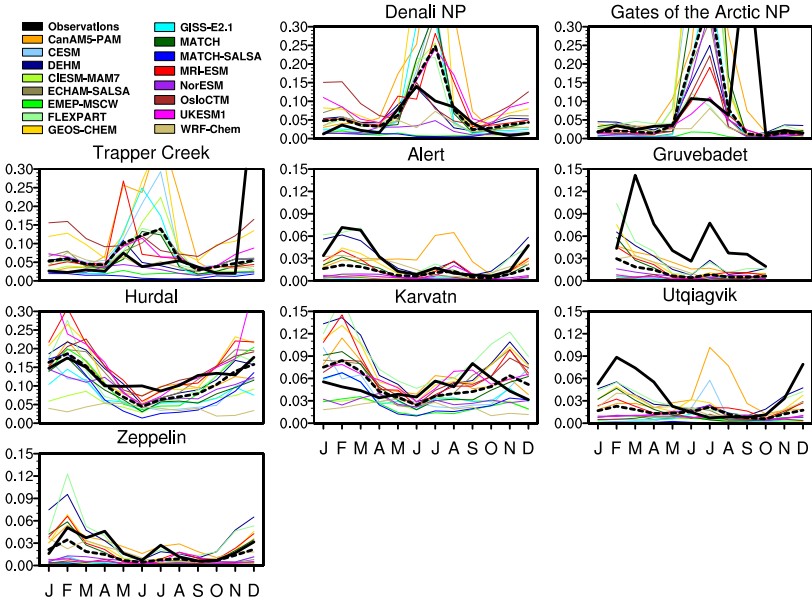

**Figure 15.** Modelled (thin colored lines) and measured (thick black line) monthly mean BC concentrations (in $\mu$g m$^{-3}$) at surface Arctic measurement sites in 2014-15. Multi-model mean is shown by the black dashed line.

The seasonal cycles of surface-level atmospheric BC concentrations at several Arctic locations are shown in Figure 15. As seen in Eckhardt et al. (2015), some models still underestimate wintertime BC, but many models now show similar seasonality as the observations. The multi-model mean also captures well the monthly variations including the summertime peak at some Alaskan sites caused by fire emissions. The multi-model mean Arctic BC is underestimated in winter (-24%), and overestimated in the summer (+32%), though overall, this is an improvement in model performance in simulating Arctic BC since the 2015 AMAP assessment on black carbon and ozone as climate forcers - the latter of which had -59% winter bias and +88% summertime bias (AMAP, 2015a; Eckhardt et al., 2015). However, it is difficult to make direct comparisons to that report as those values were for a smaller set of Arctic locations, different observation periods, and with a different set of models (though many overlapping). The model improvement may be due to the improved anthropogenic emissions of BC, particularly from northern Russia, where flaring emission factors were increased in ECLIPSEv6B compared to those used for the 2015 AMAP report.

Most models have reasonable spatial correlation with the measurements across the Arctic, in that they correctly simulate the range of BC concentrations that appear across the Arctic (e.g., higher concentrations at Hurdal, lower concentrations at Zeppelin, etc), as shown in Figure 16. However, there are still large differences and low R values in the statistics shown in Figure 16.

There are positive model biases at mid-latitudes (in North America and Europe; not shown) for surface-level BC. The vertical analysis of BC from the aircraft campaigns (below) provides further insight and support for the suggestion that models do not



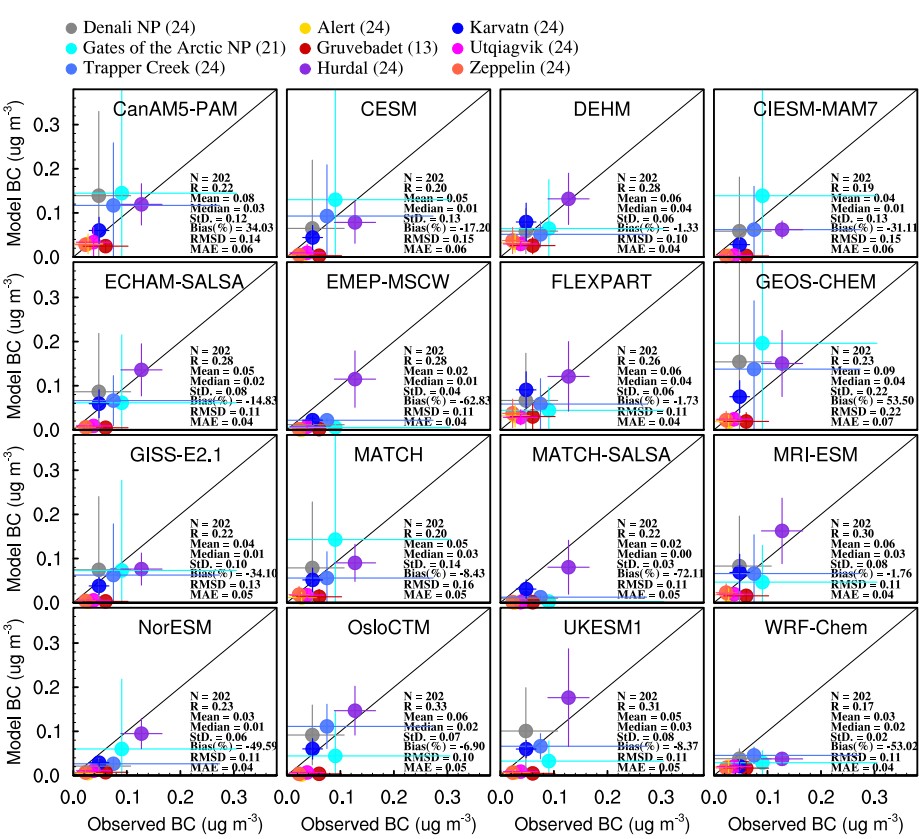

**Figure 16.** Modelled vs measured BC concentrations at surface Arctic measurement sites in 2014-15. Results for 2008-9 (not shown) had lower correlation coefficients and higher biases. Filled circles represent the mean for each location and the lines represent ± one standard deviation from mean. The number of monthly mean values available from individual sites is shown in brackets next to site names in the legend, with a max of 24 months in the 2 years.





have adequate long-range transport of the pollutants from their the sources in the mid-latitude, and thus, do not simulate enough pollution in the Arctic.

*Vertical profiles of BC: aircraft campaigns*

Gridded BC output at three-hourly intervals was provided by 11 of the participating models and was compared to aircraft campaign measurements of BC. The interpolation of model output to flight track coordinates was carried out by tools from the Community Intercomparison Suite (CIS; Watson-Parris et al. (2016)), which co-located the extracted model tracks with their corresponding observational values.

Figure 17 shows the median vertical profiles of BC concentrations from the aircraft measurements and from the models. At
mid-latitudes, from 0-2km, all of the models agree well with the measurements. However, BC concentrations decline steeply in a few models (e.g., MATCH-SALSA, EMEP-MSC-W, and GEOS-Chem) above 2 km. It would appear that they do not have enough vertical lifting of BC and/or perhaps too short of a BC lifetime. Indeed, one of these is EMEP MSC-W, in which the short BC lifetime was previously reported in Gliß et al. (2021). That said, in Lund et al. (2018b), the Oslo-CTM and ECHAM models were shown to *over*estimate the BC lifetime. In our case, OsloCTM isn't shown in Figure 17 because it didn't provide
BC at 3-hourly time scales. But ECHAM-SALSA results are consistent with the Lund et al. (2018b) study, in that they are particularly overestimating BC in the upper altitudes, in both the mid-latitudes and the Arctic, implying too long a lifetime, and too much long-range transport into the upper Arctic atmosphere. The measured BC profile at mid-latitudes drops off more quickly around the tropopause at 11-12 km, and, except for CanAM5-PAM, the models do not reproduce this drop.

In the Arctic profiles (Figure 17), the modelled and observed profiles do not decline with altitude throughout the troposphere,
but the observed median BC concentration does drop sharply around 9 km - again near the Arctic tropopause, and again, the only model to capture that change is CanAM5-PAM. In the Arctic comparisons, the models that didn't simulate enough BC aloft at the mid-latitudes, stand out as having larger underestimates of BC in the Arctic. For example, MATCH-SALSA and EMEP MSC-W have very low BC throughout the Arctic vertical profile. These results are consistent with the surface BC underestimation in Figure 15. Therefore, the underestimation seen in the Arctic for those two models is likely due to a lack of
long-range transport from the mid-latitudes, as well as errors in BC deposition mentioned above. In addition, the Zhao et al. (2021) study showed that different parts of the Arctic BC vertical profile are sensitive to BC transported from different areas of the world. For example, the lower tropospheric BC is influenced by emissions transported from North America, Russia-Belarus-Ukraine, Europe, and East Asia. Whereas, upper tropospheric Arctic BC is mainly influenced by transport from South Asia. Thus, the differences in the model results could be related to differences in how they simulate these transport pathways.

In Mahmood et al. (2016), they found that, overall, considerable differences in wet deposition efficiencies in the models exist and are a leading cause of differences in simulated BC burdens. Results from their model sensitivity experiments indicated that convective scavenging outside the Arctic reduces the mean altitude of BC residing in the Arctic, making it more susceptible to scavenging by stratiform (layer) clouds in the Arctic. Consequently, scavenging of BC in convective clouds outside the Arctic acts to substantially increase the overall efficiency of BC wet deposition in the Arctic, which leads to low BC burdens compared
to simulations without convective BC scavenging. Oshima et al. (2013) also found that convective scavenging at mid- and sub-tropical latitudes removes a significant fraction of BC. In contrast, BC concentrations in the upper troposphere are only weakly





**Figure 17.** Median vertical profiles of observed (heavy black line) and modelled (coloured lines) BC concentrations for all aircraft campaigns combined, separated into (left) mid-latitudes, and (right) Arctic. The multi-model median is shown by the dashed black line.



influenced by wet deposition in stratiform clouds, whereas lower tropospheric concentrations are highly sensitive (Mahmood et al., 2016) - these are consistent with the results we find in this study, where the multi-model median is too high above about 9 km and too low from 0-9 km. Indeed, the MATCH and MATCH-SALSA models, for example, assumes reduced scavenging of

aerosol in mixed phase clouds following Liu et al. (2011), which increases long-range transport to the Arctic. It is odd then that MATCH is one of the better performing models in Figure 17, and MATCH-SALSA is not. Despite the large range in modelled vertical BC concentrations, the multi-model median is close to the observed throughout the troposphere at both mid-latitudes and in the Arctic.

*Arctic seas analysis: ship campaigns*

From the ship-based measurements, we see that there is a consistent model overestimate of BC in the Pacific region where measured concentrations are very low (Figure 18). Indeed, Taketani et al. (2016) report that BC concentrations were in the range 0-66 ng m$^{-3}$, with an overall mean value of just $1.0 \pm 1.2$ ng m$^{-3}$. The models, possibly due to their coarse resolutions, were not able to simulate such low background BC concentrations. However, even the model with the highest resolution (GEM-MACH at 15 km resolution) overestimated BC in the Pacific - though that limited area model, in that region near

the boundary, would have been heavily influenced by the upwind, coarser resolution boundary conditions that were assumed ($1° \times 1°$ MOZART4 chemical boundary conditions). The high bias in the Pacific may be due to all models overestimating the amount of BC that gets transported off of the Asian continent. That model bias is consistent with our low-altitude comparisons of the models to the HIPPO aircraft campaign measurements, which were taken over the north-western Pacific (Figure 2). The BC overestimate over the Pacific was also found in the Schwarz et al. (2013) study looking at simulated BC from the AeroCom

global model intercomparison initiative as compared to HIPPO measurements.

Conversely, the modelled BC concentrations generally agree with measurements in the Russian Arctic ocean, though biased slightly low for the most part. Popovicheva et al. (2017) attributes the higher BC concentrations measured near the Kara Straight (north of 70°N) to gas-flaring emissions, and when near Arkhangelsk (White Sea), important sources were mid-latitude biomass burning, transportation, and combustion (residential and commercial). Since models were able to simulate

this well, their improvement is likely due to improved Russian anthropogenic emissions in ECLIPSE v6B (Section 2.1, AMAP (2021a)) compared to previous emissions datasets, which didn't include enough Russian flaring emissions. The best model results were from ECHAM-SALSA and MATCH when compared to all of the ship campaign data.

Therefore, the general model evaluation for BC indicates that while there is a large variability in models results, they tend to overestimate surface BC at mid-latitudes (including over the Pacific ocean) and underestimate surface BC in the Arctic. Again,

these results point to a lack of transport to the Arctic, and in this case, too much BC deposition along the way. Though we were only able to evaluate BC deposition in the Arctic in this study, those results support the hypothesis of some models having too much BC deposition. The BC vertical profile evaluation also implies that modelled tropopause height may be too low.





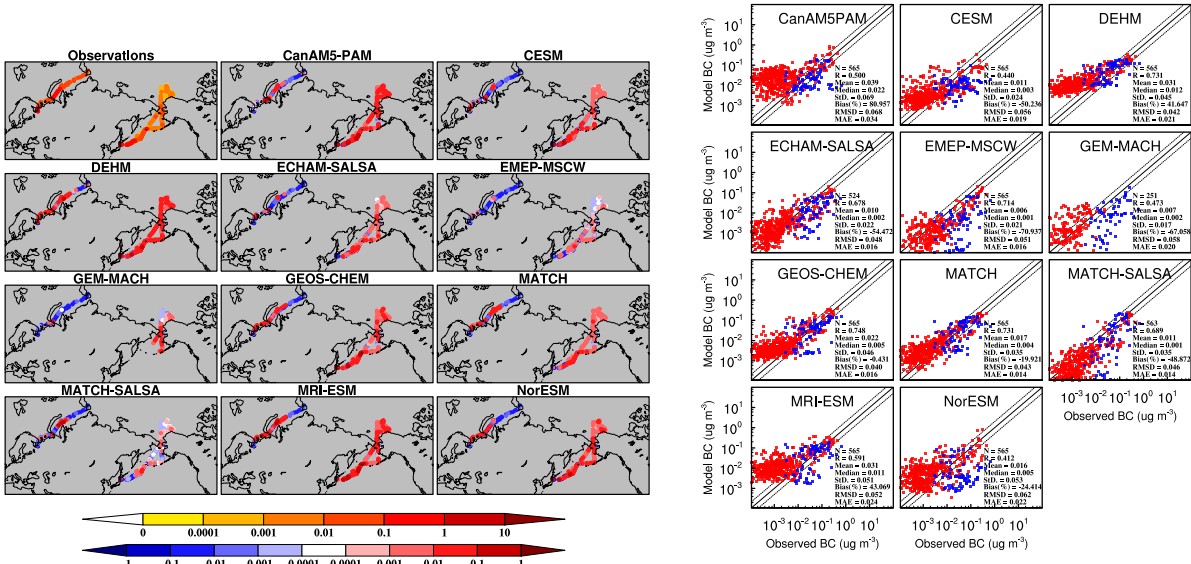

**Figure 18.** (left) Observed BC concentrations (in $\mu$g m$^{-3}$, top colour bar) along the ship paths, and the model biases ($\mu$g m$^{-3}$, bottom colour bar). (right) Modelled vs measured 3-hour-average BC concentrations along the ship paths. Note the logarithmic scale.

## 4.5  Sulfate

We used monthly mean surface level observations of SO$_4^{2-}$ from 18 Arctic sites to evaluate the models. Figure 19 shows
that, similar to BC, the SO$_4^{2-}$ concentrations in the high Arctic are underestimated by most of the models. A few models
overestimate SO$_4^{2-}$ in Scandinavia and Alaska.

The model underestimations of SO$_4^{2-}$ could be mainly due to higher efficiencies of models in removing aerosol during the
long-range transport to the high Arctic. This is consistent with a previous study based on AMAP 2015 model simulations that
found that the convective wet deposition outside the Arctic region may have led to different seasonal cycles of aerosol con-
centrations in the Arctic (Mahmood et al., 2016). Dimethylsulphide (DMS) a naturally occurring source of sulfur from marine
algae emissions could also be misrepresented in models. However, this source would be more pronounced in the summer when
there is less sea ice in the high Arctic, and it does not appear as though models are underestimating only in the summertime
(Figure 20). Rather, some models appear too low in the winter and spring - which point towards underestimating local Arctic
sources, and to a lack of transport from mid-latitudes as being the key issues. Despite the individual model differences in repre-
senting the seasonal cycle, the multi-model mean compares well with observations at most locations. However the multi-model
mean SO$_4^{2-}$ is significantly underestimated at Alert and Irafoss sites. Mean model biases for all Arctic sites ranges from -65%
to +80% among different models, and correlation coefficients are typically around 0.5 (Fig. 21).

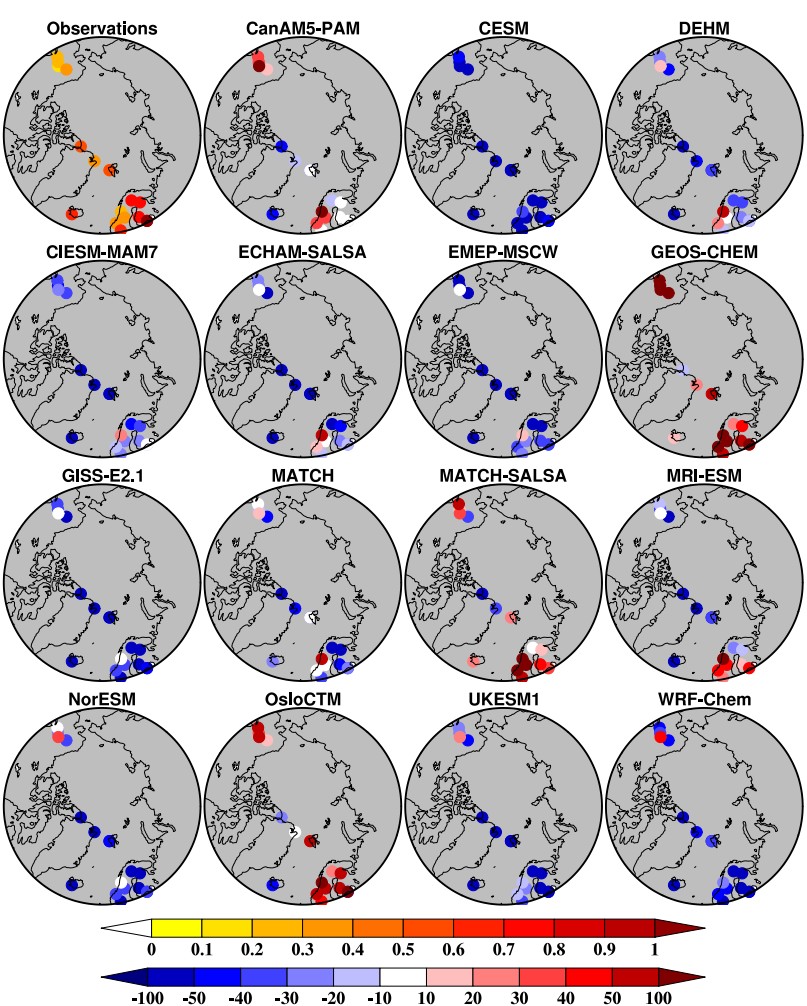

**Figure 19.** Mean measured $SO_4^{2-}$ concentrations (in $\mu g\ m^{-3}$, top colour bar) at surface Arctic measurement sites, and model bias [as (model-measurement)/measurement in %, bottom colour bar] for 2014-15. Results from 2008-9 are similar and not shown.



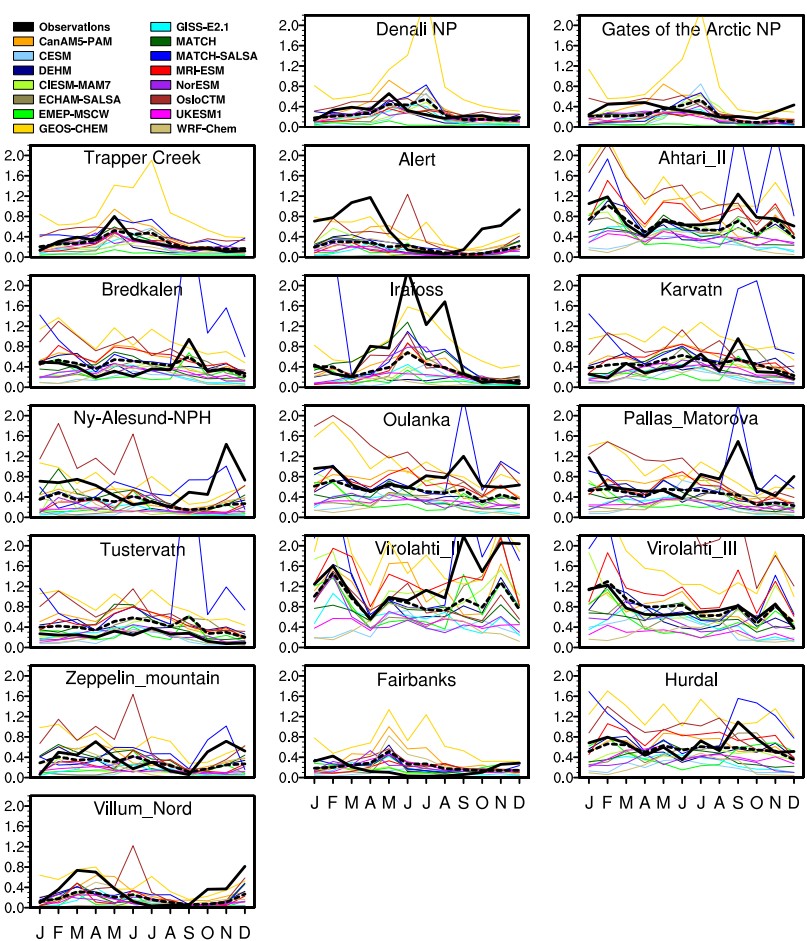

**Figure 20.** Modelled (thin colored lines) and measured (thick black line) monthly mean $SO_4^{2-}$ concentrations (in $\mu g\ m^{-3}$) at surface Arctic measurement sites in 2014-15. Multi-model mean is shown by the black dashed line.





The seasonal cycle for observations grouped together is shown in Figure S1, showing a consistent seasonal cycle for 2008-9 as seen in the observations. Most models (e.g. CanAM5-PAM, DEHM, MATCH, OsloCTM) are able to capture the seasonal

cycle well. However several models (e.g. CESM, CIESM-MAM7, ECHAM-SALSA, and EMEP-MSCW) strongly underestimate observed springtime peak values. Conversely, the models and the measurements showed a weaker seasonal cycle during the 2014-15 time period (Figure S2). It may be partly due to the local pollution sources in the Arctic during wintertime (e.g., Fairbanks). Those highly localised pollution events, caused by local emissions getting trapped in a stable boundary layer occur on scales that are smaller than the model resolutions employed here can represent. Many models are also missing chemical

formation processes for $SO_4^{2-}$ in the absence of sunlight, which may explain underestimations seen in winter (e.g. Moch et al. (2018); Alexander et al. (2009)). An evaluation of $SO_2$ could help with our understanding, but was beyond the scope of this study. A lack of dark chemistry may be true for organic aerosol, discussed in the next section as well.

From October to December 2014, the Honoluraun volcanic eruption may have elevated $SO_4^{2-}$ concentrations at some locations in the Arctic. However, in our model-measurement comparisons, there does not appear to be a large underestimate during

those months, which implies that model performance wasn't impeded by not including those volcanic emissions.

As mentioned above, the uncertainty in wet deposition could be a significant factor in atmospheric $SO_4^{2-}$ model biases. Previous studies have shown that models have too much washout in winter and not enough wet deposition in summer, leading to a "flatter" seasonal cycle than observed (e.g., Fig 20; Browse et al. (2012); Mahmood et al. (2016)). As with BC in the previous section, the $SO_4^{2-}$ deposition was evaluated here in the same manner. The average measured $SO_4^{2-}$ deposition fluxes

from ice cores for all locations (only Greenland was available here) is 18 mg/m$^2$/yr. The lowest observed fluxes are found at D4 (12 mg/m$^2$/yr) and highest at ACT11D (30 mg/m$^2$/yr). The model average for all locations is overestimated by around 20% compared to measured fluxes. This is similar to the model biases in atmospheric $SO_4^{2-}$ concentrations.

Therefore, the general model evaluation for $SO_4^{2-}$ indicates that while there is a large variability in models results, as with BC, models underestimate surface $SO_4^{2-}$ in the Arctic. The evaluation of $SO_4^{2-}$ deposition in the Arctic is similar to BC, with

both overestimating deposition.

### 4.6   Organic aerosol

Unfortunately, there is only one high-Arctic station with OA measurements (Alert, NV, Canada), however there are a few additional stations measuring OA at the sub-Arctic (still >60°N). These are all shown in Figure 23. The seasonal cycles are shown in Figure 24, and the model vs. measurement scatter plot along with some comparison statistics are presented in Figure

550   25.

Model biases have a large range of ±200% at the different locations, but the multi-model mean for the region is +65%. At mid-latitudes (30-60°N), measurements are conducted mainly in the U.S, where the multi-model mean bias is +83% for the 2014-15 average (not shown).

Several models (CanAM5-PAM, DEHM, CIESM-MAM7, ECHAM-SALSA, GEOS-Chem, MRI-ESM2, NorESM, OsloCTM,

and UKESM1) are able to simulate the summertime peak in Arctic OA concentrations, however the other seven models in Fig-



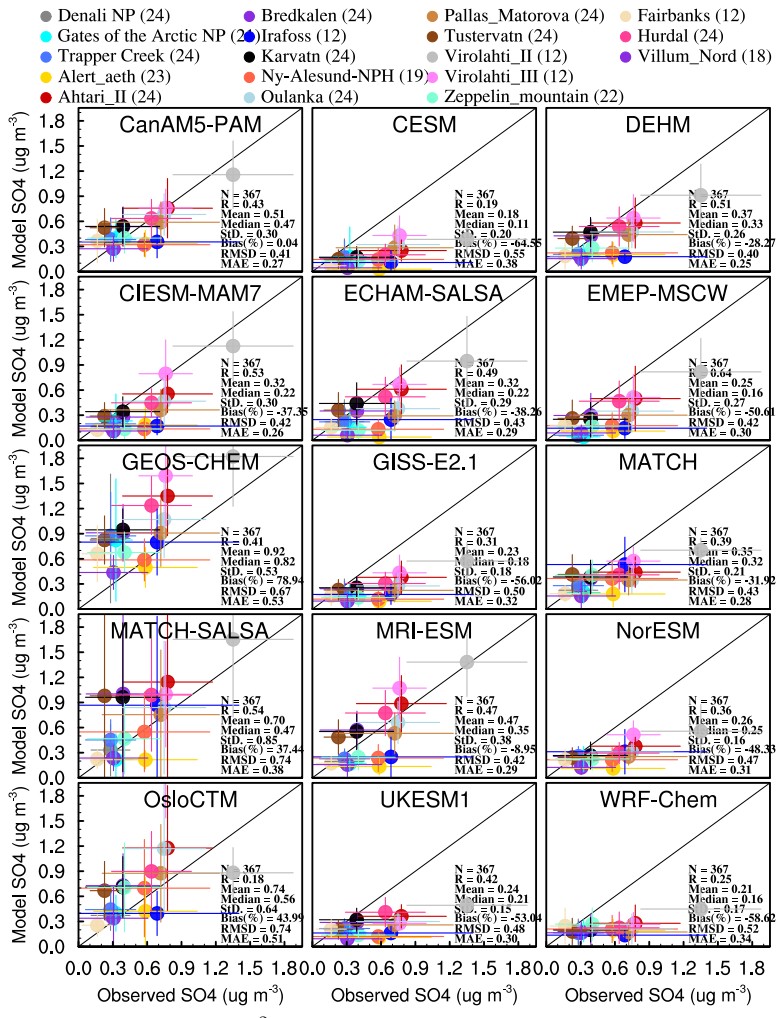

**Figure 21.** Modelled vs measured surface $SO_4^{2-}$ concentrations at Arctic measurement sites for 2014-15. Filled circles represent the mean at a site and the lines represent +/- one standard deviation from mean based on available monthly mean data. Numbers in brackets show the number of months used, with a maximum of 24.





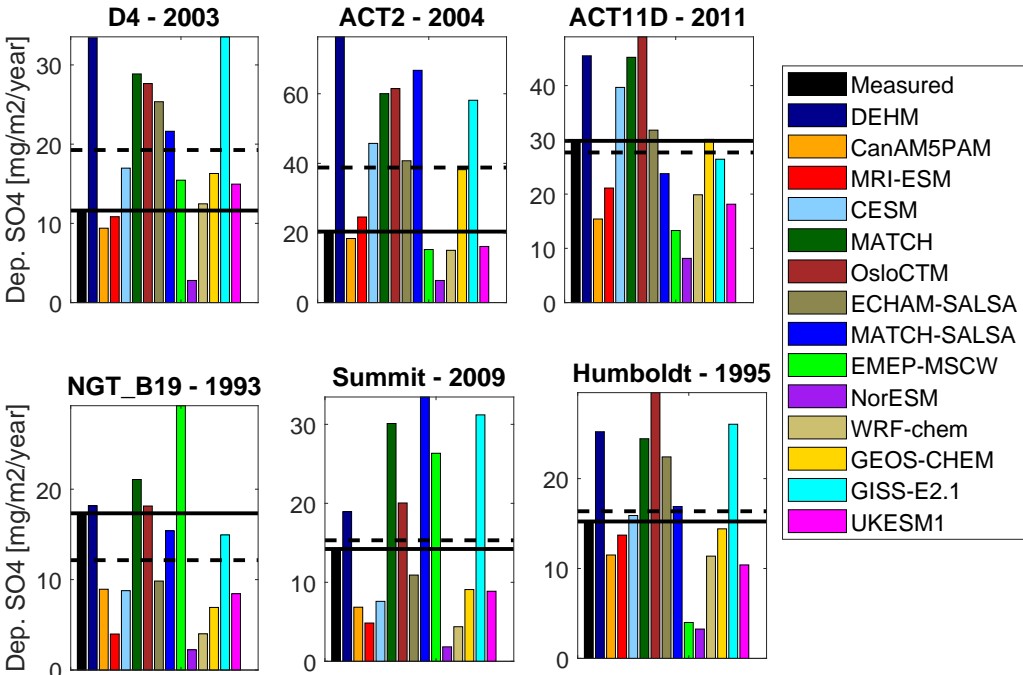

**Figure 22.** $SO_4^{2-}$ deposition fluxes for the Greenland ice core locations shown in Figure 6.3.3.1.1. The observed fluxes are plotted in black and a black line indicates the level of the average observed flux, the black dashed line is the multi-model mean at that location. The period used for plotting is based on all available years after 1990, the title indicates the last available year from the ice core record.

ure 24 simulate a seasonal cycle that is too flat or peaks at the wrong time (e.g., the CESM seasonal cycle peaks too early in the year).

Figures 23 and 25 both show that most models consistently overestimate Alaskan OA and underestimate European OA, consistent with our assessment of other species showing that models are likely overestimating wildfire influence in summertime 560 Alaska. The comparison statistics in Figure 25 show highly varying comparison statistics.

### 4.7 Fine particulate matter

$PM_{2.5}$ is partly connected to direct and indirect climate effects via its interactions with clouds. It is mainly composed of BC, $SO_4^{2-}$, OA, $NO_3^-$, and $NH_4^+$, as well as crustal material, sea-salt and water, though the water component is often dried off during measurements. Model biases of those species will contribute to the total $PM_{2.5}$ biases. In order to minimize concern 565 over errors in the last three - mainly natural - $PM_{2.5}$ components, for this analysis, we have used a consistent set of CM, SS,

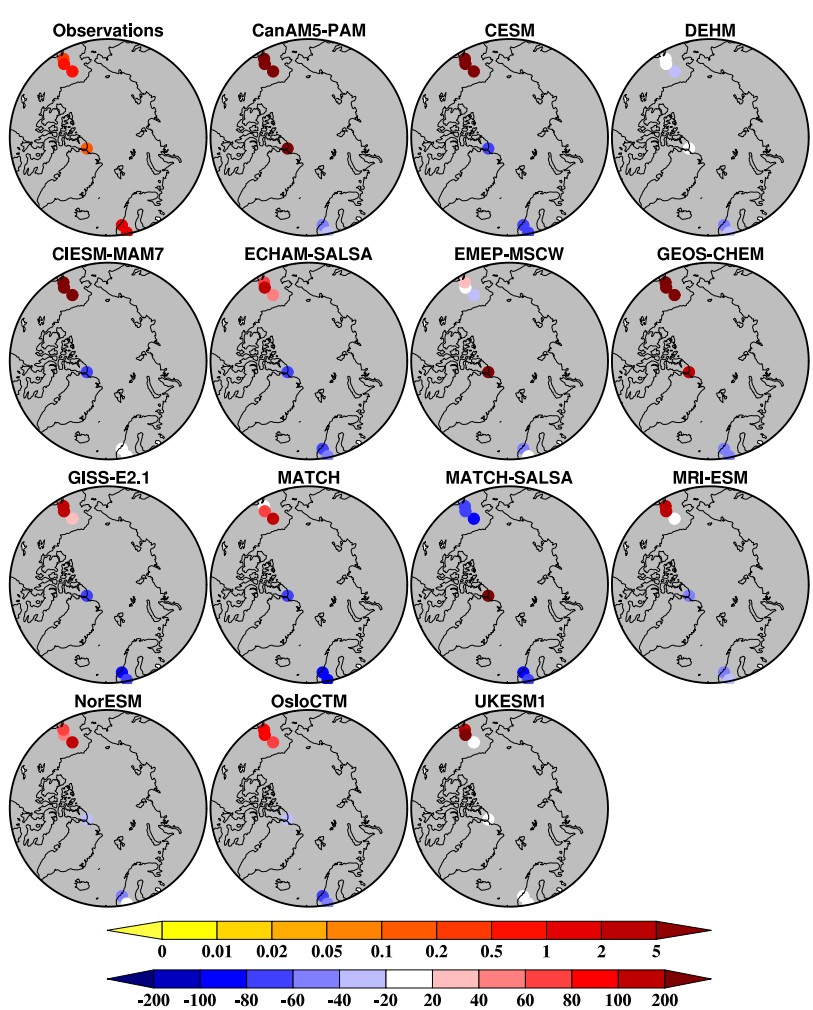

**Figure 23.** Mean OA concentrations (in $\mu$g m$^{-3}$, top colour bar) at surface Arctic measurement sites, and model bias [as (model-measurement)/measurement in %, bottom colour bar] for 2014-15. Results from 2008-9 are similar and not shown.



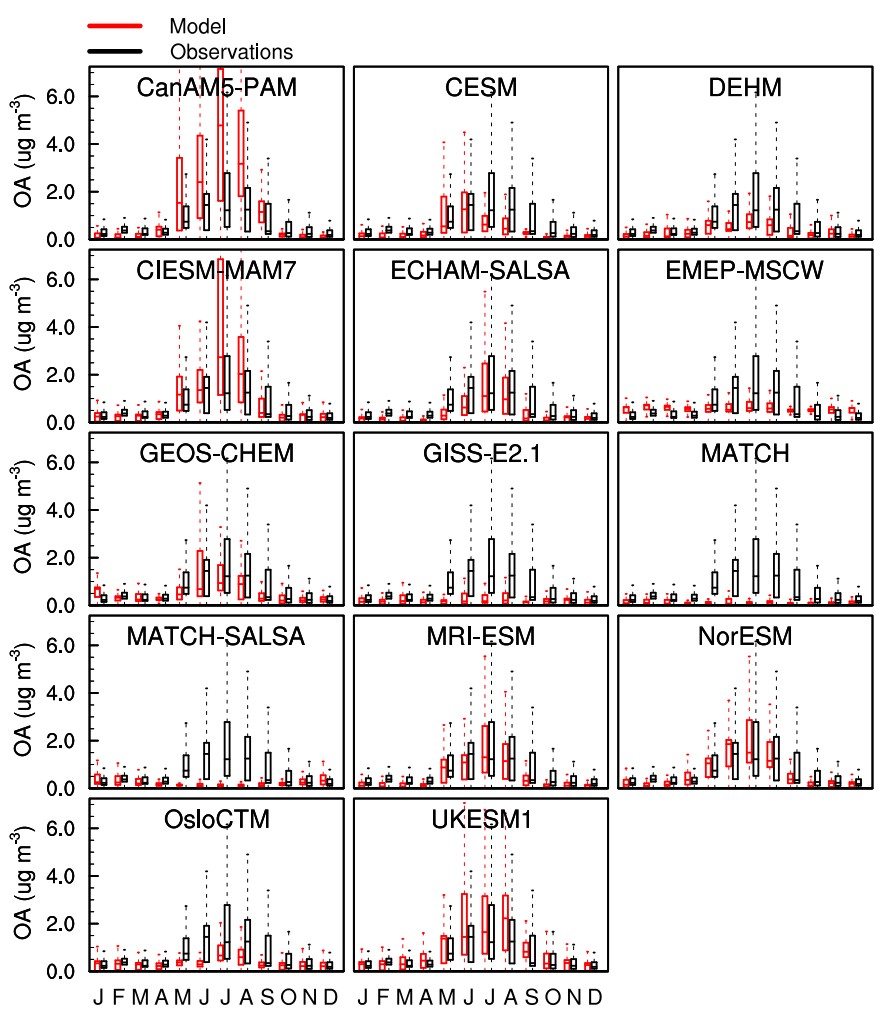

**Figure 24.** Modelled (thin colored lines) and measured (thick black line) monthly mean OA concentrations (in $\mu$g m$^{-3}$) at surface Arctic measurement sites in 2014-15. Multi-model mean is shown by the black dashed line.



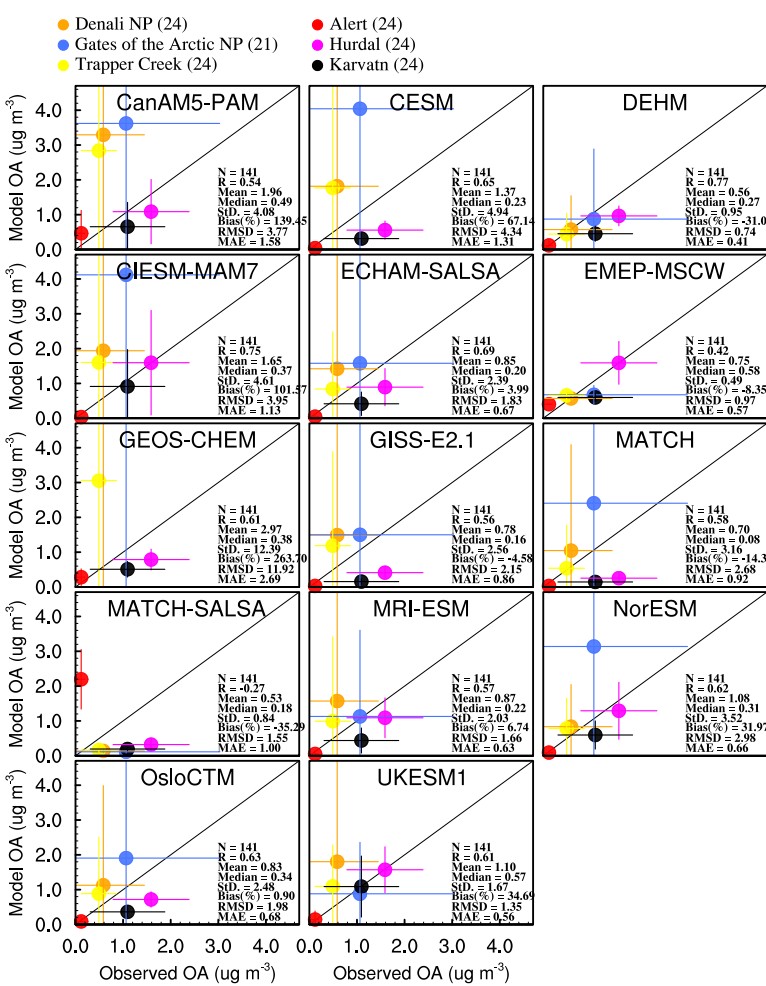

**Figure 25.** Modelled vs measured surface OA concentrations at Arctic measurement sites for 2014-15. Filled circles represent the mean at a site and the lines represent +/- one standard deviation from mean based on available monthly mean data. Numbers in brackets show the number of months used, with a maximum of 24.





and H$_2$O in PM$_{2.5}$ calculations, while using each models results for all of the other PM$_{2.5}$ components. Thus, the inter-model variability is only due to differences in the anthropogenic components of PM$_{2.5}$.

While BC, SO$_4^{2-}$, and OA were discussed above, it is beyond the scope of this project to evaluate the other major PM species, which, aside from water, have a smaller radiative impact. Note that the analysis in this section is focused on sub-Arctic
and mid-latitude sites, closer to human populations, rather than remote high Arctic sites due to a lack of data. PM$_{2.5}$ is not a typical parameter included in the longer-term remote Arctic observations. Since PM$_{2.5}$ has important health impacts, it is well-measured at air quality monitoring networks.

The model PM$_{2.5}$ biases at several locations in the United States, Europe and China are within 60-80% range. However, some models (CanAM5-PAM, CIESM-MAM7, GEOS-CHEM, GEM-MACH and Oslo-CTM) show biases larger than 200%,
especially in the Western US and Alaska. The spatial patterns in the model biases, such as being higher in the U.S. and lower in Asia, are the same for both 2008-9 (not shown) and 2015 26 though the biases are overall skewed slightly lower for 2008-9 than they are in 2014-15. The EMEP MSC-W results are consistent with EMEP annual evaluations for Europe, where the model underestimates PM$_{2.5}$ by 10-25%, including a few Arctic sites in Norway and Finland (Gauss et al. (2020), and annual model Evaluation reports at https://www.emep.int/publ/common_publications.html).

The simulated surface level SLCFs were quite sensitive to the different meteorological conditions such as boundary layer stability and levels of photochemistry which differed between the two time periods chosen in this study. 2014-15 was also a time period with more wildfires compared to 2008-9. For example, according to the CMIP6 emission data that were used in most of the models, emissions of BC from Canadian wildfires in 2014-15 were 340% higher than in 2008-9 whereas the emissions from the USA and Russia were similar for these years. Given the very intense wildfire emissions and low anthropogenic emissions
in northern Canada in 2014-15, differences in simulated PM$_{2.5}$ concentrations over Canada and Alaska can be partly attributed to differences in simulations of wildfire aerosols in the models.

Some models (CanAM5-PAM, CESM, CIESM-MAM7, GEOS-Chem, and WRF-Chem) simulate higher PM$_{2.5}$ and more variable PM$_{2.5}$ in the summertime (e.g., Fig S.3 in the supplemental material). While this is seen to some extent in the observations, this may be due to the way fire emissions and sea salt emissions are treated in these models. Fire emissions, fire
plume injection height, and plume rise are all highly uncertain model parameters and a subject of ongoing research (e.g., Urbanski (2014); Heilman et al. (2014); Paugam et al. (2016)). Indeed, the individual model PM$_{2.5}$ Arctic biases are more tightly clustered for 2008-9 when there were fewer fires. Mölders and Kramm (2018) showed that Arctic PM$_{2.5}$ seasonal pollution is mainly due to local air pollution in the winter and due to fires in the summer.

Figure 27 shows that the annual mean simulated PM$_{2.5}$ concentrations compare well with observations and the correlation
coefficients are relatively high (R= 0.8 or higher for all models). The high concentrations in China and low concentrations in US and Europe are captured by the models providing confidence for health impact assessments that utilize these model results.

## 4.8 Multi-species summary

The 2014-15 average modelled percent biases for surface concentrations of SLCFs are shown in Figure 28 for each model and the multi-model mean (mmm); and Figure 29 with box and whiskers showing the range of how the model percent biases



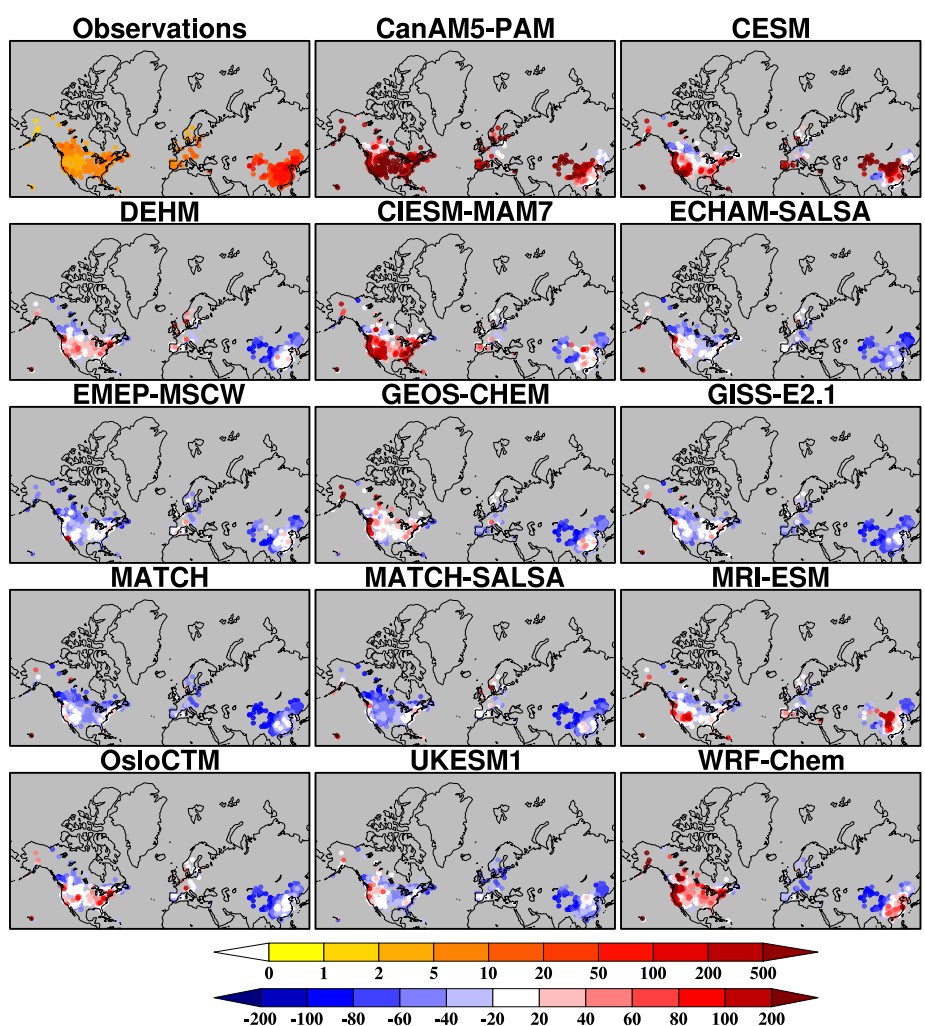

**Figure 26.** Measured ground-level PM$_{2.5}$ concentrations ($\mu$g m$^{-3}$) and model biases [as (model-measurement)/measurement in %, bottom colour bar] for 2015. The upper colorbar represents observations and lower bar represents model biases.



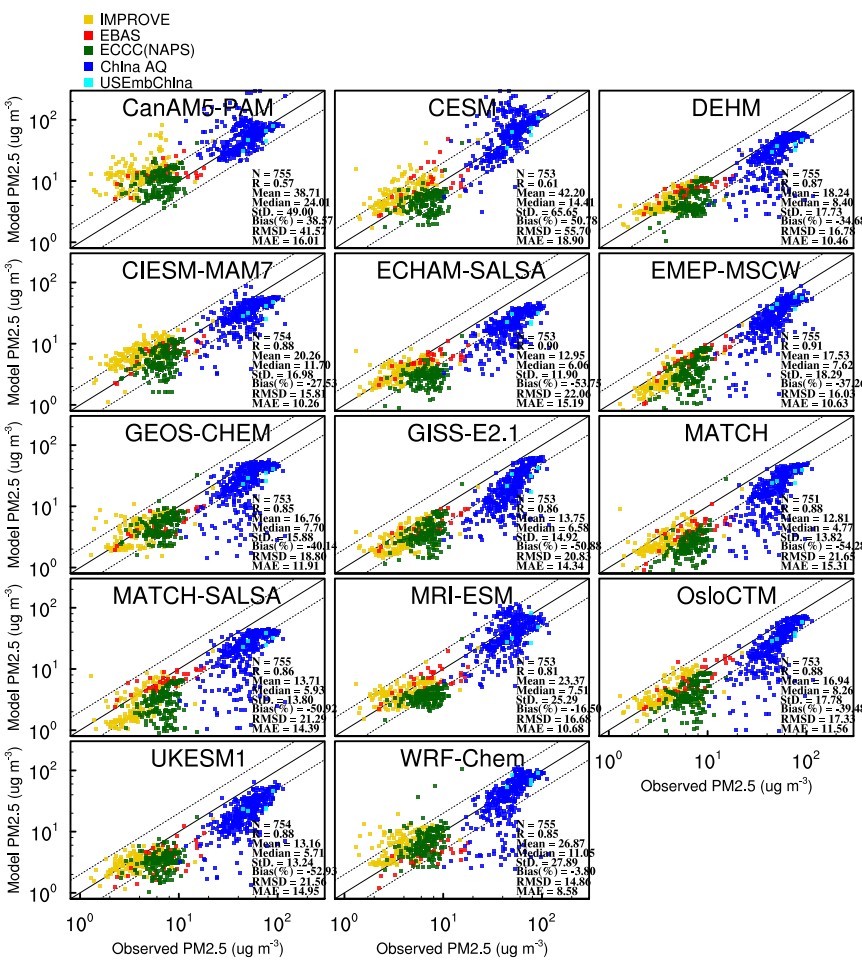

**Figure 27.** Annual mean PM$_{2.5}$ comparisons between station observations and model simulations for year 2015.





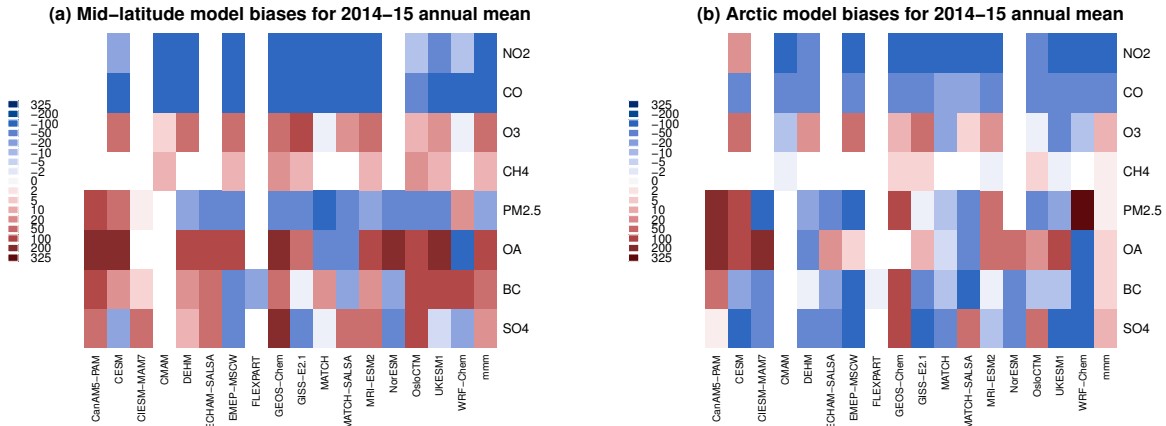

**Figure 28.** Mean 2014-15 model % biases for each model and the multi-model mean, for (a) mid-latitudes, and (b) the Arctic. Note that the colour scale isn't linear.

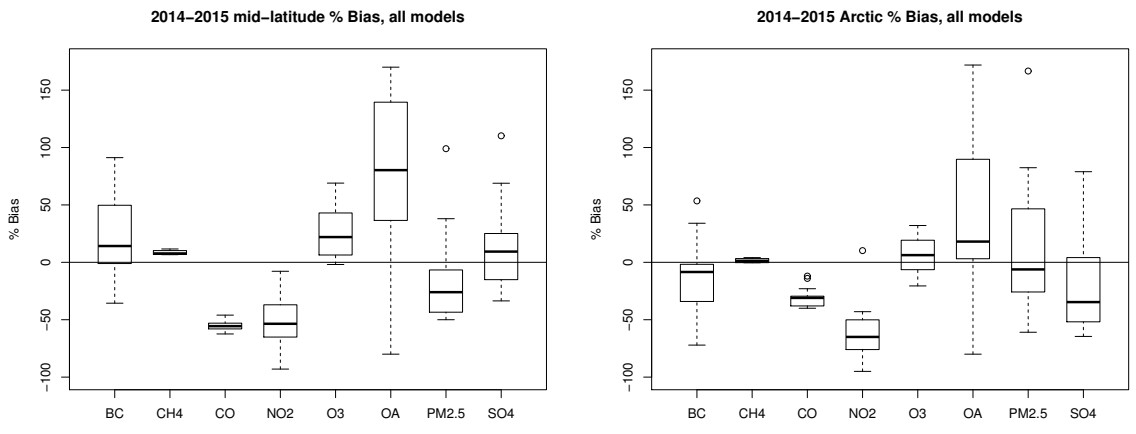

**Figure 29.** (a) Mid-latitude and (b) Arctic % biases from all models and all species together for 2014-2015. Thick line is the median, boxes extend to the 25th and 75th percentiles, and whiskers extend to 1.5 times the interquartile range.

compare across species. These two figures are based on the observations/locations shown Figures 3, 6, 9, 10, 13, 19, 23, and 26 (with additional American observations from the IMPROVE network for BC, $SO_4^{2-}$, and OA). Figure 28(b) shows that, for surface Arctic concentrations, no one model performs best for all species, but that the mmm performs particularly well.

Figure 29 shows that the model biases vary quite a bit among SLCF species for both the mid-latitudes and the Arctic. It is important to note that there are many more measurement locations at mid-latitudes compared to in the Arctic. BC, CH$_4$, O$_3$, 605 and PM$_{2.5}$ have the smallest model biases out of the SLCFs of this study, whereas OA, CO and NO$_2$ have larger model biases.





The summertime evaluation of surface $O_3$, BC, and OA all imply that models overestimate the amount of these pollutants coming from wildfires in the western Arctic. This could be due to uncertainties in wildfire emissions, fire plume transport, or in the case of $O_3$ and secondary OA, the plume chemistry. The model overestimations in the summer could be due to a combination of all of these uncertainties.

Aside from the surface concentrations discussed above, our analysis from ACE-FTS, TES, and the aircraft datasets show that $CH_4$, $O_3$, and BC model biases all imply that modelled tropopause height is likely too low. Tropospheric species like $CH_4$ and BC should drop rapidly above the tropopause, but model biases increase sharply at that point. Stratospheric species like $O_3$ should increase rapidly above the tropopause, but model biases decline sharply there.

## 5 Conclusions

In this study, we evaluated the SLCF simulation capabilities of 18 models that were used in the 2021 AMAP SLCF Assessment Report. Our conclusions are grouped into the questions we aimed to answer in the introduction:

### 5.1 How well do the AMAP SLCF models perform in the context of measurements and their associated uncertainty?

Recall that the in situ SLCF measurements had the following reported uncertainties: $CH_4$ 1%, $O_3$ 3%, CO 5%, $NO_2$ 5-100%, BC 200%, $SO_4^{2-}$ 20%, OA 20%, and $PM_{2.5}$ 1-6%. However, since the variability in measurements from different techniques 620 was only really taken into account for the BC uncertainty, and since we are comparing annual mean results to each other, it is not a fair comparison to say that models and measurements agree with each other if model biases are within the reported measurement uncertainty range. However, we do use those numbers as a rough guideline for "good" model performance, in the absence of other quantitative criteria.

Some model annual mean biases were within those uncertainty ranges. For example, CMAM, MRI-ESM2, and UKESM1 625 simulate Arctic $CH_4$ to within 1%, thus agreeing the $CH_4$ measurements. However, at mid-latitudes, they are all out of range at around +6-10%. MATCH and WRF-Chem simulated mid-latitude $O_3$ to within 2%, but only MATCH-SALSA was within 3% in the Arctic region. The Arctic $NO_2$ measurements are highly uncertain at around 100%, so all of the models agreed with Arctic $NO_2$ measurements. However, in the higher-$NO_2$ mid-latitudes environment, $NO_2$ measurement uncertainty is at the smaller end of the range. OsloCTM, and WRF-Chem mid-latitude $NO_2$ biases were within 10%. All models agree with BC 630 measurements in both mid-latitudes and the Arctic, as all biases are less than 80%. CESM, CIESM-MAM7, DEHM, MATCH, UKESM1, and WRF-Chem all simulate mid-latitude $SO_4^{2-}$ to within 20%. But only CanAM5-PAM, and MRI-ESM2 do the same for the Arctic region. OA had some of the largest model biases (Figure 29), though ECHAM-SALSA, EMEP-MSC-W, GISS-E2.1, MATCH, and OsloCTM are all within 20% in the Arctic, though none at mid-latitudes. Finally, with the small uncertainty on $PM_{2.5}$, only CIESM-MAM7 in the mid-latitudes, and GISS-E2.1 in the Arctic agree within 2%.

To summarize the mmm annual mean performance, it "matches" observations in the Arctic for $CH_4$, $NO_2$, BC, $SO_4^{2-}$, OA, and $PM_{2.5}$ - and as such, the mmm has the best overall performance for the Arctic. In the mid-latitudes the mmm "matches" observations for BC, and $SO_4^{2-}$ only.



Regarding the comparisons of trace gases to the TES, MOPITT, and ACE-FTS satellite measurements (which have roughly 5-20% uncertainty), models agree well. Free-tropospheric distributions of trace gases are somewhat easier to simulate as common problems like a too-stable boundary layer or too much deposition do not negatively impact the free-tropospheric SLCF distributions. The variability in the free troposphere is smaller compared to at the surface as well. It is also because of the previously noted difference between the spatial range that remote measurements cover being more akin to the spatial scale of model grid boxes, compared to the point measurements from in situ observations.

### 5.1.1 What do the best-performing models have in common?

There were no models that performed best for all SLCF species and for all regions, highlighting that it is difficult for any one model to bring together numerous complex processes and get results comparable to observations for all SLCFs. This would involve simulating aerosols and chemistry together with the right transport processes, meteorology and clouds, which is difficult, especially for a remote region like the Arctic where parameterisations might have been built on datasets that are not always applicable there. In addition, studies like that of Tsigaridis et al. (2014) have showed that there was no clear change in model skill (in that case for OA) with increasing model complexity.

However, several models such as CanAM5-PAM, DEHM, NorESM, and MATCH have better representation of the vertical distribution of BC. DEHM and MATCH also had relatively small biases throughout the $O_3$ tropospheric profile (CanAM5-PAM and NorESM did not simulate gas-phase SLCFs). MATCH in particular has the smallest surface $O_3$ bias at mid-latitudes, which may be related to its high vertical resolution in the boundary layer (the lowest two layers are 20 m thick and four lowest layers are below 150 m). These models are a mix of air quality and climate-focused models, thus it is important to note that there is no obvious difference between climate and air quality model biases for annual mean SLCFs. Despite the lack of complex tropospheric chemistry, CMAM had some of the lowest $O_3$ biases at both mid-latitudes and the Arctic. This may imply that the more complex chemistry is not needed in the context of climatological tropospheric $O_3$ for climate studies (though of course, $O_3$ on shorter time scales would need more complete complete tropospheric $O_3$ chemistry). In the lower-stratosphere, however, models with simplified/climatological $O_3$ schemes did not perform as well as models that had full stratospheric chemistry included.

### 5.1.2 Are there regional patterns in the model biases?

Generally speaking, when comparing the mid-latitude model biases to those of the Arctic, they all skew more negative (Figure 29), except for CO and $PM_{2.5}$, implying a lack of long-range transport to the Arctic. The best Arctic results for BC throughout the tropospheric profile were from models (CanAM5-PAM, DEHM, MATCH, WRF-Chem) that simulated the vertical mixing of BC well at mid-latitudes. These first three of the four models were nudged to the ERA-Interim analysis and WRF-Chem was nudged to the NCEP FNL analysis. The Arnold et al. (2015) study showed that a key determinant in model differences for PAN export relative to CO was the meteorology used in the models. Their results implied that the ERA-Interim models had more efficient vertical transport and mixing in mid-latitude source regions compared to GEOS-driven models. In the current study, Arctic BC was greatly underestimated throughout the Arctic profile by MATCH-SALSA, EMEP MSC-W, and GEM-MACH,





which had BC concentrations at mid-latitudes dropping much too low in the free troposphere. Those models had different sources of meteorology (Table 2) and some may have insufficient convection schemes. For example GEM-MACH is missing sub-grid scale deep and shallow convection, which is important for the exchange between planetary boundary layer and free troposphere, and thus for transport at mid and high latitudes. This subject could be studied further via sensitivity tests with and

without nudged meteorology, while keeping the aerosol physics the same.

### 5.1.3 Are there patterns in the model biases between SLCF species?

Some patterns one might expect between SLCF species were not demonstrated in this study's results. For example, both $O_3$-precursors, CO and $NO_2$, are too low in models, though that underestimation is worse in the winter. Despite that, surface $O_3$ tends to be overestimated, though that is overestimation is mainly in the summer. Also the west-east pattern in Arctic surface

$O_3$ was to be overestimated in Alaska and underestimated in Scandenavia. However, for CO and $NO_2$ those skewed biases are reversed.

At mid-latitudes $SO_4^{2-}$, BC, and OA are biased high in most models, yet despite that, $PM_{2.5}$ is underestimated. Thus, the $PM_{2.5}$ biases must be significantly influenced by the other, mainly natural, aerosol components.

In addition to expected patterns, there were no other discernible patterns in model biases between SLCF species either.

## 5.2 How does the model performance impact model applications, such as simulated climate and health impacts?

In the AMAP (2021a) report, these models go on to be used to simulate future emission scenarios, and from those results, the future temperature changes due to these SLCFs are predicted. They are further used to determine future changes to human health due to the changes in SLCFs. Given the model evaluation of this study, we have determined that using the multi-model mean to predict SLCF climate impacts is generally robust. Considering the SLCFs with greatest radiative impact ($CH_4$, $O_3$, BC,

and $SO_4$), the mmm was within $\pm 25\%$ of the measurements across the northern hemisphere. The mmm also performed very well for $PM_{2.5}$, which is a main component considered for human health impacts. Thus, for climate studies, where relatively large regions time periods are considered, the model performance is sufficient.

### 5.3 What processes should be improved or studied further for better model performance?

The model evaluation in this study brought about results that have been reported in previous publications, and several notable

issues remain. Here we recommend some future work that may help improve model performance.

– Models simulate too much surface $O_3$ at mid-latitudes, and this may be due inadequate treatment of dry deposition (Val Martin et al., 2014), and/or not including parameterizations for the shade provided by vegetation that reduces photo-chemistry, as reported in Makar et al. (2018). That said, MATCH had the smallest mid-latitude surface $O_3$ bias without accounting for canopy shading, hence, precursor emissions, vertical mixing, deposition, and $O_3$ chemistry all have a role

in model $O_3$ results, and errors in these may sometimes cancel out.



- There are a number of indications that simulated boundary layers are too stable (not enough vertical lifting of SLCFs, too much $O_3$ titration, too much BC and $SO_4$ deposition). Therefore, increased convection at mid-latitudes may be needed. However, this hypothesis is opposite to that found in Allen and Landuyt (2014), which found excessive tropical convection caused CMIP5 models to overestimate BC aloft. It is thus important to evaluate models specifically for export and long-range transport events driven by different mechanisms (e.g. frontal export, convective lifting), which is a focus within the PACES initiative (Arnold et al., 2016).

- The $O_3$, BC, $SO_4^{2-}$, and $PM_{2.5}$ model biases were all high in the Alaskan summertime, implying that many models may simulate too much pollution from wildfires there. Models need improved wildfire parameters for emissions, plume height, and plume chemistry. For example, fire emissions inventories GFED4, GFASv1.2, FINNv1.5 vary by up to a factor of 3 for BC emissions (AMAP, 2021a); light and temperature attenuation under smoke plumes means less $O_3$ is produced than precursor concentrations may imply; and plume rise and injection height need to be accurate for long range transport.

- Modelled deposition is highly uncertain, and there is evidence here that some models have too much deposition of BC at mid-latitudes. However, deposition measurements are scarce, even at mid-latitudes, and more of those measurements are needed to constrain models.

- Additional, preferably long-term, OA and $PM_{2.5}$ measurements are needed in the high-Arctic. Both are expected to be important to Arctic conditions in the future, with increasing wildfires, and shipping influencing the Arctic atmosphere, and the lack of those measurements is problematic for constraining models.

- An evaluation of $SO_2$ would help to determine if the model biases in $SO_4^{2-}$ are due to transport, to emission uncertainty, or if it can be explained by the uncertainty in chemistry. The removal of particles represents a large uncertainty but without $SO_2$ (and DMS) it cannot be concluded that the removal is too fast.

Therefore, we conclude that sensitivity tests for the above-mentioned model processes will be important for further understanding and improving model performance for SLCFs. But just as important is having additional Arctic measurements, and the continuation of existing Arctic measurements in order to assess the model improvements.

*Code and data availability.* The models' output files in netCDF format from the simulations used in this project can be found here: http://crd-data-donnees-rdc.ec.gc.ca/CCCMA/products/AMAP/.

Some of the models' code are available online at the following locations:

CanAM5-PAM: https://gitlab.com/cccma

CESM2: https://www.cesm.ucar.edu/models/cesm2/

ECHAM-SALSA: The codes used for the ECHAM-SALSA simulations are available from the ECHAM-HAMMOZ repository under https://redmine.hammoz.ethz.ch/projects/hammoz/repository/1/show/echam6-hammoz/branches/fmi/AMAP/AMAP_evaluation, after obtaining the HAMMOZ license.





FLEXPART: https://www.flexpart.eu

GEOS-Chem: http://wiki.seas.harvard.edu/geos-chem/index.php/GEOS-Chem_12#12.3.2

GISS-E2.1: https://www.giss.nasa.gov/tools/modelE/ and https://simplex.giss.nasa.gov/

NorESM: https://github.com/NorESMhub/NorESM

Oslo CTM: https://github.com/NordicESMhub/OsloCTM3

and the other models' code may be available upon request.

The model evaluation programs can be found on gitlab here: https://gitlab.com/cynwhaley/amap-slcf-model-evaluation.

The surface monitoring datasets are available online here:

WDCGG for $CH_4$: https://gaw.kishou.go.jp/login/user

EBAS for European (EMEP) and several Arctic locations: http://ebas.nilu.no/

NAPS: https://open.canada.ca/data/en/dataset/1b36a356-defd-4813-acea-47bc3abd859b

IMPROVE: https://views.cira.colostate.edu/fed/Express/ImproveData.aspx

Beijing Air Quality for China: https://beijingair.sinaapp.com/

$PM_{2.5}$ from the US embassy in China from the data portal: www.stateair.net.

The satellite measurement data used in this study are available online here:

ACE-FTS v4.1 measurements are available, following registration, from http://www.ace.uwaterloo.ca

TES: https://tes.jpl.nasa.gov/tes/data/products/lite

MOPITT: https://www2.acom.ucar.edu/mopitt/products

## Appendix A: Model Descriptions

The eighteen models used in this study are described in each subsection below. Table A1 contains further information summarizing the models' setup for the AMAP SLCF simulations.

## A1 CanAM5-PAM

The Canadian Atmospheric Model version 5 (CanAM5), with Piecewise lognormal approximation Aerosol Model (PAM) was used. CanAM5-PAM is an improved version of CanAM4 (von Salzen et al., 2013). The improvements include a higher vertical resolution, improved parameterizations for land surface and snow processes, DMS emissions, and clear-sky radiative transfer. CanAM5-PAM has 49 vertical levels extending up to 1 hPa with a resolution of approximately 100 m near the surface. Model

simulations are performed using a spectral resolution of T63, which is equivalent to the horizontal resolution of approximately $2.8° \times 2.8°$. The model uses separate parameterizations for layer and convective clouds. Aerosol microphysical processes are based on the piecewise lognormal approximation (von Salzen, 2006; Ma et al., 2008; Peng et al., 2012; Mahmood et al., 2016, 2019; AMAP, 2015a). The model simulates binary homogeneous nucleation of sulfuric acid and water vapor. Newly formed particles grow by condensation and coagulation.



A detailed description of parameterizations of ocean DMS flux to atmosphere, oxidation, and removal processes is provided in Tesdal et al. (2016). In-cloud production of sulfate requires $O_3$ and hydrogen peroxide ($H_2O_2$) as oxidants (von Salzen et al., 2000), with oxidant (OH, $NO_3^-$, $H_2O_2$, $O_3$) concentrations specified as climatological results from CMAM. Dry deposition of aerosol depends on concentrations of aerosols in the near-surface model layer (Zhang et al., 2001). Wet deposition includes in-cloud scavenging in both convective clouds and layer clouds, as well as below-cloud scavenging.

Cloud droplet number concentrations are calculated based on the assumption of a parcel of air which ascends from the subcloud layer into the cloud layer with a characteristic vertical velocity (Peng et al., 2005), where the standard deviation of the subgrid-scale cloud vertical velocity probability distribution is parameterized using the approach by (Ghan et al., 1997). Aerosol particles that are suspended in the parcel of air may activate and grow into cloud droplets by condensation of water vapor. A numerically efficient solution of the condensational droplet growth equation (e.g., Seinfeld and Pandis (2006)) is employed for this purpose. In grid cells that are affected by clouds, CanAM5-PAM accounts for cloud albedo and lifetime effects (first and second aerosol indirect effects) as well as semi-direct effects.

## A2   CESM

The Community Earth System Model version 2 (Danabasoglu et al., 2020) is an ESM that can be configured in many different ways. The configuration applied for this assessment utilized the Community Atmosphere Model (CAM) version 6 and Modal

Aerosol Model (MAM4) with 4 mixed-species aerosol modes (Liu et al., 2016). CAM6 employs a spectral element dynamical core (Lauritzen et al., 2018). Type 0 and Type 1 CESM runs were conducted at $1.9° \times 2.5°$ horizontal resolution, while Type 3 runs at $0.9° \times 1.25°$, all with 32 vertical layers. For Type 0 and Type 1 simulations, CESM version 2.0 was used with "CAM6-chem" representations of chemical reactions (Emmons et al., 2020b), enabling prognostic simulation of tropospheric ozone concentrations, along with a volatility basis set (VBS) parameterization for the formation of secondary organic aerosols (SOA) (Tilmes et al., 2019) and stratospheric chemistry. CAM6-chem is coupled to the interactive Community Land Model (CLM5),

which provides biogenic emissions, calculated online using the MEGANv2.1 algorithm (Guenther et al., 2012), and handles dry deposition. Tracked aerosol species simulated by MAM4 include sulfate, primary and aged black carbon and organic matter, dust, sea-salt, and secondary organic aerosols. Both sea salt and dust emissions are calculated on-line and are highly sensitive to the surface wind speed (Mahowald et al., 2006a,b). These runs were also forced with prescribed SSTs and sea-ice concentrations, created from merged Reynolds/HADISST products as in Hurrell et al. (2008). Type 3 transient runs utilized

CESM version 2.1.1 without atmospheric chemistry and with fully-coupled atmosphere, ocean, land, and sea-ice components (component set "BSSP245cmip6"), as applied to simulate future scenarios for CMIP6. All CESM runs specified global-mean mixing ratios of methane and carbon dioxide.

## A3   CIESM-MAM7

CIESM-MAM7 is the Community Integrated Earth System Model (CIESM) (Lin et al., 2020) using the Modal Aerosol Model (MAM7) with 7 mixed-species aerosol modes (Liu et al., 2012). Current CIESM version 1.1 (see Table 1 of Lin et al. (2020)) is based on NCAR Community Earth System Model (CESM version 1.2.1) with several novel developments and modifications





aiming to overcome some persistent systematic biases, such as the double Intertropical Convergence Zone (ITCZ) problem and underestimated marine boundary layer clouds. CIESM-MAM7 employs a finite volume dynamical core with $0.9° \times 1.25°$

for horizontal resolution and 31 layers for vertical resolution. The large-scale meteorology (horizontal wind field) is nudged towards ERA-Interim reanalysis data and the relaxation time is set to 6 hours. In CIESM-MAM7, the primary emission of black carbon (BC), organic carbon (OC), ammonia ($NH_3$), volatile organic compounds (VOCs), sulfur dioxide ($SO_2$) and oxidizing gases ($H_2O_2$, $O_3$, OH) are prescribed by the input data uniformly provided by AMAP-SLCF group. The emission amount of dust (DU) and sea salt (SS) are calculated online. Aerosol size distributions in CIESM-MAM7 are described by the seven

overlapping log-normal distributions, including Aitken, accumulation, primary carbon, fine dust and sea salt, coarse dust and sea salt modes. The geometric standard deviation of each mode is prescribed (see Table 1 of Liu et al. (2012)). A simplified gas and liquid phase chemistry is included in CIESM-MAM7. $SO_2$ and dimethyl sulfate (DMS) can be oxidized to sulfuric acid gas ($H_2SO_4^{2-}$) and then condenses to form the sulfate aerosols, while the evolution of oxidizing gases is not considered. Primary organic matter (POM) and BC are emitted to the primary carbon mode, then are aged and transferred to the accumulation

mode by condensation of $H_2SO_4^{2-}$, $NH_3$ and semi-volatile organics and by coagulation with Aitken and accumulation modes. The effect of stratospheric sulfate aerosol from volcanic emission on radiative forcing is considered, by following the CMIP6 procedure (Thomason, 2012). No specific stratospheric chemistry is included in CIESM-MAM7.

## A4   CMAM

The Canadian Middle Atmosphere Model (CMAM) is based on the third generation CanAM model, with the model lid raised

to approximately 95 km and the necessary radiative processes for the mesosphere included (Scinocca et al., 2008). A representation of gas phase chemistry has also been included that contains a relatively complete description of the HOx, NOx, Clx and Brx chemistry that controls stratospheric ozone along with the longer-lived source gases such as $CH_4$, $N_2O$ and CFCs (Jonsson et al., 2004). For the troposphere the chemical mechanism can be considered as methane-NOx chemistry as it does not include the chemistry of larger volatile organic compounds. The model does, however, include a description of associated tropospheric

chemical processes such as wet and dry deposition, interactive NOx emissions from lightning, corrections of clear-sky photolysis rates for clouds and $N_2O_5$ hydrolysis on prescribed sulfate aerosol distribution using the reaction probabilities of Davis et al. (2008). The simulation analysed here used a "Specified Dynamics" setup, one where the model horizontal winds and temperature are nudged towards a meteorological reanalysis dataset that represents the observed historical evolution of the atmosphere. In this way the day-to-day variability of the model meteorology is much more closely aligned with the historical

evolution of the atmosphere than would be possible in a free-running model. Here the CMAM model was nudged to six-hourly fields from the ERA-Interim reanalysis (Dee et al., 2011) on all model levels below 1 hPa and with a relaxation time constant of 24 hours.

## A5   DEHM

The Danish Eulerian Hemispheric Model (DEHM) (Christensen, 1997; Brandt et al., 2012; Massling et al., 2015) is a 3D

Eulerian atmospheric chemistry-transport model developed at the Department of Environmental Science at Aarhus University





in Denmark. The model domain covers the Northern Hemisphere using a polar stereo-graphic projection with a grid resolution of 150 km × 150 km. It includes nesting capabilities to make simulations with a higher grid resolution in a limited area of the domain, and in this work an Arctic sub-domain with 50 km × 50 km have been applied covering the Arctic area down to about 40-54°N. The Model have 29 vertical levels in sigma coordinates, where the lowest 15 levels are below 2000 m above the surface. The lowest model levels are 22 m thick and the top of the model domain is at 100 hPa. i.e the whole troposphere and very lowest part of the stratosphere. DEHM includes a SOx-NOx-VOC-ozone chemistry with 71 components including secondary organic aerosols (SOA), where VBS mechanism are used, and 9 particulates including hydrophobic and hydroscopic BC , primary organic aerosols, primary anthropogenic dust, $PM_{2.5}$ fraction and coarse fraction of $PM_{10}$ of seasalt and Pb. $CH_4$ are prognostic species, where the boundary conditions have large influence. The model is driven by meteorological data from a numerical weather prediction model from the WRF model (Skamarock et al., 2008), version 3.9, with 1-hour resolution. The WRF model system is driven by reanalysis data from the ERA-Interim made by ECMWF by nudging.

## A6  ECHAM-SALSA

ECHAM-SALSA is the general aerosol-climate model ECHAM-HAMMOZ (ECHAM6.3-HAM2.3-MOZ1.0) (Tegen et al., 2019; Schultz et al., 2018) using the Sectional Aerosol module for Large Scale Applications SALSA (Kokkola et al., 2018) to solve the aerosol microphysics. ECHAM6 (Stevens et al., 2013) computes the atmospheric circulation and fluxes using a semi-Lagrangian transport scheme. In the setup used here, the large-scale meteorology (vorticity, divergence, and surface pressure; relaxation times of 24, 6, and 48 h, respectively) was nudged towards ERA-Interim reanalysis data (Berrisford et al., 2011). In SALSA the aerosol size distribution is modelled using 10 size sections (or bins), which span particle sizes between 3 nm and 10 $\mu$m. The size distribution is further divided into a soluble and an insoluble sub-population, which are treated as externally mixed. Within one size bin of one sub-population, all aerosol particles are considered internally mixed. In its standard setup, SALSA describes the aerosol compounds, BC, organic carbon (OC), $SO_4^{2-}$, SS, and mineral dust (DU). In the model, BC, OC, SS, and DU are emitted as primary particles, while $SO_4^{2-}$ is emitted as either $SO_2$ or as DMS, which are oxidized using a simplified chemistry (Stier et al., 2005) to form $H_2SO_4$, which then either nucleates or condenses onto existing particles. BC, OC, and $SO_2$ emissions are prescribed using input files, while SS and DU emissions are computed online. All greenhouse gas concentrations are fixed to pre-defined concentrations. The model resolution for the simulations performed here was T63 (roughly 2° by 2°), further using 47 hybrid sigma-pressure levels.

## A7  EMEP MSC-W

The EMEP MSC-W model is a 3-D Eulerian chemistry transport model developed at the Norwegian Meteorological Institute within the Framework of the UN Convention on Long-range Transboundary Air Pollution. It is described in detail in Simpson et al. (2012). Although the model has traditionally been aimed at simulations of acidification, eutrophication and air quality over Europe, global modelling has been performed and evaluated against observations for many years (Jonson et al., 2010; Wild et al., 2012). The model uses 20 vertical levels defined as eta-hybrid coordinates. The 10 lowest levels are within the PBL (with the bottom layer being 92 m thick), and the top of the model domain is at 100 hPa. Model updates since Simpson et al. (2012),





resulting in EMEP model version rv4.33 as used here, have been described in Simpson et al. (2019) and references cited therein.
The main revisions were made to the parameterisations of coarse $NO_3^-$ formation on sea salt and dust aerosols, $N_2O_5$ hydrolysis
on aerosols, and additional gas-aerosol loss processes for $O_3$, $HNO_3^-$ and $HO_2$. The EMEP model, including a user guide, is
publicly available as Open Source code at https://github.com/metno/emep-ctm. EMEP-modelled $PM_{2.5}$ and $PM_{10}$ include
primary and secondary aerosols, both anthropogenic and natural. Secondary aerosol consists of inorganic sulfate, nitrate and
ammonium, and SOA; the latter is formed from both anthropogenic and biogenic emissions using the 'VBS' scheme detailed
in Bergström et al. (2012) and Simpson et al. (2012). The model also calculates sea salt aerosols and windblown dust particles
from soil erosion. AOD is calculated based on the mass concentrations of individual aerosols multiplied by corresponding
Mass Extinction Coefficients. In these simulations, we did not use the BC and OC emissions from EclipseV6b directly, but
applied EclipseV6b $PM_{2.5}$ and coarse PM emissions instead, which were split into elementary carbon (EC), organic matter
(OM) (here assumed inert) and the remaining inorganic dust. The EC and OM emissions in the fine and coarse fractions were
further divided into fossil fuel and wood-burning compounds for each country and source sector. The split applied to the PM
emissions is the same as used in EMEP operational runs (IIASA, personal communication). 80% of emitted EC is assumed to
be hydrophobic, ageing to become hydrophilic within 1 to 1.5 days. As in Bergström et al. (2012), the Organic Matter/Organic
Carbon ratio of emissions by mass is assumed to be 1.3 for fossil-fuel sources and 1.7 for wood-burning sources. Note that
different wildfire emissions were used here, i.e. from FINN (the Fire INventory from NCAR, version 15). The EMEP model
runs were driven by 3-hourly meteorological data from the ECMWF IFS model at $0.5° \times 0.5°$ resolution.

## A8  FLEXPART

The Lagrangian particle dispersion model FLEXPART version 10.4 (Pisso et al., 2019) releases computational particles that are
simulated forward in time following 3-hourly ECMWF meteorological fields with 137 vertical layers and a spatial resolution
of 1° x 1°. For each year around 330 million particles were released to calculate turbulent diffusion (Cassiani et al., 2014),
unresolved mesoscale motions (Stohl et al., 2005) and convection (Forster et al., 2007). A recently updated wet-deposition
scheme taking into account in-cloud and below cloud removal was used (Grythe et al., 2017). Gravitational settling for spherical
BC particles with an aerosol mean diameter of 0.25 $\mu$m and a normalised standard deviation of 3.3 and a particle density of
1500 $kg\,m^{-3}$ (Long et al., 2013) are used in the calculation of dry deposition. The surface concentration and deposition fields
were retrieved on a monthly basis on a resolution of $0.5° \times 0.5°$.

## A9  GEM-MACH

GEM-MACH (Global Environmental Multiscale model-Modelling Air quality and CHemistry) is the Environment and Climate
Change Canada (ECCC) air quality prediction model. It consists of an online tropospheric chemistry module embedded within
ECCC's GEM numerical weather forecast model (Côté et al., 1998b,a; Charron et al., 2012). The chemistry module includes
a comprehensive representation of air quality processes, such as gas-phase, aqueous-phase, and heterogeneous chemistry and
aerosol processes (e.g. Moran et al. (2013); Makar et al. (2015b,a); Gong et al. (2015). Specifically, gas-phase chemistry is rep-
resented by a modified ADOM-II mechanism with 47 species and 114 reactions (Lurmann et al., 1986; EPRI, 1989); inorganic



heterogeneous chemistry is parameterized by a modified version of the ISORROPIA algorithm of Nenes et al. (1999), as described in detail in Makar et al. (2003); SOA formation is parameterized using a two-product, overall or instantaneous aerosol yield formation (Odum et al., 1996; Jiang, 2003; Stroud et al., 2018); aerosol microphysical processes, including nucleation
and condensation (sulfate and SOA), hygroscopic growth, coagulation and dry deposition/sedimentation are parameterized as in Gong et al. (2003); the representation of cloud processing of gases and aerosols includes uptake and activation, aqueous phase chemistry, and wet removal (Gong et al., 2006, 2015).

Aerosol chemical composition is represented by eight components: sulfate, nitrate, ammonium, elemental carbon (EC), primary organic aerosol (POA), secondary organic aerosol (SOA), crustal material (CM) and sea salt; aerosol particles are
assumed to be internally mixed. A sectional approach is used for representing aerosol size distribution. For the 2015 Arctic simulation, a 12-bin (between 0.01 and 40.96 $\mu$m in diameter, logarithmically spaced: 0.01-0.02, 0.02-0.04, 0.04-0.08, 0.08-0.16, 0.16-0.32, 0.32-0.64, 0.64-1.28, 1.28-2.56, 2.56-5.12, 5.12-10.24, 10.24-20.48 and 20.48-40.96 $\mu$m) configuration is used.

Type 0 simulation was conducted for the year of 2015 over a limited-area model (LAM) domain on a rotated lat-lon grid at
$0.1375° \times 0.1375°$ (or 15-km) resolution covering the Arctic (>60°N) and extending to the southern US-Canada border. Some of the model upgrades for the Arctic simulation are described in Gong et al. (2018). Anthropogenic emissions used are based on a combination of North American emission inventories (specifically, the 2016 US National Emission Inventories and 2015 Canadian national Air Pollution Emission Inventories) and global ECLIPSE v6b 2015 baseline emissions. North American wildfire emissions are processed using the Canadian Forest Fire Emission Prediction System (CFFEPS) from satellite detected
fire hotspot data (MODIS, AVHRR, and VIIRS). CFFEPS consists of a fire growth model, a fire emissions model, and a thermodynamic-based model to predict the vertical penetration height of a smoke plume from fire energy (see Chen et al. (2019) for details). Biogenic emissions are calculated online in GEM-MACH based on the algorithm from BEIS version 3.09 with BELD3-format vegetation land cover. Sea salt emissions are computed based on Gong et al. (2003).

The chemical lateral boundary conditions were from MOZART-4/GEOS-5 (https://www.acom.ucar.edu/wrf-chem/mozart.shtml;
Emmons et al. (2010). The meteorology was initialized daily (at 00:00 UTC) using the Canadian Meteorological Centre's global objective analyses.

**A10   GEOS-Chem**

GEOS-Chem is a global three-dimensional chemical transport model driven by assimilated meteorological observation from the Goddard Earth Observing System (GEOS) of the NASA Data Assimilation Office (DAO) which was first introduced in 2001
(Bey et al., 2001). GEOS-Chem is a grid-independent model which operates on a 1-D column with default or user-specified horizontal grid points, vertical grid points, and timestep. GEOS-Chem Classic can use archived GEOS meteorological data on a rectilinear latitude-longitude grid to compute horizontal and vertical transport and use Open-MP in parallelization. Two of the assimilated meteorological data from the NASA Global Modeling and Assimilation Office (GMAO) can be used to drive the off-line mode of GEOS-Chem. The first one is the operational data starting from 2012, the GEOS Forward Processing
(GEOS-FP, Lucchesi (2013) which native resolution was $0.25° \times 0.3125°$. The second one is the consistent Modern-Era



Retrospective Analysis for Research and Applications version 2 (MERRA-2, Randles et al. (2017) starting from 1979-present, with the native resolution $0.5° \times 0.625°$. Both meteorological data have 72 hybrid sigma-pressure levels with the top at 0.01 hPa and 3-hourly temporal resolution for 3-D fields and 1-hour resolution for 2-D fields. The advection scheme of GEOS-Chem uses the TPCORE advection scheme (Lin and Rood, 1996) on the latitude-longitude grid, while the convective transport

uses the convective mass flux described by Wu et al. (2007). The wet deposition scheme in GEOS-Chem is based on Liu et al. (2001) for water soluble aerosols and Amos et al. (2012) for gases. The dry deposition is based on the resistance-in-series scheme of Wesely (1989). Aerosol deposition is from Zhang et al. (2001). Emission of dust aerosol, lightning NOx, biogenic VOCs, soil $NO_x$, and sea salt aerosol are dependent on the local meteorological conditions. CEDS global inventory is the default anthropogenic emissions, while EDGAR v4.3.2 (M. et al., 2018) is also available as an alternate option to CEDS.

Future anthropogenic emissions following the RCP scenarios (Holmes et al., 2013), aircraft emissions (Stettler et al., 2011), ships emission (from CEDS), lighting $NO_x$ emissions (Murray et al., 2012) are also include and configured at run-time using the HEMCO module described (Keller et al., 2014). Biogenic VOC emissions in GEOS-Chem are from the MEGAN v2.1 inventory (Guenther et al., 2012). Chemical solver in the standard GEOS-Chem simulation uses KPP (Damian et al., 2002) as implemented in GEOS-Chem. The gas phase in troposphere in GEOS-Chem included detailed HOx-NOx-VOC-ozone-

halogen-aerosol tropospheric chemistry mechanism which generally follows JPL/IUPAC recommendations including the PAN (Fischer et al., 2014), Isoprene (Travis et al., 2016; Fisher et al., 2016), Halogenes (Sherwen et al., 2016; Chen et al., 2017) and Criegees (Millet et al., 2015). A linearized stratospheric chemistry scheme has been implemented since GEOS-Chem v9.0. The model will read from an archived 3D monthly mean production rates and losing frequency for each species at the beginning of each month. The Linoz chemistry (McLinden et al., 2000) is also applied and as a recommended option for stratospheric

ozone layer. The original sulfate-nitrate-ammonium aerosol simulation in GEOS-Chem coupled to gas-phase chemistry (Park et al., 2004). The black carbon simulation (Wang et al., 2014), organic aerosol (J. et al., 2020), complex SOA (Pye et al., 2010), the aqueous-phase isoprene SOA scheme (Marais et al., 2016) and the dust simulation (Duncan Fairlie et al., 2007) are also implement into GEOS-Chem. The dust size distributions are from L. et al. (2013). The GEOS-Chem v12.3.2 with uniformly $2° \times 2.5°$ MERRA-2 meteorological data for 2008-9, GEOS-FP meteorological data for 2014-15, and ECLIPSEv6b emissions

was used in this study.

## A11 GISS-E2.1

NASA Goddard Institute of Space Studies (GISS) Earth system model (ESM), GISS-E2.1, is a fully-coupled ESM. A full description of GISS-E2.1 and evaluation of its coupled climatology during the satellite era (1979-2014) and the historical ensemble simulation of the atmosphere and ocean component models (1850-2014) are described in Kelley et al. (2020). GISS-

E2.1 has a horizontal resolution of $2°$ in latitude by $2.5°$ in longitude, and 40 vertical layers extending from the surface to 0.1 hPa in the lower mesosphere. The tropospheric chemistry scheme used in GISS-E2.1 (Shindell et al., 2001, 2003) includes inorganic chemistry of Ox, NOx, HOx, CO, and organic chemistry of CH4 and higher hydrocarbons using the CBM4 scheme (Gery et al., 1989), and the stratospheric chemistry scheme (Shindell et al., 2006) which includes chlorine and bromine chemistry together with polar stratospheric clouds. The meteorology was nudged to the NCEP reanalysis.





In the present work, we used the OMA, the One-Moment Aerosol scheme (Bauer et al., 2007a,b; Bauer and Koch, 2005; Koch et al., 2006; Miller et al., 2006; Tsigaridis et al., 2013; Bauer et al., 2020). OMA is a mass-based scheme in which aerosols are assumed to remain externally mixed and have a prescribed and constant size distribution, with the exception of sea salt that has two distinct size classes, and dust that is described by a sectional model with an option from 4 to 6 bins. The OMA scheme treats sulfate, nitrate, ammonium, carbonaceous aerosols (black carbon and organic carbon, including the

NOx-dependent formation of SOA and methanesulfonic acid formation), dust and sea-salt. The model includes secondary organic aerosol production, as described by (Tsigaridis and Kanakidou, 2007). The default dust configuration that is used in this work includes 5 bins, a clay and 4 silt ones, from submicron to 16 $\mu$m in size. The first three dust size bins can be coated by sulfate and nitrate aerosols (Bauer and Koch, 2005). OMA only includes the first aerosol indirect effect. The aerosol number concentration that impacts clouds are obtained from the aerosol mass as described in (Menon and Rotstayn, 2006).

The natural emissions of sea salt, DMS, isoprene and dust are calculated interactively. Anthropogenic dust sources are not represented in ModelE2.1. Dust emissions vary spatially and temporally only with the evolution of climate variables like wind speed and soil moisture (Miller et al., 2006). The version of the model we use in this work uses prescribed sea surface temperature (SST) and sea ice thickness and extent during the historical period (Rayner et al., 2003).

## A12   MATCH

MATCH - Multiscale Atmospheric Transport and Chemistry (Robertson et al., 1999) is an offline, Eulerian, 3-D chemistry transport model developed at the Swedish Meteorological and Hydrological Institute. MATCH can be run on global to urban domains to study a range of atmospheric chemistry/air quality problems, but for this study model runs were performed for the  20°N-90°N region focusing on long-transport to the Arctic. ERA-Interim reanalysis data from the European Centre for Medium-Range Weather Forecasts (ECMWF) were used as meteorological input to the model. Six-hourly data (3-hourly for

precipitation) were extracted from the ECMWF archives on a 0.75° × 0.75° rotated latitude-longitude grid. The original data had 60 levels, but the 38 lowest levels reaching about 16 km in the Arctic were used in the model.

The scheme for gas-phase tropospheric chemistry and bulk aerosols as described in Andersson et al. (2007) was used. Methane concentrations were prescribed. Boundary conditions at the top of the model and at the lateral boundaries for a range of species including ozone were based on monthly mean values from the Copernicus Atmospheric Monitoring Service. The

aerosol scheme was extended with BC and OC simulated as two fractions: fresh, hydrophobic and aged, and hydrophilic. Eighty percent of anthropogenic emissions from all sectors were emitted into the hydrophobic and 20% into the hydrophilic fraction except for fire/biomass combustion where 100% was emitted into the hydrophilic component following Genberg et al. (2013). Scavenging and aging was parameterized following Liu et al. (2011), i.e., aging is proportional to OH and scavenging in mixed-phase clouds is reduced. The hydrophobic fraction is assumed to be 5% activated in the scavenging scheme, while

the hydrophilic fraction is 100% activated. If the clouds are mixed phase, then the scavenging efficiency is scaled by the ratio of cloud ice water content to total cloud water content assuming zero scavenging for 100% ice clouds.





## A13  MATCH-SALSA-RCA4

The chemistry transport model, MATCH (Robertson et al., 1999; Andersson et al., 2007) described above is online coupled to the aerosol dynamics model, SALSA (Kokkola et al., 2008). SALSA describes the whole chain from nucleation to the growth
and deposition of particles and computes the size distribution, number concentration and chemical composition of the aerosol species. A sectional representation of the aerosol size distribution is considered with three main size ranges (a: 3-50 nm, b: 50-700 nm and c: > 700 nm) and each range is again subdivided into smaller bins and into soluble and insoluble bins adding up to a total of 20 bins. A schematic of the sectional size distribution and the aerosol species considered in each bin is shown in Figure A.1.13.1. The seasonally varying emissions are based on the sector-wise ECLIPSE inventory. Isoprene emissions are modelled
online depending on the meteorology based on the methodology by Simpson et al. (1995). The terpene emissions ($\alpha$-pinene) are taken from the modelled fields by the EMEP model. Sea salt is parameterized following the scheme of Foltescu et al. (2005) but modified for varying particle sizes, wherein Mårtensson et al. (2003) scheme is used if the particle diameter is 1 $\mu$m and Monahan et al. (1986) scheme is used otherwise. The coupling of MATCH with SALSA and the evaluation of this model setup is described in detail in Andersson et al. (2015). A cloud activation model that computes 3-D CDNCs (Cloud Droplet Number
Concentrations) based on the prognostic parameterization scheme of Abdul-Razzak and Ghan (2002) specifically designed for aerosol representation with sectional bins is embedded in the MATCH-SALSA model. This scheme simulates the efficiency of an aerosol particle to be converted to a cloud droplet depending on the number concentration and chemical composition of the particles given the updraft velocity and supersaturation of the air parcel. The updraft velocity is computed as the sum of the grid mean vertical velocity and turbulent kinetic energy (TKE) for stratiform clouds (Lohmann et al., 1999). These CDNCs are
then offline coupled to a regional climate model, RCA4 (SAMUELSSON et al., 2011), that provides us information on cloud properties such as cloud cover, cloud droplet radii, cloud liquid-water path as well as radiative fluxes. The schematic of the model coupling is shown in Figure A.1.13.2. RCA4 is run with 6-hourly ERA-Interim meteorology and the 3-hourly RCA4 meteorological fields along with the fields needed to calculate updraft velocity are used to drive the MATCH-SALSA-cloud activation model. The CDNCs are then used to re-run the RCA4 model to obtain the cloud properties and radiative effects. The
validation and more details of this model set up is described in Thomas et al. (2015).

## A14  MRI-ESM2

MRI-ESM2 (Meteorological Research Institute (MRI) Earth System Model version 2.0, developed by the MRI of the Japan Meteorological Agency) consists of four major component models; an atmospheric general circulation model (MRI-AGCM3.5) with land processes, an ocean-sea-ice general circulation model (MRI.COMv4), and aerosol and atmospheric chemistry mod-
els (YUKIMOTO et al., 2019; Kawai et al., 2019; Oshima et al., 2020), however, we do not couple OGCM in this study's simulations. MRI-ESM2 uses different horizontal resolutions but employs the same vertical resolution in each atmospheric component model as follows: TL159 (approximately 120 km), TL95 (approximately 180 km), and T42 (approximately 280 km) in the MRI-AGCM3.5, the aerosol model, and the atmospheric chemistry model, respectively, all with 80 vertical layers (from the surface to a model top of 0.01 hPa) in a hybrid sigma-pressure coordinate system. Each component model is in-




teractively coupled by a coupler, which enables an explicit representation of the effects of gases and aerosols on the climate system. The atmospheric chemistry component model in MRI-ESM2 is the MRI Chemistry Climate Model version 2.1 (MRI-CCM2.1), which calculates the evolution and distribution of ozone and other trace gases in the troposphere and in the middle atmosphere. The model calculates a total of 90 gas-phase chemical species and 259 chemical reactions in the atmosphere. The aerosol component model in MRI-ESM2 is the Model of Aerosol Species in the Global Atmosphere mark-2 revision 4-climate

(MASINGAR mk-2r4c) that calculates atmospheric aerosol physical and chemical processes and treats the following species; nonsea-salt sulfate, BC, OC, sea salt, mineral dust, and aerosol precursor gases (e.g., sulfur dioxide and dimethyl sulfide). The size distributions of sea salt and mineral dust are divided into 10 discrete bins and the sizes of the other aerosols are represented by lognormal size distributions. The model assumes external mixing for all aerosol species; however, in the radiation process in MRI-AGCM3.5, hydrophilic BC is assumed to be internally mixed with sulfate with a shell-to-core volume

ratio of 2; the optical properties of hydrophilic BC are calculated based on Mie theory with a core-shell aerosol treatment, in which a concentric BC core is surrounded by a uniform coating shell composed of other aerosol compounds (Oshima et al., 2009b,a). MRI-ESM2 employs a BC aging parameterization (Oshima and Koike, 2013) that calculates the variable conversion rate of BC from hydrophobic BC to hydrophilic BC, which generally depends on the production rate of condensable materials such as sulfate. In the radiation and cloud processes in MRI-ESM2, sulfate is assumed to be $(NH_4^+)_2SO_4^{2-}$ and OC

is assumed to be organic matter (OM) by lumping OC species using an OM-to-OC factor of 1.4. MRI-ESM2 represents the activation of aerosols into cloud droplets based on the parameterizations, and detailed descriptions and evaluations of the cloud processes and cloud representations in MRI-ESM2 are given by Kawai et al. (2019). Evaluations of the effective radiative forcing (ERF) of anthropogenic gases and aerosols in present-day conditions relative to preindustrial conditions in the global and the Arctic using MRI-ESM2 are given by Oshima et al. (2020). The simulations in this study were performed from Jan-

uary 2008 (or January 1990) to December 2015 after a 1-year spin-up run using the prescribed SST and sea ice data (provided by the AMIP experiment in CMIP6, https://www.wcrp-climate.org/modelling-wgcm-mip-catalogue/modelling-wgcm-mips-2/240-modelling-wgcm-catalogue-amip). The horizontal wind fields were nudged toward the 6-hourly Japanese 55-year Re-analysis (JRA55) data (Kobayashi et al., 2015) (https://jra.kishou.go.jp/JRA-55/index_en.html) in the simulation. We used the monthly anthropogenic emissions from the ECLIPSE V6B emission dataset and the monthly biomass burning emissions from

the CMIP6 in the simulations. Major volcanic aerosols are given by the stratospheric aerosol dataset used in the CMIP6 experiments (Thomason et al., 2018). A second simulation with volcanic $SO_2$ emission including Holuhraun eruption was also performed for 2014-2015.

**A15  NorESM1**

NorESM1 (Bentsen et al., 2013; Iversen et al., 2013) is based on the fourth version of the Community Climate System Model

(CCSM4) (Gent et al., 2011), with coupled models for the atmosphere, ocean, land and sea-ice. Here, we have used a 1° horizontal resolution in the atmosphere (0.95° latitude by 1.25° longitude, version 'NorESM1-Happi'). The model has 26 vertical levels on a hybrid sigma-pressure co-ordinate up to the model top at 2.194 hPa. The model calculates the lifecycles of a range of natural and anthropogenic aerosol components from emissions and physico-chemical processing in air and cloud





droplets. The only prescribed aerosol concentrations are stratospheric sulfate from explosive volcanoes. The direct and indirect

aerosol effects on climate are calculated by parameterization of aerosol interactions with schemes for radiation and warm cloud microphysics (Kirkevåg et al., 2013). The model uses a prognostic calculation of cloud droplet numbers, allowing for competition effects between aerosols of different hygroscopic property and size.

### A16   OsloCTM

The Oslo CTM3 is an offline global three-dimensional chemistry transport model driven by 3-hourly meteorological fore-

cast data from the Integrated Forecast System (IFS) model at the European Centre for Medium-Range Weather Forecasts (ECMWF). The Oslo CTM3 consists of a tropospheric and stratospheric chemistry scheme (Søvde et al., 2012) as well as aerosol modules for sulfate, nitrate, black carbon, primary organic carbon, secondary organic aerosols, mineral dust and sea salt (Lund et al., 2018a).

### A17   UKESM1

UKESM1 (United Kingdom Earth System Model) is a fully-coupled Earth System model (Sellar et al., 2019) with a coupled atmosphere ocean physical climate model (HadGEM3-GC3.1) at its core (Kuhlbrodt et al., 2018; Williams et al., 2018). For UKESM1 various Earth system components are incorporated with the physical climate model including ocean biogeochemistry, an interactive stratosphere-troposphere chemistry and aerosol scheme and terrestrial carbon and nitrogen cycles coupled to interactive vegetation. The model has a horizontal resolution of  135 km at the mid-latitudes ($1.875° \times 1.25°$), with 85

levels on a terrain-following hybrid height coordinate system, ranging in height from the surface to a model top of 85 km. The combined stratosphere-troposphere United Kingdom Chemistry and Aerosol (UKCA) scheme is used within UKESM1 and is fully described and evaluated in (Archibald et al., 2020; Mulcahy et al., 2020).

The chemical scheme in UKCA is built upon the scheme described for the stratosphere in Morgenstern et al. (2009) and that for the troposphere described in O'Connor et al. (2014). Chemical reactions are included within UKCA for odd-oxygen

(Ox), nitrogen (NOy), hydrogen (HOx = OH + HO$_2$), CO, CH$_4$ and short-chain non-methane volatile organic compounds (NMVOCs), including isoprene. Reactions involving NMVOCs are simulated as discrete species. UKCA includes an interactive photolysis scheme, as well as representations of both wet and dry deposition for gas and aerosol species. Additional chemical reactions for DMS, SO$_2$ and monoterpenes (C$_{10}$H$_{16}$) are included to enable coupling to the aerosol scheme within UKCA. A two-moment aerosol microphysical scheme, GLOMAP (Global Model of Aerosol Processes; Mann et al. (2010, 2012)),

is used to simulate four aerosol components (SO$_3$, BC, organic matter, sea-salt) across five log-normal modes, ranging from sub to super micron sizes. Mineral dust is simulated separately using a 6 bin mass only scheme, ranging in size from 0.6 to 60 microns in diameter (Woodward, 2001). Ammonium nitrate is not currently included within the UKCA aerosol scheme. The formation of secondary organic aerosols (SOA) is included based on a fixed yield rate of 26% from the products of monoterpene oxidation. The higher fixed yield value accounts for the underlying uncertainty in SOA formation and the absence

of anthropogenic, marine and isoprene sources.





Precursor emission fluxes are either prescribed using specified input files or calculated interactively using online meteo-rological variables within UKESM1. Methane is represented by using prescribed global concentrations. Interactive emission fluxes are calculated online for sea salt, DMS, dust, lightning NOx and biogenic volatile organic compounds (BVOCs). Emissions of isoprene and monoterpenes from the natural environment are calculated online by coupling to the land surface scheme within UKESM1. Simulations provided by UKESM1 and used in the AMAP assessment have been undertaken using different configurations. For this study, experiments UKESM1 has been set up using an atmosphere only configuration that is nudged to ECMWF reanalysis (ERA-interim) of temperature and wind fields above the boundary layer. Prescribed values of sea surface temperatures and sea ice are used for each year of simulation based on historical simulations conducted as part of CMIP6 using the fully coupled atmosphere-ocean configuration of UKESM1. For other ancillary inputs a multi-year climatology was used; equivalent to an AMIP type simulation.

## A18   WRF-Chem

WRF-Chem (Weather Research and Forecasting model with online coupled chemistry) is used to simulate the transport and chemical transformation of trace gases and aerosols simultaneously with the meteorology. The model dynamics (WRF) are non-hydrostatic. The model version used for AMAP is WRF-Chem version 3.8.1 also including updates reported in Marelle et al. (2017) and Marelle et al. (2018). The simulation was performed on a polar stereographic projection with a horizontal resolution of 100 km and 50 vertical hybrid terrain-following vertical pressure levels using hydrostatic pressure. The center of the domain is placed at the North pole and the latitude at the domain's outside boundary varies from 7° South to 7° North. The WRF-Chem chemical lateral boundary conditions are from MOZART-4/GEOS-5 (https://www.acom.ucar.edu/wrf-chem/mozart.shtml; Emmons et al. (2010). Pressure at the model top is set to 50 hPa with stratospheric concentrations (e.g. ozone) taken from climatologies. The model was run with Morrison double-moment scheme microphysics, long and short wave radiative effects treated by RRTMG scheme, Kain-Fritsch-Cumulus Potential (KF-CuP) cumulus parameterization scheme. The model was run with SAPRC-99 chemical scheme providing gas-phase tropospheric reactions including VOCs and NOx, coupled with the MOSAIC 8 bin sectional scheme including VBS treatments for SOA. Methane concentrations are prescribed. Stratospheric or tropospheric halogen chemistry is not included. It was run using anthropogenic emissions from ECLIPSE v6b and the GFED fire emissions. Boundary and initial meteorological conditions were given by the global NCEP Final Analysis (FNL) and used to nudge the temperature, relative humidity, and winds at every dynamical time-step above the planetary boundary layer.

## Appendix B: Observational Datasets

The following datasets were used to evaluate the models in this study. As BC measurements vary by instrument, Table B1 summarizes the different Arctic BC datasets used in this study.

For SLCFs other than Arctic BC, Table B2 summarizes some information about the observation networks.



*Author contributions.* CHW organized the model-measurement comparisons in the AMAP SLCF expert group, led the trace gas model evaluations, and wrote the manuscript with contributions from her co-authors. RM developed and ran the aerosols model-measurement comparison scripts, including the ship-based BC measurements and did the aerosols analysis; KvS led the AMAP SLCF modelling strategy and developed and ran the CanAM5-PAM model; SE and NE provided FLEXPART model output, and SE did the model-measurement comparisons for BC and $SO_4^{2-}$ deposition. DW-P developed the CIS tool for BC aircraft model-measurement comparisons, and BW used the tool and did the model-aircraft comparisons and analysis. LS did the ACE-FTS trace gas model comparisons, and KW provided guidance on ACE-FTS data quality, usage and model measurement comparisons. MF co-led the AMAP model strategy and ran the CESM model. LP provided additional CESM model runs. DP developed and ran the CMAM model. YP provided CIESM-MAM7 model output; JC provided DEHM model output, TK provided ECHAM-SALSA model output, ST and MG provided EMEP-MSC-W model output and some analysis. UI, GF, and KT provided the GISS-E2.1 model output; WG and SB provided the GEM-MACH model output; JF, R-YC, and XD provided the GEOS-Chem model output; JL provided the MATCH model output; MT provided the MATCH-SALSA model output; NO provided the MRI-ESM2 model output; MS and SK provided the NorESM model output; RS provided the OsloCTM model output; SA and ST provided UKESM1 model output; and KL, J-CR, TO, and LM provided WRF-Chem model output. SS and LH provided Alert datasets; OP provided Russian cruise ship measurements; FT and YK provided the Japanese ship measurements; HS and AM provided Villum Research Station datasets; JS provided aerosol datasets and expertise.

*Competing interests.* No competing interests are present.

*Acknowledgements.* The work reflected in this publication was produced with the financial support of the Arctic Monitoring and Assessment Programme (AMAP). The authors would like to thank all operators and technicians at the Arctic stations for the collection of observational data. Thanks are also due to the following people for providing us with data: Mauro Mazzola, Stefania Gilardoni, and Angelo Lupi from the Institute of Polar Sciences for eBC measurements at Gruvebadet lab; and Mirko Severi from University of Florence for $SO_4^{2-}$ measurements at Gruvabadet. Assessments of Russian ship based campaign was performed according to the Development program of the Interdisciplinary Scientific and Educational School of M.V.Lomonosov Moscow State University "Future Planet and Global Environmental Change", development of the methodology for aethalometric data treatment was supported by RSF project #19 -77-30004. The BC observations on R/V Mirai were supported by the Ministry of Education, Culture, Sports, Science and Technology (MEXT), Japan (Arctic Challenge for Sustainability (ArCS) project). Fairbanks aerosol measurements came from William Simpson. For Villum data, we acknowledge the Aarhus University, Department of Environmental Science (ENVS). NOAA/ESRL/GMD, EMEP (http://ebas.nilu.no) and WMO GAW network are acknowledged for Barrow and Zeppelin observational datasets. Contributions by SMHI were funded by the Swedish Environmental Protection Agency under contract NV-03174-20 and Swedish Climate and Clean Air Research program (SCAC) and partly by Swedish National Space Board (NORD-SLCP, Grant agreement ID: 94/16) and EU Horizon 2020 project Integrated Arctic Observing System (INTAROS, Grant agreement ID: 727890. The ACE-FTS is a Canadian-led missiong mainly supported by Canadian Space Agency (CSA). Work on ACE-FTS analysis was supported by the Natural Sciences and Engineering Research Council of Canada (NSERC). IMPROVE is a collaborative association of state, tribal, and federal agencies, and international partners. US Environmental Protection Agency is the primary funding source, with contracting and research support from the National Park Service. The Air Quality Group at the University of California, Davis is the central analytical laboratory, with ion analysis provided by Research Triangle Institute, and carbon analysis provided by Desert Research Institute. JS holds the





Ingvar Kamprad Chair for Extreme Environments Research sponsored by Ferring Pharmeceuticals. JS acknowledges funding from the Swiss National Science Foundation (project no. 200021_188478). DWP acknowledges funding from NERC projects NE/P013406/1 (A-CURE) and NE/S005390/1 (ACRUISE) as well as funding from the European Union's Horizon 2020 research and innovation programme iMIRACLI under Marie Skłodowska-Curie grant agreement No 860100. $PM_{2.5}$ observation from the US embassy in China belongs to the U.S. Department of State and is not fully verified or validated and could be subject to changes/corrections/errors. LATMOS acknowledges support from the EU iCUPE (Integrating and Comprehensive Understanding on Polar Environments) project (grant agreement n°689443), under the European Network for Observing our Changing Planet (ERA-Planet), and from access to IDRIS HPC resources (GENCI allocation A009017141) and the IPSL mesoscale computing centre (CICLAD: Calcul Intensif pour le CLimat, l'Atmosphère et la Dynamique) for model simulations. NO was supported by the Japan Society for the Promotion of Science KAKENHI (grant numbers: JP18H03363, JP18H05292, and JP21H03582), the Environment Research and Technology Development Fund (JPMEERF20202003 and JPMEERF20205001) of the Environmental Restoration and Conservation Agency of Japan, the Arctic Challenge for Sustainability II (ArCS II), Program Grant Number JPMXD1420318865, and a grant for the Global Environmental Research Coordination System from the Ministry of the Environment, Japan (MLIT1753). The research with GISS-E2.1 has been supported by the Aarhus University Interdisciplinary Centre for Climate Change (iClimate) OH fund (no. 2020-0162731), the FREYA project funded by the Nordic Council of Ministers (grant agreement nos. MST-227-00036 and MFVM-2019-13476), and the EVAM-SLCF funded by the Danish Environmental Agency (grant agreement no. MST-112-00298). JC (for DEHM model) acknowledges Danish Environmental Protection Agency (DANCEA funds for Environmental Support to the Arctic Region project; grant no. 2019-7975). The ECHAM-HAMMOZ model is developed by a consortium composed of ETH Zurich, Max Planck Institut für Meteorologie, Forschungszentrum Jülich, University of Oxford, the Finnish Meteorological Institute and the Leibniz Institute for Tropospheric Research, and managed by the Center for Climate Systems Modeling (C2SM) at ETH Zurich.



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

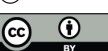

**Table 1.** Summary of models used in this study. *GCM=global climate model, CCM=chemistry climate model, ESM=Earth system model, CTM=chemical transport model.

| Name | type | simulation period | meteorology | SLCF output | primary reference(s) |
|---|---|---|---|---|---|
| CanAM5-PAM | GCM | 1990-2015 | nudged to ERA-Interim reanalysis | BC, $SO_4^{2-}$, OA, $PM_{2.5}$, AOD, AAOD, AE | von Salzen et al. (2000); von Salzen et al. (2013) |
| CESM2.0 | ESM | 2008-9, 2014-15 | free-running | $O_3$, CO, $NO_2$, BC, $SO_4^{2-}$, OA, $PM_{2.5}$, AOD, AAOD, AE | Ma et al. (2008); Peng et al. (2012); Mahmood et al. (2016, 2019); Danabasoglu et al. (2020); Liu et al. (2016) |
| CIESM-MAM7 | GCM | 1990-2015 | nudged to ERA-Interim reanalysis | BC, $SO_4^{2-}$, OA, $PM_{2.5}$, AOD | Lin et al. (2020); Liu et al. (2012) |
| CMAM | CCM | 1990-2015 | nudged to ERA-Interim reanalysis | $O_3$, CO, $NO_2$, $CH_4$ | Jonsson et al. (2004); Scinocca et al. (2008) |
| DEHM | CTM | 1990-2015 | nudged to ERA-Interim reanalysis | $O_3$, CO, $NO_2$, $CH_4$, BC, $SO_4^{2-}$, OA, $PM_{2.5}$, AOD, AAOD, AE | Christensen (1997); Brandt et al. (2012); Massling et al. (2015) |
| ECHAM6-SALSA | GCM | 2008-9, 2014-15 | nudged to ERA-Interim reanalysis | BC, $SO_4^{2-}$, OA, $PM_{2.5}$, AOD, AAOD, AE | Tegen et al. (2019); Schultz et al. (2018); Kokkola et al. (2018) |
| EMEP MSC-W | CTM | 1990-2015 | driven by 3-hrly ECMWF met | $O_3$, CO, $NO_2$, $CH_4$, BC, $SO_4^{2-}$, OA, $PM_{2.5}$, AOD | Simpson et al. (2012, 2019) |
| FLEXPART | Lagrangian CTM | 2014-2015 | driven by 3-hrly ECMWF met | BC, $SO_4^{2-}$ | Pisso et al. (2019) |
| GEM-MACH | online CTM | 2015 | driven by GEM numerical forecast | $O_3$, CO, $NO_2$, BC, $SO_4^{2-}$, OA, $PM_{2.5}$ | Moran et al. (2013); Makar et al. (2015b,a); Gong et al. (2015) |
| GEOS-Chem | CTM | 2008-9, 2014-15 | Driven by GEOS meteorology | $O_3$, CO, $NO_2$, $CH_4$, BC, $SO_4^{2-}$, OA, $PM_{2.5}$, AOD, AAOD, AE | Bey et al. (2001) |
| GISS-E2.1 | ESM | 1990-2015 | nudged to NCEP reanalysis | $O_3$, CO, $NO_2$, $CH_4$, BC, $SO_4^{2-}$, OA, $PM_{2.5}$, AOD, AAOD, AE | Kelley et al. (2020); Miller et al. (2021); Bauer et al. (2020) |
| MATCH | CTM | 2008-9, 2014-15 | ERA-Interim reanalysis | $O_3$, CO, $NO_2$, BC, $SO_4^{2-}$, OA, $PM_{2.5}$, AOD, AAOD, AE | Robertson et al. (1999) |
| MATCH-SALSA-RCA4 | CCM | 2008-9, 2014-15 | RCA4 | $O_3$, CO, $NO_2$, BC, $SO_4^{2-}$, OA, $PM_{2.5}$, AOD, AAOD, AE | Robertson et al. (1999); Andersson et al. (2007); Kokkola et al. (2008) |
| MRI-ESM2 | ESM | 1990-2015 | nudged to JRA55 reanalysis | $O_3$, CO, $NO_2$, $CH_4$, BC, $SO_4^{2-}$, OA, $PM_{2.5}$, AOD, AAOD | YUKIMOTO et al. (2019); Kawai et al. (2019); Oshima et al. (2020) |
| NorESM1-happi | ESM | 2008-9, 2014-15 | free running | BC, $SO_4^{2-}$, OA, AOD, AAOD, AE | Bentsen et al. (2013); Iversen et al. (2013); Gent et al. (2011); Graff et al. (2019) |
| Oslo-CTM | CTM | 2008-9, 2014-15 | driven by 3-hrly ECMWF meteorology | $O_3$, CO, $NO_2$, $CH_4$, BC, $SO_4^{2-}$, OA, $PM_{2.5}$ | Søvde et al. (2012); Lund et al. (2018a) |
| UKESM1 | CCM & ESM | 1990-2015 | nudged to ERA-Interim reanalysis | $O_3$, CO, $NO_2$, $CH_4$, BC, $SO_4^{2-}$, OA, $PM_{2.5}$, AOD, AAOD, AE | Sellar et al. (2019); Kuhlbrodt et al. (2018); Williams et al. (2018) |
| WRF-Chem | CCM & CTM | 2014-15 | nudged to NCEP FNL reanalysis | $O_3$, CO, $NO_2$, BC, $SO_4^{2-}$, OA, $PM_{2.5}$, AOD, AAOD, AE | Marelle et al. (2017, 2018) |





**Table 2.** Summary of emissions used in the models

| Model | biogenic | volcanic | forest fire | agricultural waste burning |
|---|---|---|---|---|
| CanAM5-PAM | none | specified climatological emissions and CMIP6 stratospheric aerosol | CMIP6 | ECLIPSEv6b |
| CESM2.0 | MEGANv2.1 | CMIP6 | CMIP6 | ECLIPSEv6b |
| CIESM-MAM7 | none | CMIP6 | CMIP6 | ECLIPSEv6b |
| CMAM | none | none | CMIP6 | ECLIPSEv6b |
| DEHM | MEGANv2 | none | GFAS | ECLIPSEv6b |
| ECHAM6-SALSA | GEIA inventory (PM only) | 3D emissions based on AeroCOM III | CMIP6 | ECLIPSEv6b |
| EMEP MSC-W | EMEP scheme Simpson et al. (2012) | Degassing from Ethna, Stromboli, Eyjafjallajökull (2010), Grimsvotn (2011), Holuhraun (2014, 2015) | FINN Wiedinmyer et al. (2011) | ECLIPSEv6b |
| FLEXPART | none | none | CMIP6 | ECLIPSEv6b |
| GEM-MACH | BEIS v3.09 | none | CFFEPS | ECLIPSEv6b outside NA, US NEI and Canadian APEI |
| GEOS-Chem | MEGANv2.1 with updates | NASA/GMAO | GFEDv4.1 | ECLIPSEv6b |
| GISS-E2.1 | Guenther et al. (2012) isoprene, ORCHIDEE terpenes, online DMS, SS and dust | AeroCom | CMIP6 | ECLIPSEv6b |
| MATCH | MEGANv2 | climatological and Honoluraun | CMIP6 | ECLIPSEv6b |
| MATCH-SALSA-RCA4 | MEGANv | climatological and Honoluraun | CMIP6 | ECLIPSEv6b |
| MRI-ESM2 | Horowitz et al. (2003) | CMIP6 stratospheric aerosol and Honoluraun | CMIP6 | ECLIPSEv6b |
| NorESM1-happi | Dentener et al. (2006) | CMIP6 | CMIP6 | ECLIPSEv6b |
| Oslo-CTM | MEGAN-MACC at 2010 | AeroCom (Dentener et al., 2006) Andres and Kasgnoc (1998); Halmer et al. (2002) | GFEDv4 | ECLIPSEv6b |
| UKESM1 | isoprene and monoterpenes interactive with land surface vegetation scheme | climatology, CMIP6 | CMIP6 | CMIP6 |
| WRF-Chem | MEGANv2.1 | none | GFED | ECLIPSEv6b |



**Table A1.** Information about models' spatial set-ups.

| Model | horizontal resolution | scale (global or regional) |
|---|---|---|
| CanAM5-PAM | 128×64, Gaussian grid, T63 | global |
| CESM | $1.9° \times 2.5°$ lat/lon grid | global |
| CIESM-MAM7 | $0.9° \times 1.25°$ lat/lon grid | global |
| CMAM | 96×48 Gaussian grid, T47 | global |
| DEHM | 50km, >150×150 gridpoints | Polar stereographic |
| ECHAM-SALSA | T63 | global |
| EMEP-MSC-W | $0.5° \times 0.5°$ regular lon/lat | global |
| FLEXPART | Met. input data: $1° \times 1°$ | global |
| GEM-MACH | $0.1375°$ (or 15-km) | rotated Arctic LAM |
| GEOS-Chem | $2° \times 2.5°$ | global |
| GISS-E2.1 | $2° \times 2.5°$ | global |
| MATCH | 186×186, $0.75°$ | rotated lat/lon regional |
| MATCH-SALSA | 188×198, $0.75°$ | rotated lat/lon regional |
| MRI-ESM2 | TL159(AGCM), TL95(aerosol), T42(ozone) | global |
| NorESM | $0.9° \times 1.25°$ | global |
| Oslo-CTM | $2.25° \times 2.25°$ | global |
| UKESM1 | 145x192 ($1.875° \times 1.25°$ ) | global |
| WRF-Chem | 100 km | regional-Arctic |





**Table B1.** Information about Arctic BC measurements used for model evaluation

| Location or network | Method | Comments/references |
|---|---|---|
| IMPROVE | EC via thermo-optical | Malm et al. (1994) |
| EMEP | EC via thermo-optical from $PM_{2.5}$ and $PM_{10}$ | Tørseth et al. (2012); EMEP (2014) |
| CABM | EC via thermal evolution method from total suspended particle (2005-2011) and $PM_1$ (2011 to present). At Alert, also eBC via aethalometer for PM1 | Sharma et al. (2006); Huang et al. (2006) Huang et al. (2021) |
| Gruvebadet Lab | eBC via PSAP from PM1 | Gogoi et al. (2016) |
| Zeppelin Mountain | eBC via aethalometer | Eleftheriadis et al. (2009) |
| Utqiagvik (aka Barrow) | eBC via aethalometer and via PSAP from PM1 | Delene and Ogren (2002) |
| Japanese Arctic cruise | rBC via SP2 from PM10 | Taketani et al. (2016) |
| Russian Arctic cruise | eBC via aethalometer | Popovicheva et al. (2017) |
| Aircraft campaigns | rBC from SP2 | Moteki and Kondo (2010); Schwarz et al. (2006) Stephens et al. (2003) |





**Table B2.** Information about SLCF measurements from monitoring networks used for model evaluation in the Northern Hemisphere.

| Network acronym | long name | species measured | time period | Comments/references |
|---|---|---|---|---|
| CABM | Canadian Baseline Monitoring network | $SO_4^{2-}$, BC, and OC | 2000 to present | 6 sites in Canada |
| CSN | Chemical Speciation Network | $O_3$, $NO_2$, $SO_2$, CO, $PM_{2.5}$, $PM_{10}$, $SO_4^{2-}$, $NO_3^-$, $NH_4^+$, EC, and OC | 2001 to present | data from the Air Quality System, which centralizes access to numerous U.S. datasets |
| CAWNET | China Atmospheric Watch Network | $PM_{2.5}$, $PM_{10}$, $SO_4^{2-}$, $NO_3^-$, $NH_4^+$, EC/BC, OC, $O_3$, CO, $NO_x$, $CH_4$, $SO_2$ | 2000 to present | |
| EMEP | European Monitoring and Evaluation Programme | $PM_{2.5}$, $PM_{10}$, OC, BC, and $SO_4^{2-}$ | 1993 to present | park of GAW |
| GAW | Global Atmosphere Watch | $SO_4^{2-}$, BC, OC, $O_3$, $NO_x$, CO, $SO_2$, AOD, CN | 1993 to present | a portal of measurements |
| IMPROVE | Interagency Monitoring of Protected Visual Environments | $SO_4^{2-}$, EC/BC, OC, $PM_{2.5}$, $PM_{10}$, and $NO_3^-$ | 1987 to present | Data obtained at http://vista.cira.colostate.edu/Improve/improve-data/ |
| NAPS | National Air Pollution Surveillance Network | $PM_{2.5}$, $PM_{10}$, $SO_2$, $HNO_3^-$, $NH_3$, $SO_4^{2-}$, EC/BC, OC, $O_3$, NO, $NO_2$, $SO_2$, CO | 1974 to present | |
| WHO | World Health Organization (WHO) | $PM_{2.5}$ and $PM_{10}$ | 1985 to 2011 | |