# Peer review of "Model evaluation of short-lived climate forcers for the Arctic Monitoring and Assessment Programme: a multi-species, multi-model study"

_Atmospheric Chemistry and Physics, 2021_

## Author Response (AR1)

Reviewer #1
*This paper summarizes an ambitious multi-model intercomparison and evaluation focused on pollutants and short-lived climate forcers in the Arctic. 18 3D chemistry models are included, with comparisons for 2 time periods 2008-9, 2014-15. The models are evaluated with surface, aircraft and satellite measurements of CH4, O3, CO, NO, BC, PM2.5. This extensive evaluation illustrates that none of the models are particularly better than others for all variables. It appears difficult to draw any overarching conclusions about models in general, or even a subset of them. The paper concludes with a useful list of recommendations for model improvement for representing pollutants in the Arctic.*

[CW] Thank you for your thorough review and your very helpful suggestions to improve the paper.

*To make this paper more accessible to readers, the number of figures in the main paper should be dramatically decreased. I understand the desire to show all the details for each model, but I would move most of the figures to the supplement, and include summary plots (such as scatter plots and Taylor diagrams) in the main text.*

[CW] Thank you for these helpful suggestions on how to better present the results. The paper now has 24 figures: In the revision, 6 figures were moved to the supplement and 1 summary figure added to the main paper. While not a dramatic decrease of the number of figures, we feel that the nature of the study, which reports on the 3-D spatial and temporal characteristics of 18 models for 8 SLCF species, compared to so many different observation types requires many figures to tell the story and to be a helpful reference for later use. It is also helpful for each model development group to have a reference like this showing how their model performs compared to others.

    The revised paper has 2 figs for CH4 (surface, satellite), 3 for O3 (surface, seasonal, satellite), 2 for CO (surface, satellite), 1 for NO2 (surface), 5 for BC (surface, deposition, seasonal, aircraft, ship), 3 for SO4 (spatial, seasonal, deposition), 2 for OA (spatial, seasonal), and 2 for PM2.5 (spatial, scatter plot). There are also 2 multi-species summary figs and 2 observation maps for a total of 24 figures.

    That said, the revised paper has taken into account all of your suggestions to improve the figures. Six figures have been moved to the supplement, which now has 12 figures – up from the original 3 – and some of the remaining figures in the paper were made more concise (e.g. for CO & NO2, where most models have the same spatial patterns, now only show the multi-model median (mmm) biases; and for the BC ship comparison, the spatial model biases were removed from the figure).

*Also, as the paper is focused on northern mid-latitudes and the Arctic, any discussion and plots of the Southern Hemisphere could be left out. Any map plots should be shown as polar plots, but I*

*found most of them not very informative. The current Mercator projection plots (Figs. 3, 6, etc.) leave a lot of blank space with no data.*

[CW] Any remaining maps now only show the northern hemisphere and most are now centered on the North Pole (except for TES CH4 – see response below). The map figures are useful to demonstrating that there are spatial differences in the model biases.

*For methane, the annual mean bias is somewhat informative, but for all of the other compounds with much shorter lifetimes, I do not find the maps of annual mean bias very useful. Figures 7 & 8 are good examples of concisely showing useful information for each model. For example, scatter plots of model vs observations, showing each monthly average point, can give a sense if the seasonal cycle is reproduced by the model, and Taylor diagrams could show that even more concisely.*

[CW] The O3 figure now shows the summer seasonal average instead of the annual average, as summertime tropospheric O3 is more important for health impacts. However, annual averages are useful, even for short-lived species, since climate and chronic health impacts happen over long time periods. Thus, the annual model biases will provide information on how simulated climate and health impacts should be interpreted. We also felt in some cases, that it is important to report on the spatial distribution of the model biases across the Arctic, and it's important to show where the observations are located (in the Arctic they are sparse). Thus, we have kept a number of the map figures in the main paper.

*I realize that Figure 28 (& 29) is a summary of all that has been shown before, but it might be helpful to show that at the start of the paper and then show and discuss select examples that correspond to a row or column of the % bias colored boxes. This might make it easier to read and follow the discussion of each species and/or model.*

[CW] Thank you for this suggestion. We have now moved Fig 28 (and its corresponding subsection) to the beginning of the model evaluation section (now Fig 3), and we've removed Fig 29 since it was redundant. We added another useful summary figure in this section as well (the mmm biases vs latitude), and the discussion in the paper has been updated to reflect this change.

*Some additional minor comments:*

*abstract: last sentence modify to: "These processes..."*

[CW] done.

*Section. 3.2: give versions of satellite retrievals, and reference appropriate papers for that version. For example MOPITT V8 [Deeter et al., https://doi.org/10.5194/amt-12-4561-2019].*

[CW] The versions used are:
        MOPITT, version 8, level 3 gridded product (Deeter et al, 2019)

TES, version 7, Aura level 2 ozone lite nadir (data quality report : https://asdc.larc.nasa.gov/documents/tes/quality_summaries/L2_products_V007.pdf )
ACE-FTS, version 4.1 (with data quality: https://dataverse.scholarsportal.info/dataset.xhtml?persistentId=doi:10.5683/SP2/BC4ATC , and mission overview paper: Bernath et al, 2005)
These have been added to the text in Section 3.2, Satellite datasets.

*Section 4: How are model-measurement comparisons performed exactly? For surface obs are closest model grid box used, or some sort of interpolation? What impact does model resolution have on the comparison?*

[CW] For surface observations, they are compared to the model grid box that they fall in. When more than one surface observation occurs in a model grid box, then those observations are first averaged and then compared to that model grid box. This additional information was added to the beginning of the Model-measurement Comparison section.

*Figures 3 & 4: show only NH?*

[CW] We've adjusted the surface CH4 figure (Fig 3 in original manuscript) to only show the Northern Hemisphere and to be centered on the North Pole. But kept the TES comparison (original fig 4) as is, because for CH4, the discussion in the text revolves around how some of the models don't simulate the north-south gradient of the CH4 distribution properly, despite having accurate total global concentrations. That error in its global distribution explains the origin of low biases from some of the models in the Arctic. For all other species in the paper, we've adjusted the figures to be focused only on the Northern Hemisphere.

*Fig. 6: leave out - hard to see; annual mean differences are not very informative for ozone. More useful results are shown in Fig. 7.*

[CW] We changed the surface O3 figure (original Fig 6) to show just the June July August (JJA) average, which is when O3 is a pollutant of concern, and we've made the data more visible by showing the Northern Hemisphere only, as suggested above. We have kept the figure in the paper because the models have different behaviour in simulating surface O3, which we discuss in the text.

*Fig 9 & 10: hard to read - limit to polar view of NH only?*

[CW] Those figures are now a polar view of Northern Hemisphere. Thank you for this suggestion.

*Fig. 11: Why is the MOPITT data so sparse? What are you actually plotting? How have you filtered the data? It would be far more informative to show evaluation for individual months or seasons instead of a 2-year average.*

[CW] We have plotted the MOPITT, version 8, level 3 TIR/NIR product, and the smoothed model biases in this figure. On shorter time scales, the MOPITT data are even more sparse. The 2-year average has the best spatial coverage, and provides the information about model performance that we need for our study. The level 3 MOPITT data product has been quality-filtered as follows (from the data quality statement: https://asdc.larc.nasa.gov/documents/mopitt/quality_summaries/MOPV8_L3_DQ_statement.pdf):

> The filtering method used for V8 Level 3 processing relies on both pixel filtering and signal-to-noise ratio (SNR) thresholds for Channel 5 and 6 Average radiances (i.e., 5A and 6A). […]
> > V8 Level 3 TIR-only products exclude all observations from Pixel 3 in addition to observations where the 5A SNR < 1000
> > V8 Level 3 NIR-only products exclude all observations where the 6A SNR < 400
> > V8 Level 3 daytime TIR/NIR products exclude all observations from Pixel 3 in addition to observations where both (1) the 5A SNR < 1000 and (2) the 6A SNR < 400
> > V8 Level 3 nighttime TIR/NIR products exclude all observations from Pixel 3 in addition to observations where the 5A SNR < 1000.

*Fig. 15&16: Are these 2 plots showing the same data? Recommend showing only one of them.*

[CW] Yes, they are. We have kept Figure 15 (BC seasonal cycle) in the main paper to emphasize the seasonal cycle analysis, and moved Figure 16 (BC scatter plots) into the supplemental material. We have also done the same for the equivalent $SO_4^{2-}$ and OA plots for consistency.

*Fig. 18: left and right panels are redundant. Just show map of observations, and scatterplots for models. Don't need the map of each model.*

[CW] Thanks for this suggestion for the BC ship comparisons. We have removed the left panel, which showed the model biases on the map, and kept only map of observations and the right panel, as suggested.

*Fig. 24 caption: does not describe what is in figure.*

[CW] My apologies. This has been corrected.

*l.576: typo: "2015 26"*

[CW] Thank you. Corrected.

*Fig. 28 & 29: Aren't these 2 figures showing the same data? Only need one of them. And they are a nice summary of all the preceding plots, so perhaps some of them can be removed or moved to the supplement.*

[CW] Yes, they are. Figure 29 has been removed (since it was redundant with Fig 28) and Figure 28 has been kept in the main paper, but moved earlier, as suggested above.

*l.660: Scandenavia -> Scandinavia*

[CW] Corrected.

***

Reviewer #2

*Review of "Model evaluation of short-lived climate forcers for the Arctic Monitoring and Assessment Programme: a multi-species, multi-model study, by Whaley et al.*
*In this study, Whaley and coauthors carry out a detailed validation of short-lived climate forcers (SLCFs) in an ensemble of model simulations performed for the Arctic Monitoring and Assessment Programme (AMAP). Model simulations are carried out for 4 years: 2008-2009 and 2014-2015, and modeled values of methane, ozone, ozone precursors, black carbon (BC), sulfate, and organic aerosol are all compared to observations. The authors find that in general the multi-model means of these species agree reasonably well with the observations. The authors conclude their paper with several recommendations for future model improvements.*
*The careful validation and analysis of the model output was impressive, and I recommend publication once the minor issues are resolved. The authors make a concerted effort not just to report the discrepancies between observations and models, but also to offer potential reasons for these discrepancies. For example, the discussion of potential errors in transport and deposition was welcome. This study represents an essential first step in efforts to interpret present-day climate change and to project climate change in the future.*

[CW] Thank you for your thoughtful and detailed review.

*Minor issues.*

*Summertime overestimates. The authors conclude that an overestimate of wildfire emissions may account for the overestimates of ozone, BC, sulfate, and PM$_{2.5}$ over the Arctic in summer. Most of the models apparently relied on GFED. Is GFED biased high relative to other standard inventories in the Arctic? (See Liu et al., 2020). Are fires really an important source of sulfate to the Arctic? What about the source of sulfate from dimethylsulfide (DMS)? See update to DMS summer climatology as recommended by Breider et al. (2014).*

[CW] Thank you for these useful references. To start, I note that we discussed on lines 606-609 of the original manuscript that summertime O3, BC, and OA biases implied overestimates of wildfire emissions. Sulfate wasn't mentioned there, though it's true that in the Conclusion on lines 707-709 that we did include sulfate in that list. That is because for all those species, model biases were higher in the summertime at Alaskan sites, and sulfate, specifically, because its emission factor from fires is similar to that of BC (Dentener et al, 2006; van Marle et al, 2017).

However, in the revised manuscript, we removed SO4 from the conclusion bullet point since there is insufficient evidence that model biases in summertime SO4 were due to wildfires.

I see in Liu et al, 2020, Fig 6, that in the BONA (Canada and Alaska) and the BOAS (Russia/Siberia) regions, the GFEDv4s inventory for OC+BC is somewhat high compared to FINN and FEER inventories, but lower than the GFAS and QFED inventories. For EURO, GFED is the lowest one.

All models used GFED/CMIP6, except for EMEP-MSC-W, which used FINN. The summertime biases of EMEP-MSC-W are lower than many other models, but not the lowest (for example, MATCH and MATCH-SALSA have lower biases, even though they use GFED). However, our suggestion about fires was not just based on the emissions themselves, but also because of the uncertainties and model differences surrounding plume height, plume rise, plume chemistry, and wet deposition of wildfire pollutants. For the latter, for example, climate models do not simulate pyrocumulus clouds and can therefore be expected to underestimate the wet deposition due to lack of precipitation. Also, the size of the wildfire aerosol particles is highly uncertain. CanAM5-PAM, for example, specifies the size of the primary wildfire aerosols based on a small number of observations. However, aerosol sizes can change rapidly in wildfire plumes, in response to gas-to-particle conversion of secondary organic species. If those particle sizes are too small in the models, the efficiency of wet deposition would be reduced.

In the revised manuscript, we have added text to discuss the behaviour of the GFED and FINN fire inventories, referencing the Liu et al (2020) paper.

*GEOS-Chem Arctic simulation. Breider et al. (2014, 2017) used GEOS-Chem to analyze seasonal and interannual variability of aerosols over the Arctic. These authors applied several model updates so that the simulated aerosols better matched observations. These updates included an improved treatment of cloud pH and of cloud liquid water in summer over the Arctic, as well as new inventories for BC and SO$_2$. The reader is curious if the version of GEOS-Chem used here included these updates and inventories.*

[CW] Thank you for suggesting a possible reason why our summertime sulfate from GEOS-Chem was too high at the Arctic locations (e.g. Fig 19 in our revised manuscript). The new inventories for BC and SO2 weren't used in these GEOS-Chem simulations since all models used the same set of emissions from ECLIPSEv6b for this study. This new emissions dataset was developed for the AMAP SLCF project, and includes appropriate BC and SO4 decreases in the early 1990s, and is shown in other studies in preparation to match the 1990-2015 time series well. This emissions dataset also includes gas flaring from Russia, which is an important recent improvement.

Our GEOS-Chem simulations were from v12.3.2 of the model, which is a more recent version than the ones used in the Breider et al (2014, 2017) studies (v08-02 and v09-01, respectively), and the aqueous-phase SO2 oxidation in cloud water and treatment in DMS that was developed by Alexander et al. (2012) and used in the Breider et al (2014) study, has been in the model since v09-02. In our GEOS-Chem v12.3.2 simulation, the cloud liquid water and cloud pH variables were offline inputs from the standard meteorological input from GEOS-FP (for 2014-15) and MERRA2 (for 2008-9). We did not scale down the Arctic cloud liquid water content, as was done in Breider et al (2017), which may be a reason why our Arctic surface sulfate

concentrations are biased quite high. We have added the following text into the sulfate discussion in our revised manuscript:

> "The high GEOS-Chem bias in the summertime seen in Figure 19 was first reported in Breider et al (2014) and found to be due to problems with cloud pH and cloud liquid water in the summer over the Arctic. In Breider et al (2017), the summertime Arctic surface SO4 concentrations are reduced by a factor of 2 by reducing the cloud liquid water content to a uniform value of $1\times10^{-7}$ g/m3 north of 65°N in the model. The version of GEOS-Chem in our study is more recent and uses the offline cloud liquid water content from GEOS-FP and MERRA2. We did not scale this variable down, which may be a reason for the high GEOS-Chem sulfate bias in Figure 19."

*Mid-tropospheric ozone. There exists a ~10% low bias in modeled ozone throughout the troposphere over the Arctic and in the mid- to lower troposphere over mid-latitudes. This bias seems rather high, especially given the importance of ozone as a climate forcer in the mid- to upper troposphere. I recommend the authors emphasize this discrepancy and discuss its implications in the Conclusions. Is this issue also related to convection?*

[CW] Yes, it's true that we show in Figure 8 that the average model bias in the free troposphere is approximately -10%. However, that is compared to TES measurements, which have been shown to be biased high by about +13% for free-tropospheric O3 in Verstraeten et al (2013). Therefore, if the TES measurements were bias-corrected, they would agree almost exactly with the multi-model median O3 values, and this is why we haven't emphasized that discrepancy in the discussion and conclusion.

*Figures. There are many figures, and some can be pushed to the Supplement.*

[CW] Thank you. We have moved many figures to the supplement (which now contains 12 total, up from 3), and made other remaining figures more concise. The manuscript has gone from 29 figures originally to 24 figures in the revised manuscript. Given that this study reports on the 3D spatial and seasonal performance of 18 models for 8 SLCF species, many figures are required to tell the story.

*Site names are hard to see in Figure 1.*

[CW] Font size has been increased for better readability.

*The caption should explain grey areas in just some of the maps of Figure 4.*

[CW] We have added to the Figure 4 caption the following: "Gray areas have no data (either from model, TES, or both)."

*Text is too small in Figures 7 and 12.*

[CW] Font size in these figures has been increased.

*Title over right hand panel in Figure 8 has a typo.*

[CW] Thank you! It has been corrected. We also swapped the left and right panels of this figure so that the mid-latitudes are on the left and Arctic on the right, for consistency with other figures in the paper.

*Inclusion of another row for BC deposition would be welcome in Figure 28, and Figure 29 should also include BC deposition.*

[CW] Ok, since we only assessed BC (and SO4) deposition in the Arctic, we can only add those rows to the right hand (Arctic) panel of this figure. This has been done (and Fig 29 has been removed as per reviewer #1's suggestion due to redundancy with Fig 28).

*Writing. Some sentences starting out with "though" or "whereas" are actually sentence fragments. "Thus" is improperly used as a conjunction.*

[CW] The revised manuscript has been checked over for these kinds of grammatical errors and corrected.

*Line 329 has a misplaced "it's," and line 333 has another typo.*

[CW] Thank you. Both corrected.

*References.*

*Breider, T. J., et al. (2014), Annual distributions and sources of Arctic aerosol components, aerosol optical depth, and aerosol absorption, J. Geophys. Res. Atmos.,4107–4124, doi:10.1002/2013JD020996.*

*Breider, T. J., et al. (2017), Multidecadal trends in aerosol radiative forcing over the Arctic: Contribution of changes in anthropogenic aerosol to Arctic warming since 1980, J. Geophys. Res. Atmos., doi:10.1002/2013JD020996.*

*Liu, T., L.J. Mickley, M.E. Marlier, R.S. DeFries, M.F. Khan, M.T. Latif, and A. Karambelas (2020), Diagnosing spatial biases and uncertainties in global fire emissions inventories: Indonesia as regional case study, Remote Sensing Environ., 237, 11157.*

[CW] References from responses:
Alexander, B., D., J. Allman, H. M. Amos, T. D. Fairlie, J. Dachs, D. A. Hegg, and R. S. Sletten, Isotopic constraints on the formation pathways of sulfate aerosol in the marine boundary layer of the subtropical northeast Atlantic Ocean, J. Geophys. Res., 117, D06304, 2012.

Dentener et al, Emissions of primary aerosol and precursor gases in the years 2000 and 1750 prescribed data-sets for AeroCom, ACP, 6, 4321-4344, 2006.

Van Marle et al, Historical global biomass burning emissions for CMIP6 (BB4CMIP) based on merging satellite observations with proxies and fire models (1750-2015), GMD, 10, 3329-3357, 2017.

---

## Author Response (AR2)

**Reply to reviewer #1**
Reviewer comments in *italic* and author response starting with [CW]

*Overall, the authors have done a good job responding to both reviews. The re-ordering of sections and improvements in the figures helps a great deal.*

[CW] Thank you for reviewing the revised paper and for these additional helpful comments to improve the paper.

*However, I still have a couple of suggestions for the paper.*
*Abstract: The Abstract could be improved by including more conclusions from the paper, and more context for this particular work. The first 3 sentences seem quite general and could perhaps be condensed to one introductory sentence.*

[CW] We appreciate the suggestions to add more to the abstract, and we do, following your additional comments below. The first 3 sentences have been reduced to this one sentence: "While carbon dioxide is the main cause for global warming, modelling short-lived climate forcers (SLCFs) such as methane, ozone, and particles in the Arctic allows us to simulate near-term climate and health impacts for a sensitive, pristine region that is warming at three times the global rate."

*It would be good to include highlights of the conclusions that you discuss in section 5, such as what the models do well, the fact that the multi-model mean is appropriate for the radiation and health impacts determined in the AMAP report.*

[CW] We have expanded/added these sentences in the abstract, which include more of our important conclusions from section 5:
"The multi-model mean was able to represent the general features of SLCFs in the Arctic, and had the best overall performance. For the SLCFs with greatest radiative impact (CH4, O3, BC, and SO4), the mmm was within ±25% of the measurements across the northern hemisphere. Therefore, we recommend a multi-model ensemble be used for simulating climate and health impacts of SLCFs.
Of the SLCFs in our study, model biases were smallest for CH4 and greatest for OA. For most SLCFs, model biases skewed from positive to negative with increasing latitude."

*It would also be useful to mention which evaluation metrics were found to be most important and that should be used in future comparisons.*

[CW] We have added these sentences in the abstract:
"The annual means, seasonal cycles, and 3-D distributions of SLCFs were evaluated using several metrics, such as absolute and percent model biases and correlation coefficients."
"… As model development proceeds in these areas, we highly recommend that the vertical and 3-D distribution of SLCFs be evaluated, as that information is critical to improving the uncertain processes in models."

*MOPITT, Fig. 12: I am still surprised that the coverage of MOPITT data is so sparse, and urge the authors to review their processing of the MOPITT retrievals, and their treatment of missing values. Is no data shown for a given grid box if any month of the 2 years has missing data? Perhaps an average of JJA would be appropriate to show here.*

[CW] Thank you for highlighting this issue again. We did have an error related to masking null values in the MOPITT retrievals, which caused no data to be shown for a given grid box if any month of the 2 years had missing data. This has now been corrected and we have also changed the figure to show the June-July-August (JJA) mean instead, which for MOPITT has much better coverage in the Arctic region, and the models are actually biased high. Even at mid-latitudes, in the summertime the models do not underestimate CO like they do in the winter and spring (which had dominated the annual mean), and this is evident in our new plot. Therefore, we have updated the text discussing this figure, and we have put the springtime mean in the supplemental material.